# Non plasmonic semiconductor quantum SERS probe as a pathway for in vitro cancer detection

Rupa Haldavnekar [1,2], Krishnan Venkatakrishnan [1,2,3] & Bo Tan [4]

Surface-enhanced Raman scattering (SERS)-based cancer diagnostics is an important analytical tool in early detection of cancer. Current work in SERS focuses on plasmonic nanomaterials that suffer from coagulation, selectivity, and adverse biocompatibility when used in vitro, limiting this research to stand-alone biomolecule sensing. Here we introduce a label-free, biocompatible, ZnO-based, 3D semiconductor quantum probe as a pathway for in vitro diagnosis of cancer. By reducing size of the probes to quantum scale, we observed a unique phenomenon of exponential increase in the SERS enhancement up to ~$10^6$ at nanomolar concentration. The quantum probes are decorated on a nano-dendrite platform functionalized for cell adhesion, proliferation, and label-free application. The quantum probes demonstrate discrimination of cancerous and non-cancerous cells along with bio-molecular sensing of DNA, RNA, proteins and lipids in vitro. The limit of detection is up to a single-cell-level detection.

[1] Ultrashort Laser Nanomanufacturing Research Facility, Department of Mechanical and Industrial Engineering, Ryerson University, 350 Victoria Street, Toronto M5B 2K3 ON, Canada. [2] BioNanoInterface Facility, Department of Mechanical and Industrial Engineering, Ryerson University, 350 Victoria Street, Toronto M5B 2K3 ON, Canada. [3] Keenan Research Center for Biomedical Science, St. Michael's Hospital, 30 Bond Street, Toronto M5B 1W8 ON, Canada. [4] Nanocharacterization Laboratory, Department of Aerospace Engineering, Ryerson University, 350 Victoria Street, Toronto, ON M5B 2K3, Canada. Correspondence and requests for materials should be addressed to K.V. (email: venkat@ryerson.ca)

Cancer diagnostics with surface-enhanced Raman scattering (SERS) is an effective method of cancer detection because of its ultra-sensitive and analytical nature. Despite the progress in cancer medicine, majority of the cases get diagnosed when the tumor gets metastasized. Since survival of the patient mostly depends on early detection of cancer, the role of early diagnosis of cancer is very crucial[1].

Recently, many groups have investigated plasmon-induced SERS for early detection of cancer. Typically, plasmonic nanoparticles of gold and silver are used for SERS due to their ability to generate strong electromagnetic enhancement[2]. The substantial enhancement observed with plasmonic metal nanoparticles suffers from coagulation[3], selectivity[4], cost, optical loss, limited wavelength range, and adverse biocompatibility[5]. In order for the highly localized Raman hot spots to remain discrete, plasmonic materials often need surfactants for good SERS response. This is very challenging due to uncontrolled agglomeration of these materials[6]. This type of materials also need to be functionalized for specific targeting with SERS active Raman tags[7], which can result in a contaminated spectra adversely affecting the integrity of cellular structure[8]. This limits plasmon-based label-free, multiplex SERS diagnostics[9]. It is necessary to get simultaneous information on multiple biomarkers for robust diagnosis and disease monitoring as identification of specific cancer biomarkers does not provide complete information on a heterogeneous and complex disease like cancer[10]. Since measurement of biomolecules in an intact cell provides more relevant information because of the reporting of local micro-environment along with the molecular nano-environment; in vitro analysis is a more realistic situation than biochemical assays carried out with purified biomolecules in a test tube[11]. There is a need to study a biocompatible, non-plasmonic substrate that can provide substantial SERS response for in vitro cancer diagnosis of cancer.

In the past, SERS obtained with semiconductor-based nanostructures was quite low ($10-10^2$)[12]. Many strategies have been explored recently to improve this performance. Remarkable SERS activity of amorphous ZnO nanocages due to the numerous metastable electronic states facilitating interfacial charge transfer amplifying molecular polarization was reported by Wang et al.[13]. Reports on vibrational coupling between surface defects like oxygen vacancies and molecules and morphology-induced magnification of substrate–analyte molecule interaction enhancing SERS were presented by Cong et al.[14]. Lin et al. reported defect engineering strategy facilitating photo-induced charge transfer in addition to vacancy defect-induced electrostatic adsorption strategy for SERS[15]. Charge transfer efficiency was improved by vibrionic coupling of the conduction and valence band in a molecule–semiconductor system to improve SERS performance by Wang et al.[16]. Facet-dependent SERS effect in semiconductors improving sensitivity due to interfacial charge transfer leading to large molecular polarization was investigated by Lin et al.[17]. So there is an increased interest in exploration of semiconductor-based SERS. The theory based on semiconductor-enhanced SERS is still evolving[18,19]. According to Lombardi and Birke, it is possible to get SERS from semiconductors due to a combined molecule–semiconductor system. The enhancement obtained is because of the unified effect of various resonances existing in the molecule–semiconductor system. These resonances coexist and should not be considered separately. The resultant enhancement was predicted to be of multiplicative nature[19].

Current research with ZnO-based SERS is limited to nanoscale[20]. Since non-plasmonic materials have typically shown poor SERS response at nanoscale, it makes sense to reduce the size of the material to quantum scale to explore the ability for SERS excitation. Properties of material at quantum scale change rapidly due to optical, exciton energy, and quantum confinement as well

as recombination of electron–hole pairs[21]. Use of unique properties of quantum material for SERS biosensing is an emerging field. Graphene quantum dots were explored as fluorescence and Raman probe with one-dimensional nanochains of Fe3O4@Au-mediated SERS for biomolecule sensing[22]. Semiconductor-based non-plasmonic near-quantum-scale structures were used for biomolecule sensing[23]. However, applying this research for in vitro analysis is extremely difficult due to the toxic nature of two-dimensional (2D) quantum dots to biological systems, heat sensitivity, and disturbances due to photochemical effect[24]. ZnO quantum structures show compatibility with complementary metal oxide semiconductor technology for small sensors[25]. High surface area, good crystallinity, and biocompatibility makes it very desirable for multiple applications of sensing and diagnostics[26]. ZnO can dissolve in acidic as well as in basic conditions. So if applied to a tumor cell, there is a very high probability of ZnO getting dissolved into $Zn^{2+}$ and $O_2^-$, which follows use of both the ions by the cells[27]. ZnO has a wide band gap which makes it a very good sensing material. Refractive index of ZnO leads to high light confinement making ZnO as a promising candidate for SERS platform. Many groups have used gold sputtering on ZnO nanostructures or conjugated silver and ZnO nanoparticles for SERS[28,29]. To the best of our knowledge, research on quantum size SERS probe (semiconductor or other material) is yet to be explored for in vitro diagnosis of cancer.

In this study, we have introduced the next generation of label-free, in vitro SERS diagnosis of cancer by reducing the size of ZnO-based probes to quantum scale. The probes were synthesized by femtosecond pulsed laser interaction mechanism. A unique phenomenon of exponential increase in SERS is observed owing to the surface defects like oxygen vacancies and stacking faults. The probes demonstrate SERS at multiple excitation wavelengths. Enhancement factor (EF) $\sim 10^6$ with limit of detection up to nanomolar concentration is achieved, which is significantly high for a non-plasmonic material. In vitro sensing is demonstrated using two cancer cell lines and comparing the results with a non-cancer cell line. Discrimination between cancer and non-cancer is done on the basis of ratio of peak intensities of lipids and proteins ($I_{1445}/I_{1654}$). Analysis with peak positions and intensities of Raman bands demonstrate differences between cancer and non-cancer cells. The results are substantiated by principal component analysis (PCA) and discriminant analysis (DA). Enhanced biomolecular signals due to cellular uptake of quantum probes are evident with limit of detection up to single-cell level. The quantum probes are decorated on a label-free three-dimensional (3D) nano-dendrite platform mimicking the extracellular matrix allowing self-targeting, cell adherence, and proliferation. Label-free, simultaneous multiple SERS detection for in vitro diagnosis of cancer is demonstrated.

## Results

**Formation characterization and classification of quantum probes.** We have successfully engineered 3D assemblies of quantum probe decorated on the nano-dendrite platform with multi-photon ionization mechanism explained in detail in "Supplementary Methods—Formation of the quantum probe." Briefly, multiple formation mechanism was indicated from the presence of several morphologies[30,31]. Figure 1 shows schematic representation of the overall mechanism. The femtosecond pulsed laser ablation causes rapid fluctuation in the temperature of the plume[32]. The free electrons in the outer shell of metal lattice absorb energy during the multi-photon interaction mechanism in femtosecond laser ablation. The avalanche ionization of electrons due to non-linear ionization is followed by very fast energy transfer to the metal lattice causing phase transformation[33].

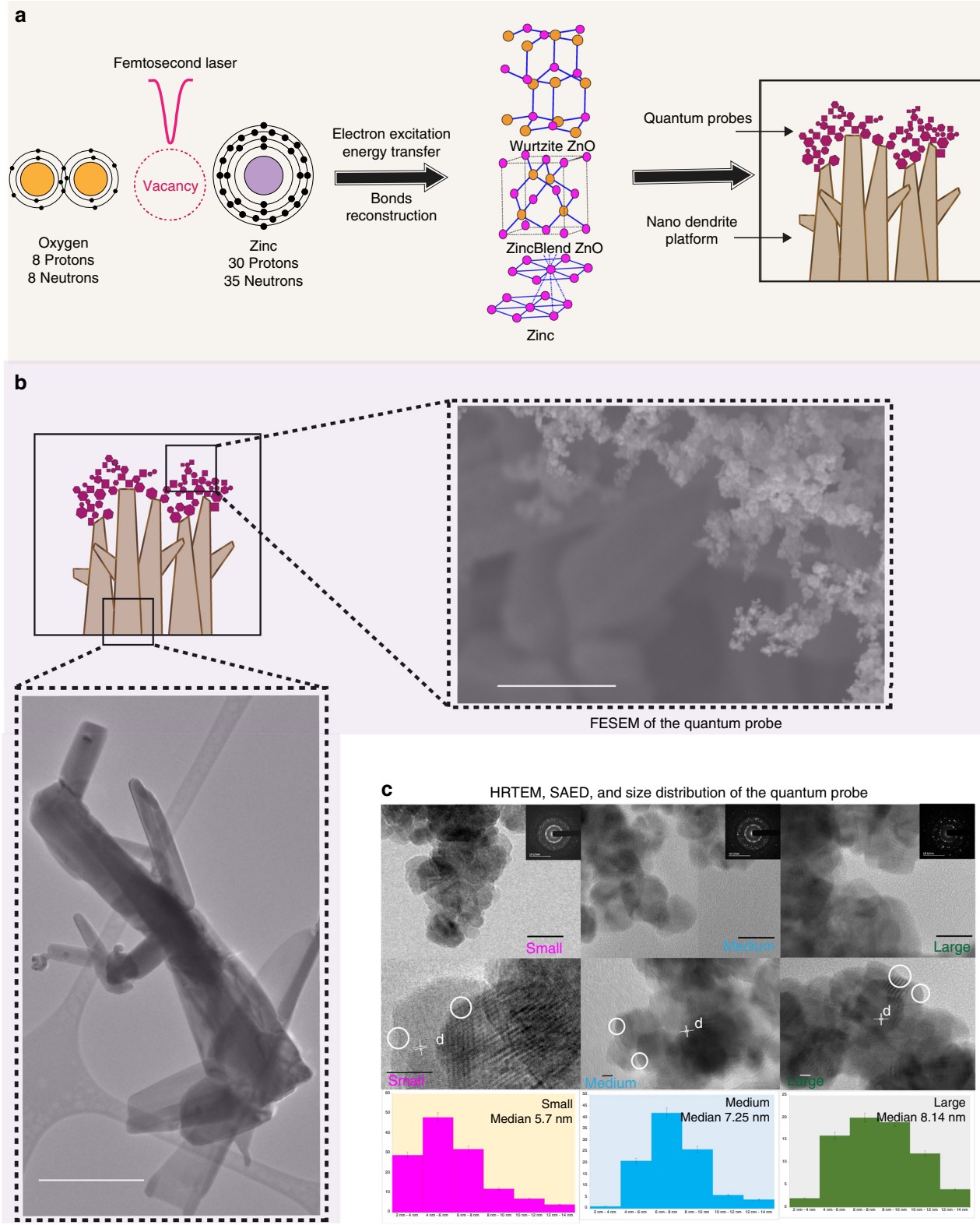

**Fig. 1** Formation and characterization of the ZnO-based semiconductor quantum probe. **a** Formation of ZnO-based semiconductor quantum probe for SERS and nano-dendrite platform for cell adhesion by femtosecond laser interaction. **b** FESEM reveals 3D mesh-like assemblies of the quantum probe. Scale bar = 500 nm. **c** HRTEM shows the presence of hexagonal and cubical particles with stacking defaults and irregular edges, scale bar = 20 nm. Lattice spacing "d" indicating the presence of zinc and wurtzite/zinc blende phases of ZnO, scale bar = 5 nm. The elemental analysis revealed the presence of elemental variation of zinc and oxygen in small (Zn 62.46%, $O_2$ 37.54%), medium (Zn 55.82%, $O_2$ 44.18%) and large (Zn 58.7%, $O_2$ 41.3%) probes. "$x$" Axis displays the size bins and "$y$" axis displays frequency. Particle size distribution charts include error bars in S.D. Total $n$ = 311 samples were measured. Quantum probe was classified into small, medium, and large probes

Breaking of bonds between the top layers of metal lattice as well as the air surrounding the metal occurs due to the released electrons resulting in the formation of ion plume containing charged ions ($Zn^{2+}$, $Zn^+$, and $O^{2-}$) as well as neutral (Zn, O) atoms[34]. Zinc ions and oxygen ions in the ion plume react to form ZnO. Simultaneously, quantum-scaled zinc particles are formed owing to material fragmentation of solid bulk material[35]. Direct vaporization creates rapidly expanding plasma. A critical state is reached by super cooling process, causing nucleation at saturation during super cooling. The seeding layer forms as a result of condensation of nuclei[36]. ZnO nanoparticles self-assemble on the seed layer resulting in the growth of nanorods. The growth of nanorods is mainly due to melting, colliding, and coalescence of nanoparticles[37].

The formation of quantum size crystals may be due to the incomplete coalescence of condensing nuclei giving rise to the formation of networks of quantum-scale crystals (Supplementary Fig. 1). The bonds between the quantum crystals and nanorods were weak and quantum crystals were easily detachable from the nanorods.

Please refer Figs. 1, 2 for characterization of quantum probe and nano-dendrite platform, respectively. Field-emission scanning electron microscopic (FESEM) images reveal the 3D assemblies in the form of mesh-like structures of the quantum probes decorated on nano-dendrites. The probe size varied from 1.7 to 18.99 nm. The crystals were typically hexagonal and cubical in shape and fused together. Stacking faults and defects were evident from the lattice pattern and irregular edges of the crystals, respectively. The lattice spacing "d" indicated the presence of wurtzite as well as zinc blende phases of ZnO[38]. The lattice spacing "d" also revealed the presence of zinc crystals and hybrid crystals with wurtzite and zinc blende phases. Please refer to "Supplementary Table 1" for planer orientations and "Supplementary Fig. 2" for transmission electron microscopy (TEM) of small, medium, and large quantum probes. This type of assembly of multi-phase crystalline structures indicated hybrid morphology of zincite and zinc with polytypism of wurtzite and zinc blende phases[39,40]. The big and blunt nano-dendrites were hexagonal in cross-section, arranged like tetrapod with diameter from 11 to 180 nm. The diameter of sharp and thin nanowires varied from 1.2 to 46 nm. X-ray diffraction (XRD) revealed well-crystallized structures with zincite and zinc phases with lattice constants corresponding to zincite phase of ZnO ($a = 3.25$, $b = 3.25$ and $c = 5.20$). The intensity of the highest peak corresponding to (002) indicated preferential growth along $c$ axis. The average crystallite size varied from 40.8 to 45.4 nm. XRD results were in confirmation with results of energy-dispersive X-ray spectroscopy (EDX) as well as high-resolution TEM (HRTEM).

Raman spectroscopy revealed six prominent peaks at 203, 329, 333, 378, 438, and 568 $cm^{-1}$ and two broad peaks in the range of 1000–1400 $cm^{-1}$[41]. A very sharp peak at 435 $cm^{-1}$ was assigned to $E_2^{High}$. This is a characteristic peak for hexagonal wurtzite structure of ZnO. This peak is also assigned to oxygen vibration. Another prominent peak observed at 568 $cm^{-1}$ was assigned to E1(LO), which is associated with oxygen vacancies. The broadening of this peak can be explained by either change in size and shape of structures or the presence of defects like oxygen vacancies[42]. Optical phonon confinement mode was observed at 378 $cm^{-1}$ assigned to A1(TO) process[43]. Peaks at 329 and 331 $cm^{-1}$ were assigned to 2$E_2$(M) at 329 $cm^{-1}$ and $E_2^{High} - E_2^{Low}$ of multi-phonon processes occurring due to zone boundary phonons. The two broad peaks observed between 1000 and 1400 $cm^{-1}$ were assigned to second-order Raman modes. Red shift was observed in all phonon peaks; this can be due to the spatial confinement in the boundaries of the nanostructures or due to the defects like oxygen vacancies, excess zinc, and surface

impurities[44]. Please refer to "Supplementary Discussion—Raman spectroscopy analysis, Supplementary Fig. 3, and Supplementary Table 2" for more details. In order to investigate presence of oxygen vacancies, X-ray photoelectron spectroscopy (XPS) and electron paramagnetic resonance spectroscopy (EPR) were performed. The Zn2P curve from XPS showed symmetric peaks at 1022.27 and 1045.36 eV. This confirmed formation of hexagonal wurtzite structures. The gap in the binding energies (23.09 eV) fitted with the values from the literature. Please refer "Supplementary Fig. 4," the oxygen curve was fitted into three peaks at 531.05, 531.81, and 532.85 eV for the quantum probe whereas for the native only two peaks were fitted at 530.26 and 532.02 eV. The peaks at lower binding energy were assigned to the presence of $O_2^-$ ions in Zn-O bonding of the wurtzite structure, the peak at 531.81 eV for the nanostructures, and 532.02 eV for native were assigned to C = O bonding due to the surface adsorption of the atmospheric carbon and the peak at 532.85 eV for the nanostructures was assigned to $O^-$ and $O^{2-}$ ion-deficient regions[45,46]. EPR provided "$g$ factor" value of 1.96 indicating the presence of oxygen vacancies[15]. The classification of quantum probe was done on the basis of the particle size into small (median size 5.7 nm), medium (median size 7.25 nm), and large (median size 8.14 nm). The nano-dendrites were classified based on shape and cell-adhesion capability into big hexagonal cell-adhering tetrapod-like nano-dendrites and sharp needle-like cell non-adhering nanowires.

**SERS-based molecular detection.** We were able to achieve the EF of up to $1.37 \times 10^6$ for Crystal Violet (CV) and $2.2 \times 10^5$ for Rhodamine 6G (R6G) (Figs. 3, 4). The limit of detection was up to nanomolar concentration (Please refer to "Methods—SERS based molecular detection" for more details.). CV and R6G physically adsorb on the surface of the sample. At lower concentrations, the enhancement increased indicating gradual decrease in the tilt angle on the substrate, resulting in more surface contact and hence more charge transfer[14]. This can also be the reason for change in the spectral data at lower concentrations. For better understanding of charge transfer mechanism, two thiol-based molecules (4-MBA and 4-ATP) bonding covalently by chemisorption process were used for comparison of the SERS mechanism at millimolar concentration with excitation using 785 nm laser. Please refer Supplementary Fig. 5 and Supplementary Table 3 for EFs. As expected, EF of the chemisorbed molecules (4ATP $3.8 \times 10^6$ and 4MBA $6.9 \times 10^6$) was much higher as compared to the physically adsorbed molecules (R6G $1.71 \times 10^5$ and CV $8.26 \times 10^5$) at millimolar concentration. All four molecules showed not-totally symmetric b2 modes (Please refer to the "Supplementary Table 4" for b2 band assignments of all the molecules.) confirming charge transfer mechanism due to Herzberg–Teller coupling[13,16]. Since all analyte molecules showed high enhancement at the excitation wavelength of 785 nm that was not in the close vicinity of the absorption maxima of the molecules, dominance of molecular resonance was ruled out. In order to further investigate dominant mechanism in the SERS, all four analyte molecules at millimolar concentration were excited at three different wavelengths of 532, 638, and 785 nm. We successfully demonstrated SERS excitation ability of the quantum probe at multi-wavelength. Please refer Fig. 3, wherein all analyte molecules showed maximum enhancement with 785 nm excitation wavelength followed by 638 nm and showed minimum enhancement at the excitation wavelength of 532 nm. The not-totally symmetric b2 modes of all the molecules were clearly visible at all wavelengths and showed maximum intensity at 785 nm excitation. This selective enhancement of b2 modes demonstrated contribution from

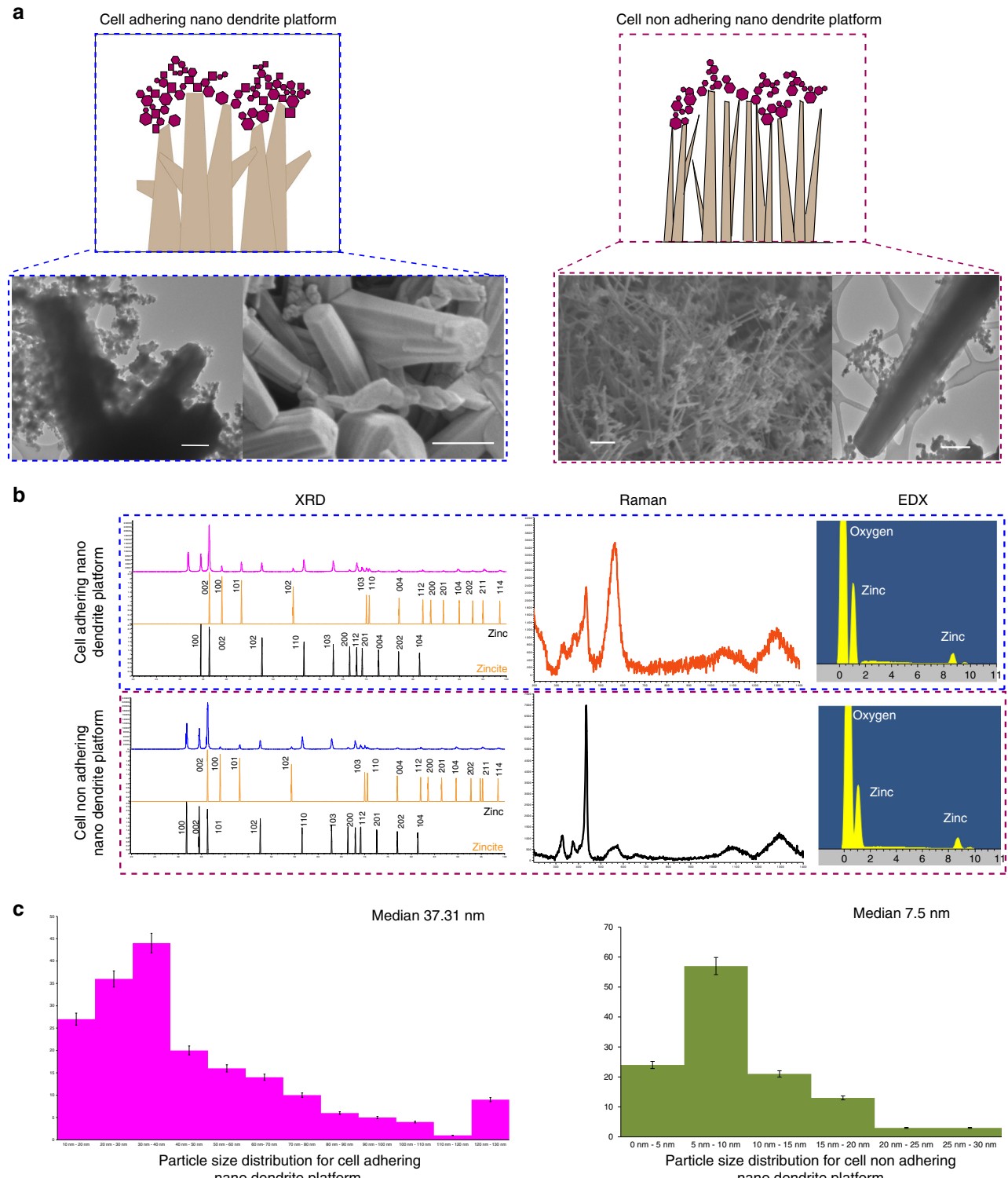

**Fig. 2** Characterization of three-dimensional nano-dendrite platform. **a** FESEM and TEM reveals cell-adhering interface with blunt, tetrapod morphology, whereas non-cell-adhering nanorods have sharp and needle-like structures. Scale bar = 500 nm. **b** Elemental analysis with XRD, Raman, and EDX confirms formation of ZnO material. "*x*" axis displays 2-theta (deg) and "*y*" axis displays intensity (arb. units). **c** "*x*" axis displays the size bins and "*y*" axis displays frequency. Particle size distribution charts include error bars in S.D. Total *n* = 313 samples were measured. Nano-dendrite platform was classified as cell adhering and cell non-adhering platform

charge transfer mechanism due to Herzberg–Teller coupling[19]. Pulsed laser deposition produces very high carrier mobilities. According to the Hamiltonian of semiconductors, very high mobility of charge carriers because of the 3D arrangements of ZnO quantum crystals may result in the combination of exciton and plasmon resonances[47]. The exciton resonance results in SERS due to charge transfer by Herzberg–Teller coupling either from molecular highest occupied molecular orbital to the conduction band edge or from valance band edge to molecular lowest unoccupied molecular orbital. This charge transfer resonance

seems to exponentially increase due to the contribution of interfacial defects like oxygen vacancies[14]. Photo-induced charge transfer also contributed to SERS. The ZnO quantum probes have appearance of defects everywhere in the crystal lattice. We experienced a phenomenon similar to the tip-enhanced Raman

scattering due to the quantum scale of the corners and defects in the edges[48]. The contribution of increased surface charges due to the geometry of the corners can lead to localized and intense electromagnetic fields at the corner apexes. The confinement and intensity of the field can further increase exponentially in the

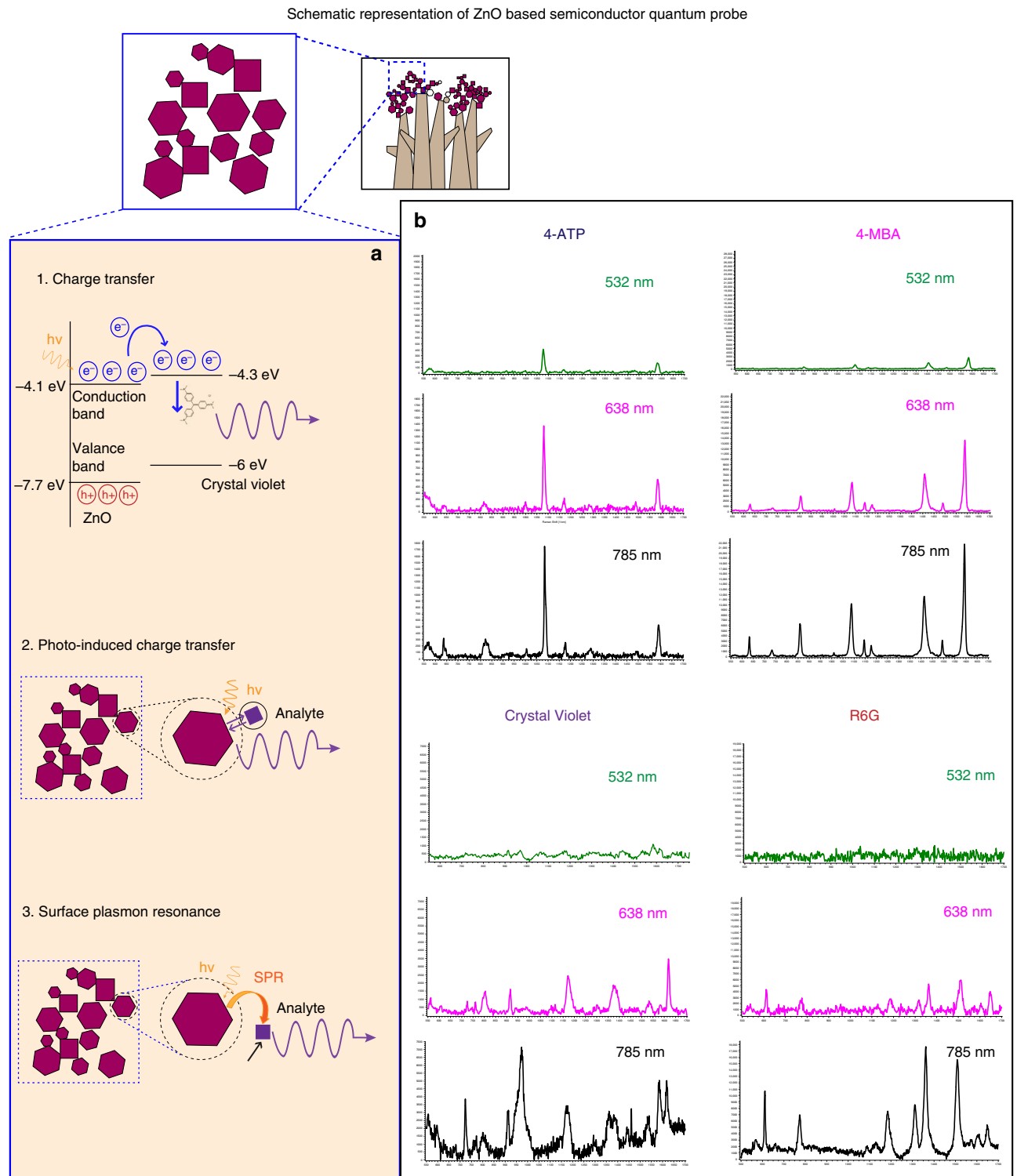

**Fig. 3** The enhancement mechanism and wavelength-dependent SERS. **a** Enhancement was due to the co-existence of (1) charge transfer resonance, (2) photo-induced charged transfer from attenuated total reflection causing near field interaction due to quantum size, and (3) surface plasmon resonance in a unified molecule–semiconductor quantum probe system. **b** The ZnO-based semiconductor quantum probe was able to demonstrate ability for SERS enhancement at multiple excitation wavelengths. The "x" axis of the spectra display Raman Shift (cm$^{-1}$) and "y" axis displays arbitrary units. The evidence of non-totally symmetric b2 modes reveal dominance of the charge transfer mechanism

proximity of quantum-scale apexes of many more quantum probes due to the gap effect[49]. Refractive index of ZnO is estimated to be ~2, which can provide moderate-to-high refractive index contrast with air. Since the quantum crystals are

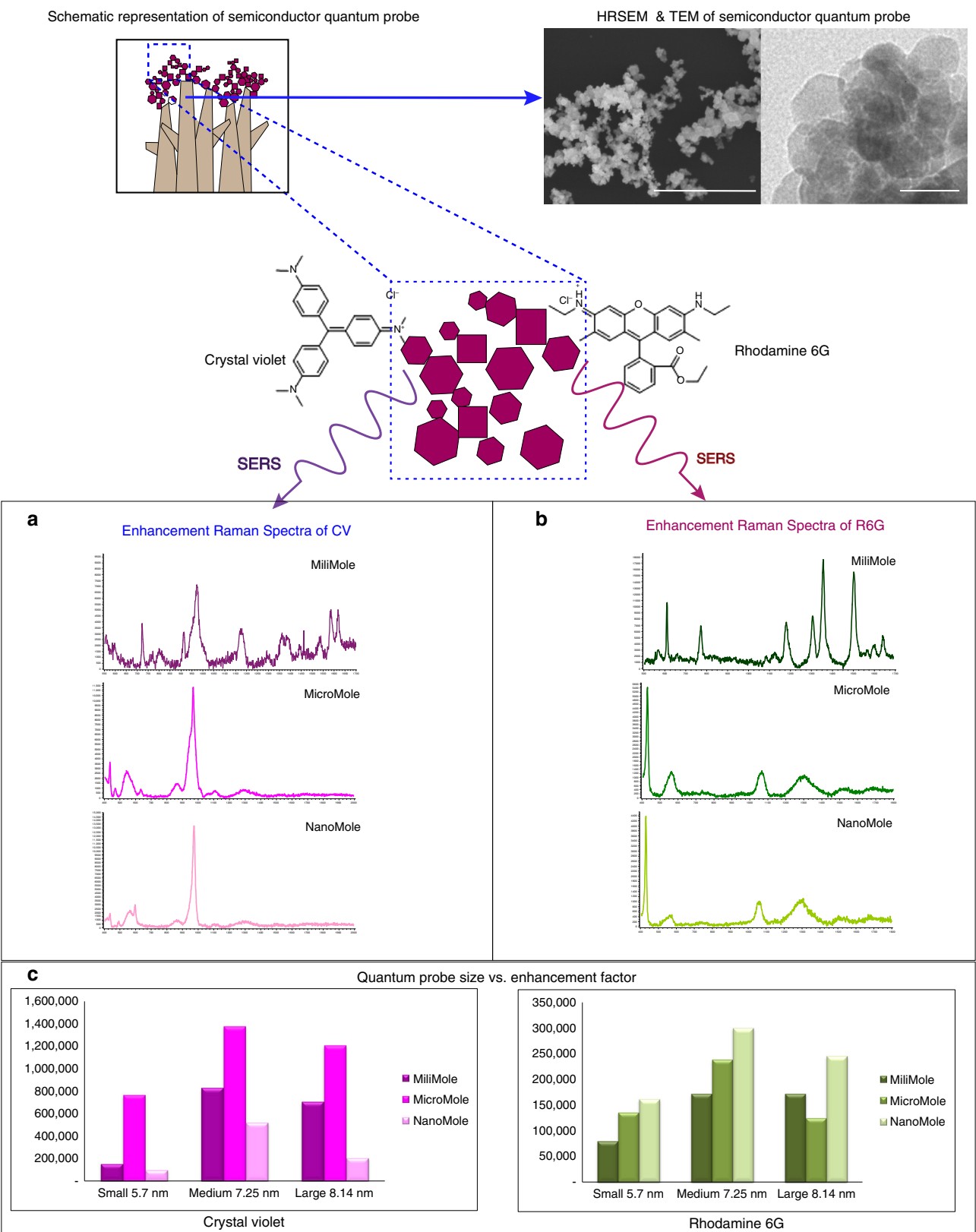

**Fig. 4** SERS-based enhancement for Crystal Violet and Rhodamine 6G. Significant enhancement ~10$^6$ was achieved with ZnO-based semiconductor quantum probe, which is outstanding for a non-plasmonic material. Concentration-based SERS spectra of Crystal Violet (**a**) and R6G (**b**). The "*x*" axis of the spectra display Raman Shift (cm$^{-1}$) and "*y*" axis displays arbitrary units. Bar chart represents enhancement factor with reference to the concentration (**c**). Scale bar for SEM 500 nm and for TEM = 20 nm

arranged in the form of 3D networks, the cumulative contrast gets increased exponentially, which can result in recovery and trapping of most incoming light minimizing Rayleigh scattering and substrate interaction[50]. This kind of effective trapping of photons may have boosted Raman scattering. The total internal reflection due to the evanescent fields may have allowed probing of the surface adsorbates with very high sensitivity providing photo-induced charge transfer.

The plasmon resonance can occur due to Coulomb interaction when electron density is sufficiently high resulting in the observation of collective behaviors of electrons[19]. Pulsed laser deposition results in very high thermal effect, which gives rise to the increased electron density in conduction band. This, in addition to electron density in valance bands which is a dominant factor in semiconductors, generates plasmon[8]. In case of ZnO, at 17 eV[51] the electron density is estimated to be $\sim 4 \times 10^{19}-8 \times 10^{19}$ cm$^{-3}$ carriers, which can display a large oscillation strength to generate surface plasmon resonance at mid-infrared (mid-IR) region[52,53], which seems to be contributing at the near-IR excitation wavelength of 785 nm. This could be the reason for maximum intensity at 785 nm excitation.

We theorize that the resultant SERS obtained with the molecule–quantum probe system was due to the co-existence of charge transfer resonance due to Herzberg–Teller vibrionic coupling and photo-induced charge transfer; surface plasmon resonance and molecular resonance charge transfer being the dominant mechanism for SERS with the ZnO-based quantum probe, which seems to have exponentially increased owing to the presence of oxygen vacancies, surface defects, and stacking faults affiliating interfacial charge transfer.

**Label-free cell adhesion on nano-dendrite platform**. The SEM and fluorescence images reveal that the cells were able to adhere to and proliferate on the nano-dendrite platform. Figure 5 shows that the nano-dendrites with hexagonal, blunt, tetrapod-like morphology act as an extracellular matrix to which cells can easily get attached and adhere to. The lamellipodium extensions of the cells can be seen adhering to the nano-dendrite platform. This is an ideal property of any scaffold[54]. The tetrahedral morphology allows the cells to adhere and migrate. It was important that the nano-dendrite platform had blunt nanorods, as any sharp protrusions can penetrate the cells preventing cell adhesion or causing cell death. The 3D structures mimicking extracellular matrix allowed the cells to easily adhere. The cells appeared flattened and polarized, which is an indication of migration. Minimum or no adherence was observed with long, sharp, needle-like nanorods. Sharp needle-like nanorods did not provide nurturing atmosphere for cell adhesion due to high aspect ratio, high crystallinity, as well as sharp tips.

**Cellular uptake of quantum probes**. The cellular uptake of the quantum probes was through endosomes. Many quantum probes were trapped inside lysosomes dispersed throughout the cytoplasm. Many quantum probes were visible on cell membrane and throughout the cytoplasm. The cellular TEM of cells incubated at 37 °C showed internalization of the quantum probe, whereas at 4 °C the presence of the quantum probes was not evident (Fig. 6). This indicated energy-dependent internalization process of endocytosis. This happened because the active transport mechanism of the cells gets blocked at 4 °C due to less consumption of ATP by the cells demonstrating very less or no quantum probes inside the cells in the TEM[55].

In conclusion, we were successful in functionalizing the nano-dendrite platform for label-free cell adhesion and proliferation. We were also able to program the size and shape of the nano-dendrite platform to enhance self-targeting, cell adhesion, and cell proliferation for label-free in vitro application of SERS. The quantum probes were successfully internalized by the cells to provide in vitro SERS through dominance of charge transfer mechanism.

**In vitro SERS application of ZnO-based semiconductor quantum probe**. Please refer "Supplementary Table 5" for detailed Raman peak assignments.

**Time series analysis with SERS**. Please refer Fig. 7, wherein the peaks' assignment for DNA and RNA was clearly visible in the spectra, but no evidence of the quantum probes within the nucleus (TEM) was present for the 6-h sample. Thus we theorize dominance of SPR for SERS signal at 6 h. TEM shows the presence of quantum probes in the nucleus at 24 h and more quantum probes in the cytoplasm as compared to the 6-h sample. Presence of increased quantum probes may have enhanced the charge transfer mechanism indicating increased SERS intensity. Substantial increase in the SERS signal at 48 h can be due to the presence of a large amount of quantum probes in cytoplasm as well as nucleus increasing contribution of the charge transfer mechanism and hence more intense signal. Another reason could be the growth and proliferation of cells in 48 h, and more cells providing higher intensity of the signal.

Figure 8 shows the presence of a peak at 520.2 cm$^{-1}$ indicating phosphatidylinositol, which is a phospholipid, a key constituent of the cell membrane. This peak was red shifted for 12- and 24-h spectra toward 564 cm$^{-1}$. This can be mainly due to stretching of membrane lipids and lipid vibrations. The peaks at 974 cm$^{-1}$ in combination with the peak at 915 cm$^{-1}$ was assigned to RNA mode. In the region of 2920–2960 cm$^{-1}$, overexpression of C-H vibrations in lipids and proteins and CH$_2$ asymmetric stretch was indicated. On applying PCA, it was observed that first two PCs F1 and F2 were the most dominant. First two PCs were used for analysis with supplementary data point of the strongest RNA peak at 917 cm$^{-1}$. The centroids show a clear separation, proving significant clustering of periodic data. The 3D graph of F1 to F2 to F3 was also plotted to visualize the clustering. The spectra for the 12- and 24-h samples form a mixed cluster in a 3D plot while clusters of 6- and 48-h samples are clearly separated, indicating similar biomolecular activities during the 12- and 24-h samples while the activities are significantly different at 6 and 48 h.

**In vitro cell discrimination by ZnO quantum probe**. Figure 9 shows distinct spectral differences for cancer cells and non-cancer cells[56]. Breast cancer (MDAMB231) showed strong vibrational modes amide I and amide III peaks in the region of 2800–3000 cm$^{-1}$. Presence of excess lipids indicated by peaks at 879, 1095, 1304, and 1450 cm$^{-1}$ were marked as a signature of malignancy in breast cancer cells[57]. HeLa cells showed high concentration of *trans* form 1667 cm$^{-1}$ and *cis* form 1660 cm$^{-1}$ unsaturated lipids[58]. Peak at 1304 cm$^{-1}$ was due to in-phase CH$_2$ twist mode while the peak at 1439 cm$^{-1}$ was due to bending modes occurring due to degenerated deformation of hydrocarbon chains in HeLa. Also high concentration of lipids—1304 cm$^{-1}$ CH$_2$ deformation (lipid), 1313 cm$^{-1}$ CH$_3$CH$_2$ twisting mode of collagen (lipid), 1319 cm$^{-1}$CH$_3$CH$_2$ twisting (collagen assignment), and 1324 cm$^{-1}$ CH$_3$CH$_2$ wagging mode present in collagen—was expressed by HeLa. At 1000 cm$^{-1}$, cancer cell lines showed more intense, almost identical peak for Phenylalanine, for proteins. Cancer cell lines showed CH$_2$ deformation at 1438 and 1454 cm$^{-1}$. Cancer cell lines showed high concentration of nucleic acid at 1662 cm$^{-1}$. In the region of 2920–2960 cm$^{-1}$, cancer cell lines showed overexpression of C-H vibrations in

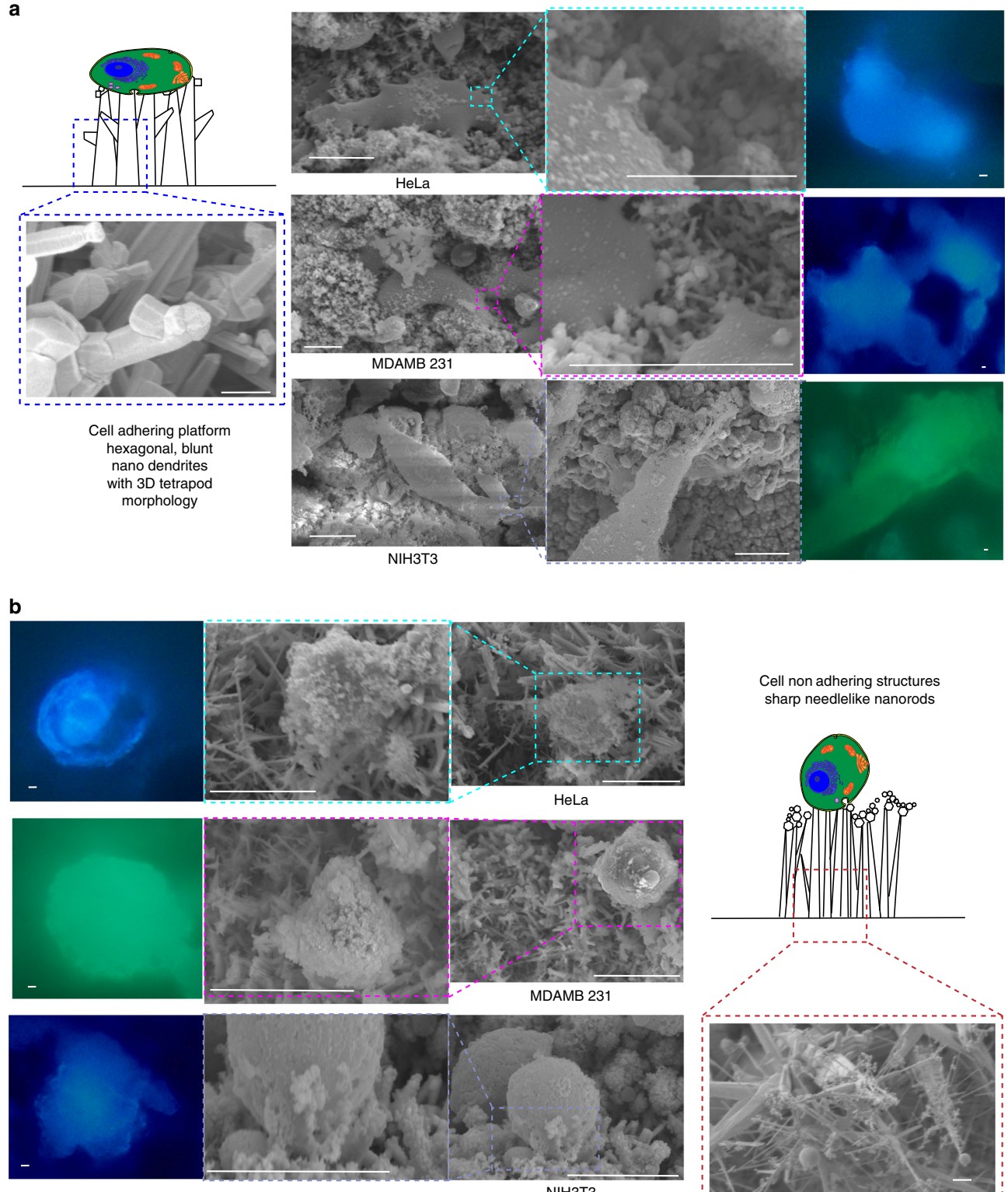

**Fig. 5** Functionalization of the label-free nano-dendrite platform for cell adhesion. **a** Three-dimensional tetrapod morphology of the nano-dendrites allowed cells adhesion and proliferation. **b** Sharp needle-like nano-dendrites did not allow cell adhesion. Scale bar for nanostructures SEM 500 nm, for cell SEM 10 μm, and for fluorescence cells 10 μm

lipids and proteins and $CH_2$ asymmetric stretch. Cancer cell lines showed overexpression of cholesterol and cholesterol ester at 418, 429, 538, 548, 608, 614, 960, 1440, 1441, and 1444 $cm^{-1}$, respectively, indicating high concentration of cholesterol, signature of cancer due to mitochondrial membrane damage. At 1450 $cm^{-1}$, cancer cell lines show $CH_2$ bending mode in malignant tissues that is absent in the non-cancer cells. The non-cancer NIH3T3 cells showed de-formative vibrations of quinoid ring at 1161 $cm^{-1}$ and antisymmetric phosphate stretching vibration at 1230 $cm^{-1}$, which were absent in cancer cells. Peak at 1750 due to

C=O for lipid in normal tissues showed low intensity in cancer cells.

The classification between "cancerous" and "non-cancerous" was done based on ratio of Raman intensities at 1445–1655 cm$^{-1}$. This ratio of Raman intensities was used to classify malignant tissue vs normal tissue regularly[59–61]. Both the bands are indicative of histological abnormalities, providing information about lipid-to-protein ratio. The ratio was very low for cancer cells as compared to fibroblast cells, which confirms the previous analysis of high concentration of lipids among cancer cells. The

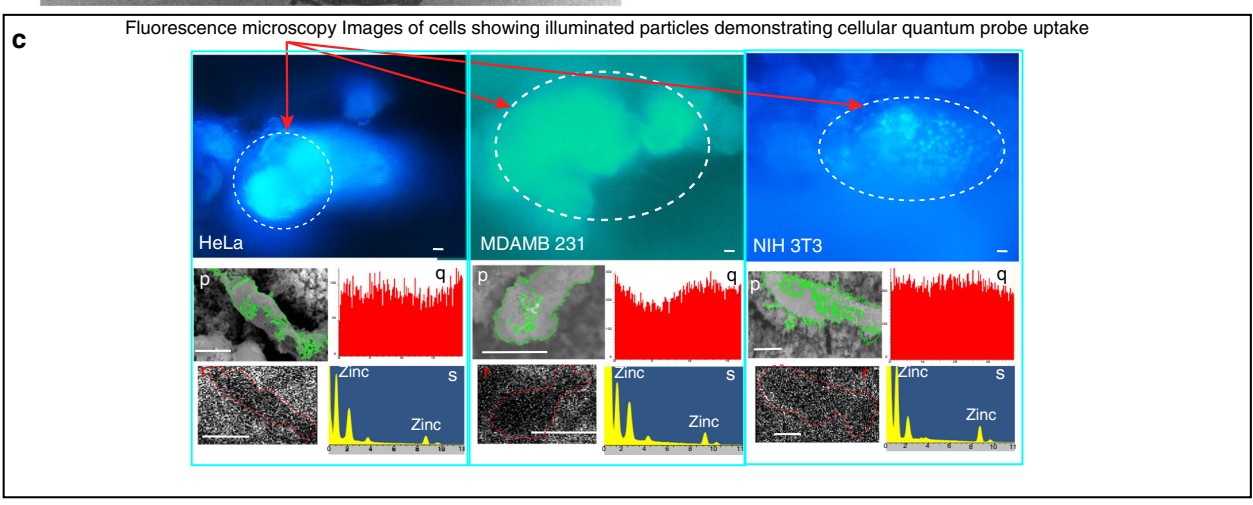

mean value of the ratio was $1.00 \pm 0.11$ for HeLa and $0.82 \pm 0.19$ for MDAMB, separating clearly from non-cancer cell line of NIH3T3 $2.34 \pm 2.26$.

The cancer and non-cancer classification was further substantiated by two multivariate analyses (PCA and DA). Eigen values of F1, F2, and F3 were used for a 3D scatter plot of F1 vs F2 vs F3. F1 interpreted 45.96% of variances, F2 interpreted 20.67% of variances, and F3 interpreted 14.02% of variances. Top 3 PCs contributed to 80.68% of cumulative contribution covering most of the information from original spectra in the 3D space (Please refer "Supplementary Fig. 6" and "Supplementary Table 6" for more details.). Scores of F1 provided reasonable classification among the three cell lines. Clusters of all three cell lines were clearly defined. Please refer to "Supplementary Table 7" for factor F1 loading indicating collagen ($1439 \, cm^{-1}$) as the highest discriminating factor. The peak at $917 \, cm^{-1}$ that belongs to the ribose vibration of RNA was explored to achieve discrimination among cancer and non-cancer cells. Clear clustering based on cell type was observed. It can be seen from the centroids that HeLa and MDAMB (cancer cells) show more activity as compared to normal cells. This can also be interpreted as more tumor-suppressing genes from mRNA are altered in cancers as compared to normal cells. DA provided 100% classification rate with first two functions F1 and F2 demonstrating clear clusters based on cell type. The classification rates were 100% with the first two discriminant functions F1 and F2. For more details, please refer to "Supplementary Discussion—Multivariate analysis of Raman spectra for discrimination of cell lines."

**In vitro biomolecule analysis by ZnO quantum probe**. Please refer Fig. 7, wherein the ZnO quantum probe provides 2–3 times enhanced signal for DNA and RNA peaks, 5–6 times enhanced signal for protein peaks, and 10–15 times enhanced signal for lipid peaks. Cell TEM display more number of quantum probes in the cell membrane and cytoplasm as compared to the nucleus. Since the quantum probes needed more time to penetrate the nucleus as compared to the cytoplasm, the number of probes inside the nucleus is far less than in the cell membrane and cytoplasm. This could be the reason for the highest enhancement of signal of lipids, which are located in or near the cell membrane and lowest for the innermost part of a cell—nucleus. SERS intensity for cellular biomolecules is less as compared to the experiment based on tag molecules. More intense signal can be achieved by incubation with more concentration of quantum probes or incubation for more time enabling increased cellular uptake for more intensity.

The nano-dendrites providing maximum enhancement for CV and R6G could not provide maximum enhancement for in vitro SERS. This happened because the cells displayed less or no ability of adherence and proliferation to the nano-dendrite platform, with sharp needle-like morphology displaying maximum SERS for the tag molecules. This signifies the importance of functionalization of the nano-dendrite platform for cell adhesion. This method is expected to be universal for discrimination among various cancer cell lines. It is evident from Fig. 10 that the

quantum probe displays varied cellular affinity with different results of SERS enhancement. The selection of the optimal quantum probe will have to be based on effective cellular uptake of the quantum probe. This mechanism will depend on the size of cellular pores, so by understanding the relationship between the pore size and quantum probe size, it is possible to engineer optimum quantum probes for maximum cellular SERS response.

**In vitro single-cell Raman mapping with ZnO quantum probe**. In order to obtain the limit of detection, Raman mapping of cells was undertaken[11,62]. From the Raman map (Fig. 10), more presence of proteins and lipids was indicated. Since proteins were present mostly in cytoplasm and lipids in the cell membrane, more affinity of the quantum probe to cytoplasm as well as cell membrane is indicated from the color brightness related to signal intensity[63]. We were able to obtain the map of a single cell demonstrating the ability of SERS from the ZnO-based semiconductor quantum probe for single-cell analysis[64].

Since the quantum probe demonstrated dominance of the charge transfer for SERS, the aggregation of quantum probe in the solution or in the lysosomes of a cell played no adverse effect on the enhancement. This was evident from the many spectra obtained for the entire single cell, which did not show any sudden increase or decrease in the peak intensities (Fig. 10).

## Discussion

We have successfully introduced a biocompatible, label-free, ZnO-based, 3D semiconductor quantum probe as a pathway for in vitro diagnosis of cancer. By reducing the size to quantum scale and engineering 3D assemblies of quantum network, very high SERS for multiple, simultaneous signals was achieved from a label-free, non-plasmonic material. Multi-wavelength excitation was demonstrated with substantial EF. The level of detection was demonstrated at nanomolar concentration of the analyte molecules. The in vitro application was demonstrated by cell discrimination and time series analysis applying several statistical techniques like combination of multivariate analyses of PCA and DA as well as ratio analysis. The ability of the quantum probes for multiple, simultaneous SERS diagnosis, cell discrimination as well as biomolecular detection up to single-cell-level detection was demonstrated with a ZnO-based non-plasmonic, semiconductor quantum probes. In addition to the enhanced Raman sensitivity, the quantum probes decorated on a 3D nano-dendrite platform promoted cell adherence and proliferation because of its structural similarity to the extracellular matrix. Thus the ZnO-based semiconductor probe has dual functions of SERS enhancement and cell-friendly agent. A non-plasmonic, label-free ZnO-based semiconductor quantum probe decorated on a nano-dendrite platform displayed multi-wavelength SERS ability, significant SERS enhancement, and single-cell-level in vitro detection.

## Methods

**Synthesis of label-free SERS probe**. Zinc sheet was purchased from McMaster-Carr and cut into small sample pieces so as to fit in petri plates. Nanostructures were synthesized using Clark-MXR IMPULSE pulsed Yb-doped fiber-amplified femtosecond laser to ionize zinc sheets in ambient conditions[65]. Following

**Fig. 6** Cellular uptake mechanism of the quantum probe. **a** Schematic representation of the endocytosis mechanism. **b** Quantum probe was internalized by energy-dependent endocytosis. (i) TEM images of cells reveal cellular uptake and internalization of the quantum probe. Scale bar = 2 μm. (ii) Cell TEM images reveal evidence of the quantum probe dispersed in the nucleus of cells. Scale bar = 500 nm. (iii) TEM images of the cells incubated 4 °C did not show cellular uptake of the quantum probe, indicating ATP-driven endocytosis mechanism. Scale bar = 2 μm. **c** Fluorescence images showing illuminated particles, indicating internalization of the quantum probes. Scale bar = 10 μm. (p) SEM of cells (scale bar = 10 μm), (q) line EDX ("x" axis is displayed in μm and "y" axis displays arbitrary units), (r) map EDX (scale bar = 10 μm), (s) elemental EDX ("x" axis is displayed in keV and "y" axis displays intensity) of cells outlined in green. Evidence of the presence of the quantum probe within the cells (p, q, r, s). Scale bar for p and r, 10 μm

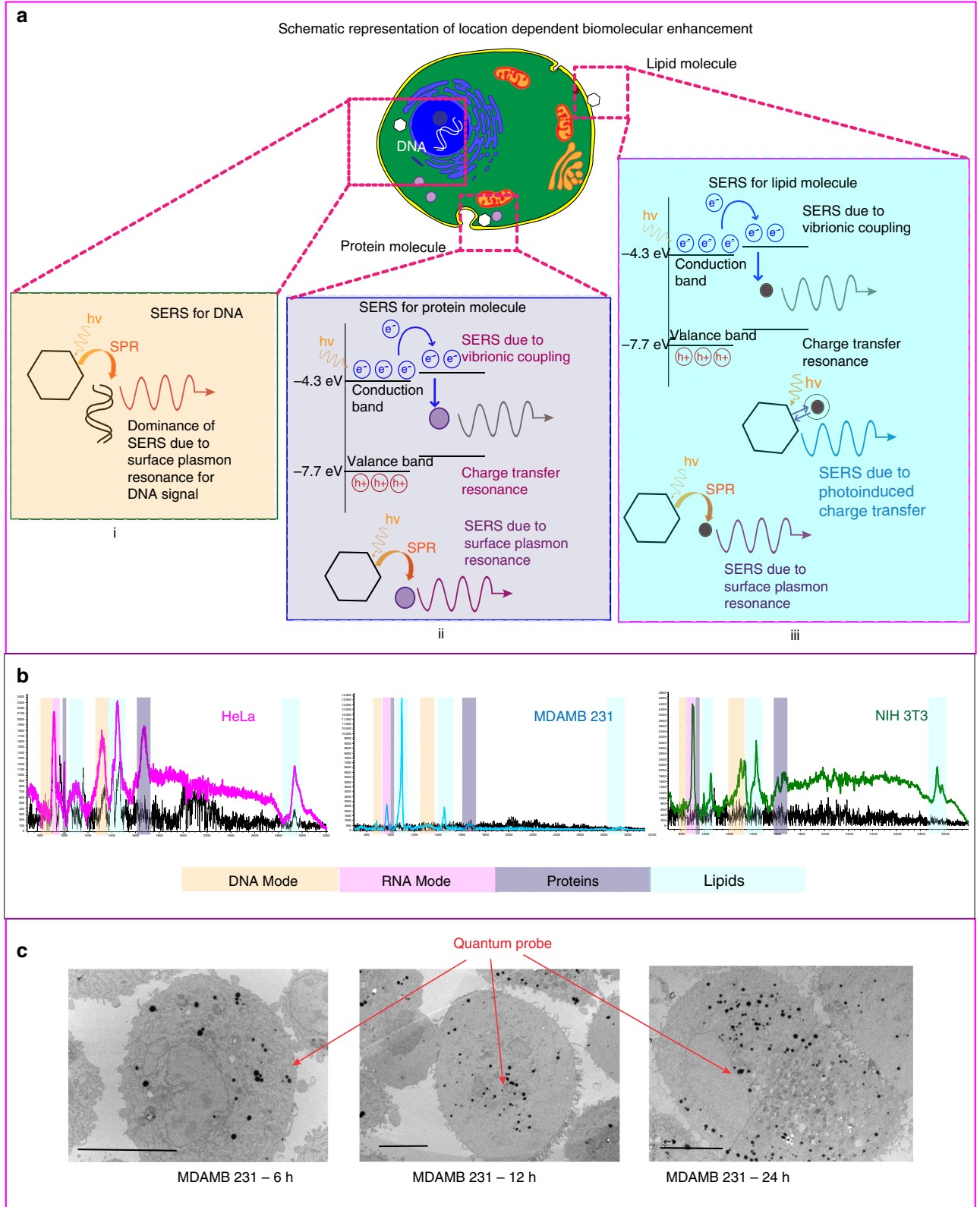

**Fig. 7** In vitro SERS with ZnO-based semiconductor quantum probe. **a** Schematic representation of biomolecular enhancement based on dominance of various resonances due to the location of the biomolecules inside the cells. (i) Less probe inside the nucleus due to time-dependent cellular uptake. Less enhancement as less availability of the charge transfer resonance providing 2–3-fold enhanced signal. (ii) Cytoplasm was benefited by the cumulative resonances of charge transfer as well as surface plasmon providing 5–6-fold enhanced signal for proteins. (iii) Maximum enhancement was observed for lipids in the close proximity of the quantum probe, generating 10–15-fold enhanced signal. **b** Enhanced SERS signal for cancer and non-cancer cells. Magenta, cyan and green represent SERS signal and black spectra for non-SERS response. The "x" axis of the spectra display Raman shift (cm$^{-1}$) and "y" axis displays arbitrary units. **c** Cell TEM reveal time-dependent cellular uptake of the quantum probes. Scale bar = 10 μm

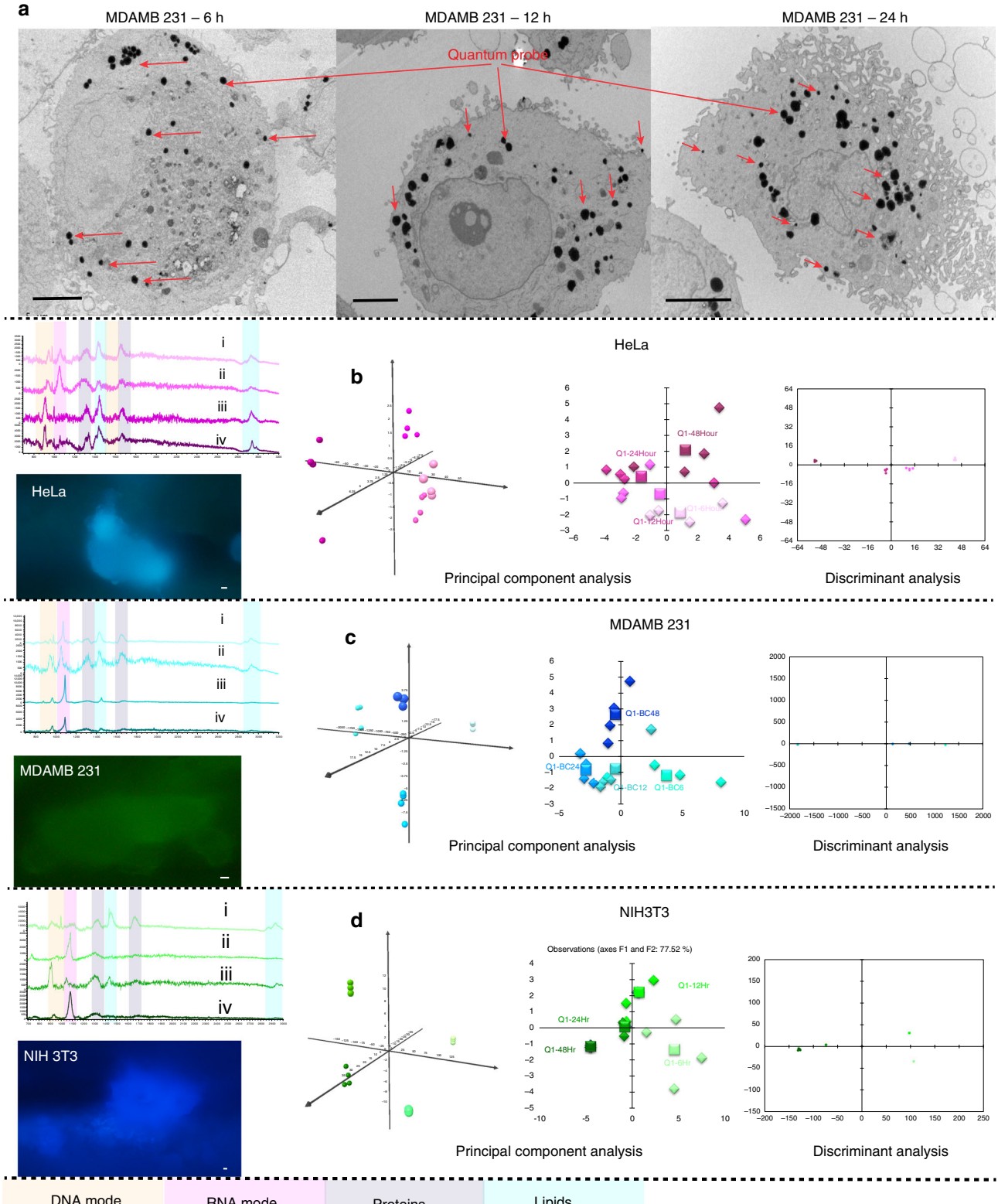

**Fig. 8** Time-based study of cancer and non-cancer cells using multivariate analysis. **a** TEM images of cells showing cellular uptake and internalization of the quantum probe. Internalization was time dependent. Scale bar = 5 μm. Scale bar for fluorescence images 20 μm. 6 h (i), 12 h (ii), 24 h (iii), and 48 h (iv) SERS spectra in **b**, **c**, **d**. The "x" axis of the spectra display Raman shift (cm⁻¹) and "y" axis displays arbitrary units. Time-dependent PCA and DA of HeLa (**b**), MDAMB231 (**c**), and NIH3T3 (**d**). PCA displays 3D plot of principal components F1 on "x" axis F2 on "y" axis and F3 on "z" axis as well 2D plot of F1 on "x" axis and F2 on "y" axis. Clear clustering and separation of centroids is evident. DA displays discriminant components F1 on "x" axis and F2 on "y" axis. Various biomolecular modes show sinusoidal enhancement trends at 12h and 24 h. The intensity increased at 48 h. This may be due to the increase in the quantity of the probe within the cells

parameters were kept constant—power at 16 watt, laser wavelength at 1030 nm polarized circular, and pulse width at 214 fs. An array of 20 lines of 5 mm length were fabricated. EzCAD software in combination with piezo-driven scanner was used for plotting the array. Different scanning speeds and repetition rates were used for fabrication. Control on etch depth was achieved by changing scanning speed. By changing repetition rate, control on generation of peak power for each

pulse can be achieved. Hence, the energy transmitted to the sample surface can be controlled. That way, the morphology of nanostructures can be programmed.

**Characterization of ZnO quantum probe and nano-dendrite platform.** Characterization of quantum probe was done using FESEM, EDX, and TEM.

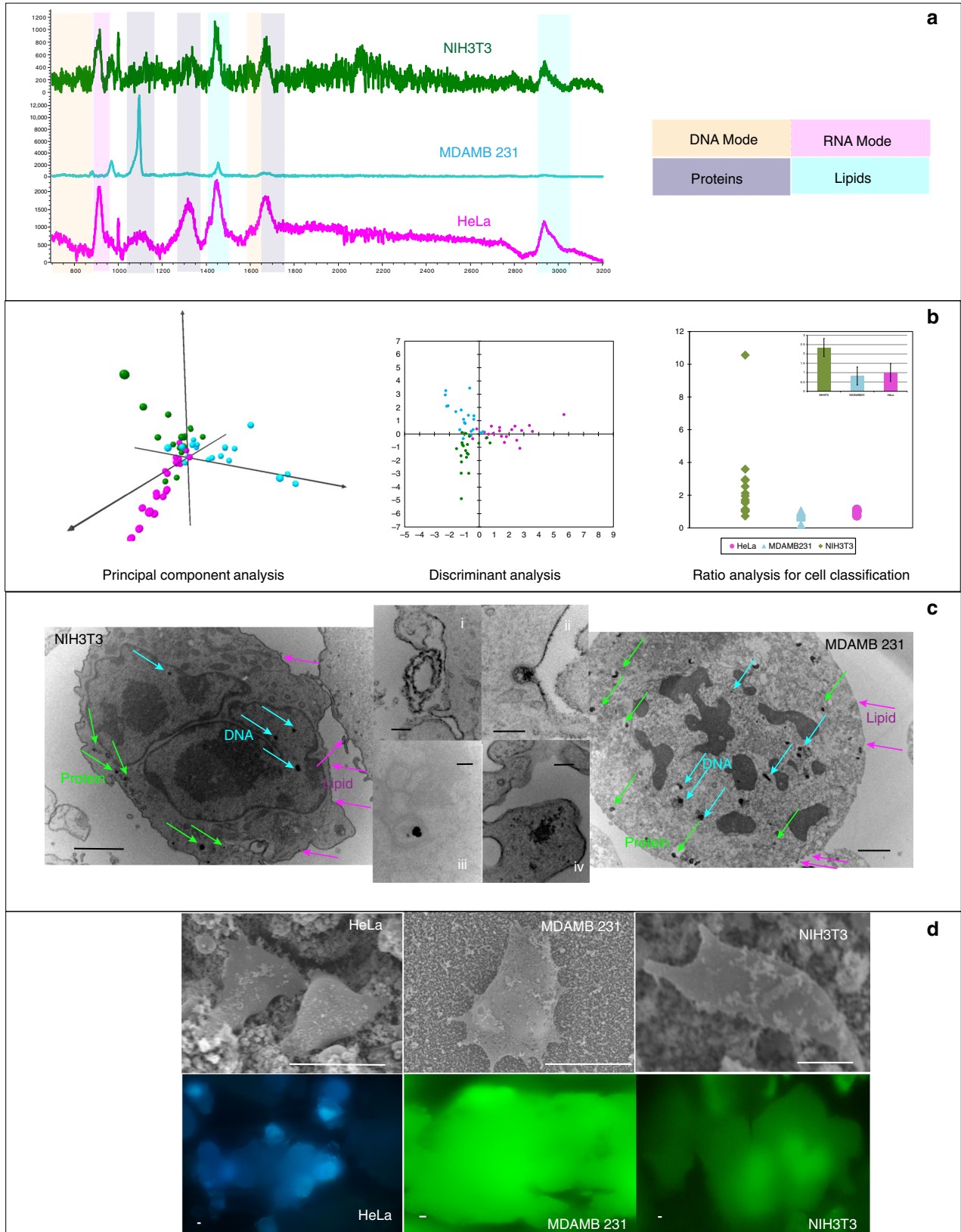

Characterization of nano-dendrites was done with FESEM, EDX, TEM, XRD, and Raman spectroscopic analysis. Additional characterization for oxygen vacancies was carried out using XPS and EPR. Morphology of the samples was analyzed using FESEM images. Elemental analysis was done using XRD and EDX. Crystallite size and crystallinity of the particles was determined using XRD. Particle size was determined using TEM and phases of metal oxide were determined using XRD and TEM lattice spacing "d". HRTEM analysis was done in order to get information about shape, size, and inter-planer distances of lattice of the quantum probe[66]. Images were analyzed using the ImageJ software. The nanostructures formed were then examined using BWTEK Handheld NanoRam device and Renishaw inVia Raman spectroscope at 785 nm wavelength. Since our experiment was for cell lines in vitro, 785 nm was chosen for its compatibility for biological samples. Raman spectra of native sample and sample with nanostructures were obtained

**SERS-based molecular detection**. CV and R6G at 1 millimolar, 1 micromolar, and 1 nanomolar concentration and 4-ATP and 4-MBA at 1 millimolar concentration were used in this experiment. CV is used for histological staining and R6G is used as a tracer dye in biotechnology. Both the dyes (physically adsorb on the substrate) are also used as SERS reporters. So, by calculating SERS with CV and R6G, we can indirectly sense biomolecules. Additional experiment with two chemisorbed analyte molecules (4-ATP, 4-MBA) at millimolar concentration and 785 nm excitation wavelength was undertaken in order to validate the charge transfer mechanism. Experiment with three different excitation wavelengths (532, 638, 785 nm) was carried out in order to understand the dominance of one of the resonances in the SERS mechanism for all four analyte molecules at millimolar concentration.

Ten micrograms of the solution was dropped directly on the nanostructures and native substrate. The solution was allowed to dry and Raman spectra of the air-dried samples were taken. Renishaw inVia Raman microscope was used for this analysis. Excitation wavelength of 532, 638, and 785 nm were used. Other parameters like HC plan APO lenses with matched polarizer/analyzer optics (magnification of ×20, N.A. 0.70); inspected spot size of 0.625 µm radius; focal length of 250 mm; solid state laser with excitation wavelengths 532 nm (5 mW), 638 nm (5 mW), and 785 nm (12.5 mW); matched polarizer optics power of lasers; spectral resolution of 0.5 cm$^{-1}$ in visible and 1 cm$^{-1}$ in near-ultraviolet and IR; and spatial resolution: <1 µm (lateral), <2 µm (depth) were used.

The EF can be derived from the following equations.

$$\text{Enhancement factor} = \left(I_{\text{Quantum probe}}/I_{\text{Native}}\right) * \left(N_{\text{Native}}/N_{\text{Quantum probe}}\right) \quad (1)$$

$$N_{\text{Quantum probe}} = I' \times \pi \times r^2 \text{ where } I'$$
$$= 1/(\text{surface density} * \text{Avogadro number}) \, \text{mole cm}^{-2} \quad (2)$$

$$N_{\text{bulk}} = \pi r^2 h \rho / \text{atomic weight} \quad (3)$$

This equation was derived from the EF calculation used in the article "Cyclic electroplating and stripping of silver on Au@SiO₂ core/shell nanoparticles for sensitive and recyclable substrates of surface-enhanced Raman scattering" by Li et al.[67]. $I_{\text{Quantum probe}}$ and $I_{\text{Native}}$ are the intensities of the characteristic peaks associated with the analytes adsorbed on the quantum probe. Peak at 1621 cm$^{-1}$ is assigned to the strongest CV vibrational mode of c-c ring stretching. (In this case, this peak was red shifted to 1618 cm$^{-1}$.). This peak was used for calculating enhancement at the millimolar concentration. For micromolar and nanomolar concentrations, peak at 1621 cm$^{-1}$ disappeared. So we have used the intensity of peak at 917 cm$^{-1}$ assigned to ring skeletal vibration for the other two concentrations. For R6G, the vibrational mode at 1512 cm$^{-1}$ was used to calculate the EF for millimolar concentration. But this peak is absent at micromolar and nanomolar concentrations, so the peak at 1276 cm$^{-1}$ assigned to C-O-C stretching was used. $N_{\text{Quantum probe}}$ is calculated based from the estimation of concentration of the analyte on the surface using surface density of the quantum probe. $N_{\text{bulk}}$ is calculated using $\rho$ the density of the molecule of the analyte, atomic weight, and $h$ the confocal depth of the Raman laser beam.

For wavelength-dependent SERS experiment, 4-ATP and 4-MBA molecules were used in addition to R6G and CV. For 4-ATP, the peak at 1590 and for 4-MBA the peak at 1596 assigned to C=C stretching mode of benzene ring was used to calculate the EF at millimolar concentration. Identification of b2 modes for all four analytes was undertaken in order to identify the dominance of a particular resonance in the SERS enhancement with the quantum probes.

**Cell culture, cell fixation, and cell analysis**. Two types of human cancer cells—breast adenocarcinoma (MDAMB 231) and human epithelioid cervix carcinoma (HeLa) along with one primary mouse cell line fibroblast (NIH3T3)—were used in this experiment. All cell lines were purchased from American Type Culture Collection (ATCC). The cells were seeded and cultured in Dulbecco's modified Eagle's medium (DMEM)/F-12 medium supplemented with 10% fetal bovine serum and 1% penicillin/streptomycin. Cells were incubated at 5% CO₂ and 37 °C. When the cells reached 85% confluency, cells were trypsonized, centrifuged, and resuspended in DMEM/F-12 media. Samples with synthesized quantum probe and nano-dendrite platform were placed in petri plates and 3 mL of media was poured in petri plates. Resuspended cells (3 million cells per petri plate) were then directly seeded on the nanostructures. The petri plates containing media, samples, and cells were then incubated at 5% CO₂ and 37 °C for 6, 12, 24, and 48 h. Samples achieving the time limit were then taken out of incubator, and the media was removed from the petri plates. The samples were then washed twice with phosphate-buffered saline (PBS) and incubated at room temperature with 4% paraformaldehyde for 20 min. Paraformaldehyde was then removed and zinc plates were then incubated at room temperature with 5% glycine. After removal of glycine, the samples were washed twice with PBS and allowed to air dry. The protocol of air drying of cells at room temperature was followed because literature suggests that air-dried cells result in better enhancement and more sensitive spectra as compared to experimentation with hydrated cells[68]. The expectation from this study was of a diagnostic application with "cancerous" or "non-cancerous" response up to a single-cell-level detection. So air drying of the samples did not have adverse effect on the SERS spectra.

Cells were also fixed for SEM imaging at Peter Gilligan Centre of Research and Learning, Toronto, ON. SEM imaging of the cells was done using Hitachi SU1510 Scanning Electron Microscope. Fluorescence imaging of same samples was done using Nikon fluorescence microscope.

Raman spectra of the air-dried samples with fixed cells were obtained for at 6, 12, 24, and 48 h fixation intervals. Raman spectroscopy was done using Renishaw inVia Confocal Raman Spectrometer Leica DMI 6000 epifluorescence microscope. Three spectra for each ablation array were obtained and averaged out to obtain final spectra. Acquisition time per spectrum was 10 s. All the spectra were plotted using the "Spectragraph spectroscopy software". Spectra of different cell lines as well as at different time intervals were then analyzed using biomolecular Raman assignments from the literature in order to identify the effect of specific resonances for cellular Raman enhancements. Please refer to "Supplementary Table 5" for Raman peak assignments of the cells. Spectral bands of various biomolecules were analyzed to distinguish between cancer cells and non-cancer cells fixed at 24-h time interval using signature Raman assignments for HeLa as well as MDAMB231 cancer cell lines widely accepted in the literature[57,58,69,70] and comparing with a non-cancer mammalian cell line of NIH3T3. In order to distinguish between cancer and non-cancer cell lines, ratio of Raman intensities at 1445 and 1655 cm$^{-1}$ were compared. This ratio has been used for classification between normal and malignant cells in the brain, breast, colon, and cervix[59]. The sensitivities of the band for CH₂ scissoring for proteins at 1445 cm$^{-1}$ and the band for C=O stretching of collagen and elastin for lipids at 1655 cm$^{-1}$ for histological abnormalities has been widely utilized as a test for malignancy in the literature. The decision ratio ($I_{1445}/I_{1655}$) was able to separate cancer cells from normal cells. Scatter plots for all three cell lines were plotted for this classification. Although Raman spectroscopy can provide fingerprint spectra of various cancer cell lines, the peaks generated due to proteins, lipids, nucleic acids, and polysaccharides are very complex and can overlap over a broad band. There is a possibility of introducing inaccuracies in the interpretation of data when the peaks are analyzed qualitatively. In order to reduce the errors due to guessing of biochemical components, qualitative as well as quantitative analysis of Raman bands was performed[71].

**Fig. 9** Ability of ZnO-based non-plasmonic quantum probe for cancerous and non-cancerous diagnosis. **a** Variation in SERS spectra of cancer and non-cancer cells due to changes in metabolic pathways and changed regulation of surface receptors. **b** Combination of PCA, DA, and ratio analysis demonstrating ability of disease diagnosis. Multivariate analysis of principal component analysis and discriminant analysis for cell discrimination showed clear distinction between cancer cells and mammalian cells. Scatter plot of Raman intensity ratios at 1445–1655 cm$^{-1}$ ($I_{1445}/I_{1655}$) separating cancer cells from non-cancer cells. Mean ratio of 1.00 for HeLa and 0.82 for MDAMB separating clearly from non-cancer cell line of NIH3T3 2.34. PCA displays 3D plot of principal components F2 on "x" axis F1 on "y" axis and F3 on "z" axis, whereas DA displays discriminant components F1 on "x" axis and F2 on "y" axis. Ratio analysis displays cell type on "x" axis and ratio ($I_{1445}/I_{1655}$) on "y" axis. **c** Cell TEM reveal the presence of many quantum probes in the nucleus, cytoplasm, and cell membrane providing label-free, simultaneous, multiplexed detection of DNA, RNA, proteins, and lipids in vitro. Scale bar = 2 µm for NIH3T3 and MDAMB 231 and scale bar = 200 nm for close-ups (i, ii, iii, iv). **d** SEM and fluorescence images of HeLa, MDAMB231, and NIH3T3 cells. Scale bar = 20 µm. The "x" axis of the spectra display Raman Shift (cm$^{-1}$) and "y" axis displays arbitrary units. Mean ratio bar chart includes error bar in S.D. n = 20

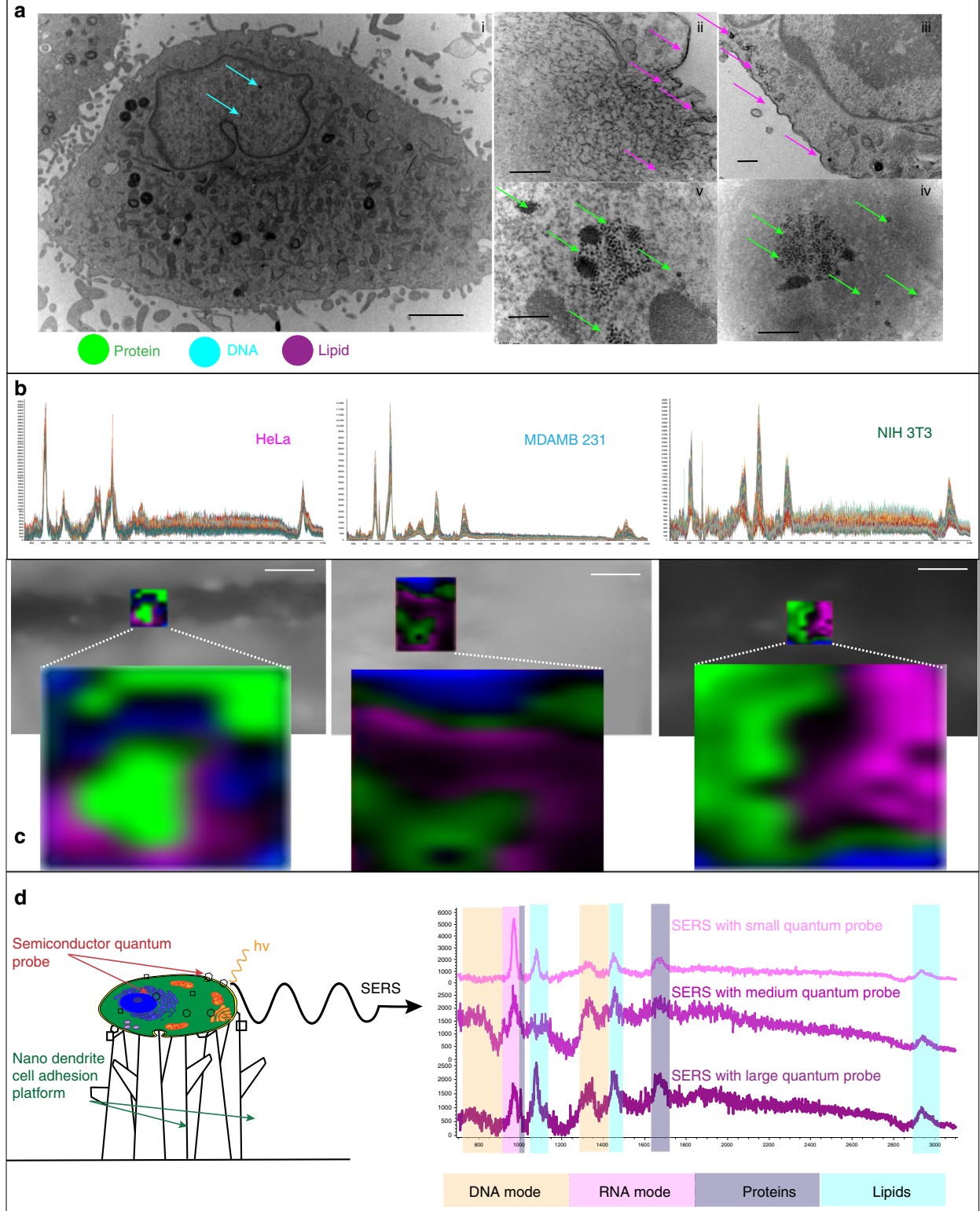

**Fig. 10** In vitro SERS detection up to a single-cell level. **a** Simultaneous signal enhancement from multiple biomolecules was due to the cellular uptake and internalization of the quantum probes. Scale bar 2 μm for (i) and 500 nm for (ii–v). **b** SERS spectra from single cells reveal that the entrapment of the quantum probes in the lysosomes does not have any adverse effect on signal enhancement. **c** Single-cell Raman mapping indicates ability of the quantum probe up to single-cell-level detection. Scale bar = 20 μm. **d** Different size of the probe plays an important role in the SERS showing ability to design variety of probes based on cellular affinity. The "x" axis of the spectra display Raman Shift (cm$^{-1}$) and "y" axis displays arbitrary units

Combination of multivariate analysis of PCA and DA was employed with "XLstat." PCA analysis is a non-parametric approach that does not need any explicit background model[72]. Dimensionality of the data can be reduced with PCA, as mathematical decomposition of Raman spectra into PCs is done. PCA is effectively used in Raman studies for discrimination of cells[71,73]. Cells were identified using an inverted microscope and spectra were gathered from 200 to 3275 cm$^{-1}$ wavenumbers. The dimensionality of the observations recorded for each spectra were reduced by getting three PCs (F1, F2, and F3), so three PCs per spectra were obtained. A 3D cluster graph of F1 vs F2 vs F3 was plotted for all 60 samples. Similarly, for DA, a cluster graph of F1 vs F2 (discriminant functions F1 and F2) was plotted for all 60 samples. Since cells contain >10,000–15,000 mRNAs, it is possible to understand approximately 150 tumor-suppressing genes from the study of RNA[74,75]. The intensity values of RNA wavenumber at 915 cm$^{-1}$ were used as an independent variable for DA for discrimination of cell lines.

Single cells on the nanostructures were identified with the help of Leica DMI 6000 epifluorescence microscope. Isolated cells were visible in the Raman mapping experiment and the bright field images of these cells were obtained. Calibration of the software was undertaken using silicon. After confirming the software calibration, Raman spectra of four analytes CV, R6G, 4ATP, and 4 MBA were obtained in order to ensure correctness of the data. Once the accuracy for the Raman peaks was confirmed, Raman mapping was carried out at 3 × 3 steps for the entire selected area by 2D maps of intensity at 752, 915, 1030, 1334, 1655, 2854, and 2940 cm$^{-1}$ wavenumbers. These wavenumbers representing various biomolecular signatures were obtained from the literature (please refer "Supplementary Table 5"). The individual maps for each wavenumber were then merged to get the final map of the single cell using the wire 3.3 software. The single-cell maps were built with the help of in-built functionality of the wire 3.3 software.

In order to understand the mechanism of the cellular uptake of the quantum probe and its effect on in vitro SERS enhancement, cell TEM were obtained. Cells were seeded and cultured as mentioned above in a petri plate for 24 h. Array of 1 × 1 cm$^2$ quantum probes was synthesized and sonicated for 5 min in 20 mL PBS. A total of 500 µL of the solution was poured on 24 h grown cells and allowed to incubate for 6, 12, and 24 h. The cells were trypsonized, resuspended in media, and centrifuged at 200 × $g$ for 8 min. The pellet formed was then stored under SEM fixative medium at 4 °C and sent for TEM sample preparation. The samples were embedded in epoxy resin; thin sections were cut and collected on copper grid. In order to image the quantum probe internalized by the cells, only unstained samples were used for imaging. The samples were then imaged with FEI Tecnai 20 TEM. In order to study the energy-dependent endocytosis of the quantum probe, 500 µL of PBS solution with quantum probe was poured on the cells and the cells were incubated at 4 °C for 4 h to block endocytosis[58]. The cells were then washed twice with PBS to remove any excess quantum probe and new media was poured in petri plates. The samples were then incubated for 24 h at 37 °C. The cells were trypsonized, resuspended in media, and centrifuged at 200 × $g$ for 8 min. The pellet formed was then stored under SEM fixative medium at 4 °C and sent for TEM sample preparation. The samples were embedded in epoxy resin; thin sections were cut and collected on copper grid. In order to image the quantum probe internalized by the cells, only unstained samples were used for imaging. The samples were then imaged with FEI Tecnai 20 TEM.

**Data availability**. The dataset generated during and/or analyzed during the current study are available from the corresponding author on reasonable request.

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

## Acknowledgements

This research was funded by NSERC Discovery Grants 132950 and 134361.

## Author contributions

All the authors worked together in designing the project. R.H. performed the experiments, wrote the manuscript, drew the schematics, and created figures. K.V. and B.T. assisted in evaluating the results, discussion, and editing the manuscript.

## Additional information

**Competing interests:** The authors declare no competing interests.

