## [Peer Review File · Nature Communications]

Reviewers' Comments:

Reviewer #1:

Remarks to the Author:

In this work, the authors studied a label free, biocompatible, ZnO based 3D semiconductor quantum probe as a pathway for in-vitro diagnosis of cancer. An ultrahigh SERS enhancement factor $\sim 10^6$ was obtained by using these quantum probe. The results of this work are interesting. This work could be considered for publishing after a major revision. Several questions are listed as following:

1. The author employed Raman spectra to study the crystal structure of ZnO and discovered red shift of the peak. The author indicated this phenomenon can be ascribed to the oxygen defects. In SERS experiments, an ultrahigh enhancement factor was obtained under the excitation of 785 nm. The energy of this excitation is much lower than the band gap energy of ZnO, thus I think the CT resonance might be originated from the contribution of interfacial defect states. The author may check it from the following two papers (S. Cong et al, Nat. Commun. 2015, 6, 8800.; J. Lin et al, Adv. Mater. 2017, 29, 1604797.).

2. In the manuscript, authors used R6G and CV dye molecules as probe molecules. My suggestion is the author can use sulfhydryl molecules, such as 4-mercaptobenzoic acid (4-MBA) and 4-Aminothiophenol (4-ATP), to further check the charge transfer mechanism. If the enhancement derived from charge transfer, the ν_{22} vibration mode of these dye molecules and sulfhydryl molecules would be selectively enhancement, the author may check this phenomenon and refer to the following two papers (X. T. Wang, et al, J. Am. Chem. Soc. 2011, 133, 16518.; Wang, X. et al. Angew. Chem. Int. Ed. 2017, 56, 9851–9855).

3. Author indicated that the enhancement mechanism in this work can be ascribed to the charge transfer and SPR effect. To investigate which effect play the dominant role, the author should perform the wavelength-dependent SERS (two or three laser wavelengths) measurement to identify the contribution of physical enhancement and chemical enhancement.

4. In the introductory section, authors should supplement the background and related literature on semiconductor-based SERS effect to make it easier for readers to understand this work.

5. The selected-area electron diffraction (SEAD) of the ZnO sample should be provided.

6. The format of Manuscript should be further improved, the current version does not meet the NC's style.

Reviewer #2:

Remarks to the Author:

This manuscript reports in vitro SERS studies on cells using non plasmonic semiconductor quantum SERS probes.

The authors highlight the "quantum size" of the probes but the reported probes are on the same size range as plasmonic gold and silver nanostructures which are very successfully applied in cellular SERS studies. Unfortunately, the authors cite only very few papers in this field. In general, I have problems to identify in this paper something that goes beyond what has been reported on cellular SERS studies in recent years.

This lack in the degree of novelty aside, I have a number of serious problems with the paper regarding its scientific quality. Here are only a few of my concerns.

The text in many places is misleading, such as in the discussion of the enhancement mechanisms operative in SERS on semiconductors. The discussion of potential contributions on various "resonances" appears very superficial and non-professional. Fig.4 and particularly also Fig. 7 are not supported by data and are based on pure speculation.

Regarding the calculation of the enhancement factors, there is no way to infer the claimed numbers from reported data, nor is any estimate or discussion of error bars provided.

I have also serious concerns about the assignment of the bands in the spectra measured from cell constituents. This makes the claim of "In-Vitro Cancer Detection" more than questionable.

Overall, the manuscript is hard to read. The Figs. are in the style of a graphical abstract but they are absolutely unacceptable when used as Figs in a scientific publication. For example, Raman spectra do not show x-axis and y-axis and therefore, it is impossible for the reader to draw any conclusion from the shown spectra.

I strongly believe that this work does not fulfill the quality requirements of Nature Photonics or any other good journal and therefore, I cannot recommend publication.

Reviewer #3:

Remarks to the Author:

This manuscript deals with the use of a non-plasmonic probe for Surface Enhanced Raman Scattering as a method for the detection of cancer making. The problem with the more used plasmonic metal nanoparticles like silver and gold is that they can suffer from selectivity, coagulation and biocompatibility and for this reason SERS diagnostic of cancer is in principle limited to the detection of biomolecules. Thus, in this work it is stated that it is necessary to develop a different approach so that SERS can be performed in-vitro. SERS using semiconductors is evolving in the last few years, and in the particular case of ZnO, this material has interesting properties, biocompatibility among them, which makes it a good candidate for being used in SERS. In this work, authors use ZnO and reduce its size reduced to quantum scale so that an enhancement of six orders of magnitude is obtained at nanoMolar concentration for model analytes. Then, the SERS activity for in-vitro sensing was studied for Breast Adenocarcinoma (MDAMB 231) and Human Epithelioid Cervix Carcinoma (HeLa) cells. The quantum probe was decorated on a nanodendrite platform, which was functionalized for cell adherence. This label free platform was engineered by three dimensional assemblies of nano dendrites with various morphologies, mimicking the extracellular matrix for self-targeting, cell adherence and proliferation. Authors are able to analyze in-vitro SERS up to single cell level. These results are novel and original and the manuscript is well written and the research shown here is of potential interest for the scientific community, in particular in the field of sensing and more specifically for bio- and medical sensing.

Results are clearly shown and detailed, however, regarding the experimental part some information is missing and should be included at least in the supplementary information. In particular:

1. Although it is clear that the important point and the novelty of the manuscript is the use of a non-plasmonic probe for SERS, some more information should be given on the preparation of the probe. It is mentioned that "manipulation of size and structure is possible by manipulation of peak power and condensation time interval". This is reasonable and known but authors should give (at least) information on irradiation time and repetition rate. Also, how is the irradiation performed? Is the laser impinging on the sample vertically and normal to the surface? Is the sample translated while irradiation? How? What is the range of scanning speed used? Why are larger crystals formed for higher scanning speed? How many pulses reach the same position of the sample? Could be that successive pulses are modifying the crystals already formed? Authors repeat throughout the manuscript that properties of the nanostructures can be tuned but it remains as a kind of mystery since they do not give details.

2. Some technical information on the Raman measurements are also missing. What is the magnification of the objective used? What is the size of the area inspected? The spectral resolution? The spatial resolution? What is the power of the laser so that no side effect is induced (no heating or damage)?
3. Although not the purpose of the work authors should explain the changes observed in the SERS spectra as a function of the concentration used, since they are showing these results. Why do the bands observed differ depending on the concentration? Is there any photodegradation? Photobleaching? Reactivity? Why is the enhancement factor increasing when the concentration is lower? Again: reactivity?
4. The nanostructures providing maximum enhancement for the model analytes crystal violet R6G do not provide maximum enhancement for in-vitro SERS and it is also mentioned that the affinity of different cell lines for in-vitro enhancement varies based on the size of quantum probe. Is there any way to predict or to know what is the optimal size of the nanostructures depending on the target we want to analyse?
5. The enhancement in the case of DNA, RNA, proteins and lipids is much lower than in the case of the model analytes. Could be this further optimized?
6. How universal is this method in general for other cancer cells? Could authors comment on this?
7. Although the figures are very nice and easy to understand, some additional information should be included. In particular, in the case of the Raman spectra (which is the relevant result to be shown) wavenumber scale should be displayed. Additionally, it should be mentioned that the intensity scale is not real (or not the same for every condition shown).
8. What is labelled as discussion is rather a conclusions section.
9. There are also some mistakes (for example in the caption of the figure 1 in the supplementary information it is mentioned that the wavelength is 780 but It should be 785 nm) and some typos.

Point by point response to the reviewers' comments

With reference to your letter regarding revision of the manuscript entitled "Non Plasmonic Semiconductor Quantum SERS Probe – Pathway For In-Vitro Detection" by Rupa Haldavnekar, Dr. Bo Tan and I, we will like to thank you and the reviewers for the feedback on the manuscript. We have addressed the reviewers' comments by undertaking extensive new experimentation. The new data is incorporated in the manuscript. Following new experiments, data gathering and analysis were undertaken for submission of this major revision of the original manuscript.

1. In order to validate the presence of the charge transfer mechanism in the SERS enhancement, additional two analyte molecules (4-MBA & 4-ATP) which were able to chemisorb on the substrate were used along-with the two analyte molecules (Crystal Violet & R6G) which were able to physically adsorbed on the substrate. New experiments with four molecules were carried out. (reviewer #1)
2. Previous work was based on only one SERS excitation wavelength (785 nm) which is critical for biological SERS experimentation. Two new excitation wavelengths (532 nm & 638 nm) were used in addition to the one excitation wavelength previously used to identify dominance of a certain resonance in the SERS response of the quantum probe (Reviewer #1, 2) and substantiating the finding by use of four analyte molecules.
3. Three new time dependent experiments for in-depth study of cellular uptake of the Quantum Probe and its analysis for better understanding of in-vitro SERS response was carried out (Reviewer 2 & Reviewer #3). New samples were prepared, new analysis with two cell lines was done to prove contribution of resonances for the in-vitro SERS.
4. Two extra material characterization experiments and analysis was undertaken to confirm the presence of the oxygen vacancies in the quantum probe which is responsible for the substantial SERS enhancement (Reviewer #1). The extra experiments and analysis was done with X-Ray Photoelectron Spectroscopy and Electron Paramagnetic resonance Spectroscopy on the quantum probe.
5. The manuscript was formatted to fit into NCs format.

Point-by-point response to the reviewers remarks are outlined below.

Reviewer #1 –

Comment 1: *"The author employed Raman spectra to study the crystal structure of ZnO and discovered red shift of the peak. The author indicated this phenomenon can be ascribed to the oxygen defects. In SERS experiments, an ultrahigh enhancement factor was obtained under the excitation of 785 nm. The energy of this excitation is much lower than the band gap energy of ZnO, thus I think the CT resonance might be originated from the contribution of interfacial defect states. The author may check it from the following two papers (S. Cong et al, Nat. Commun. 2015, 6, 8800; J. Lin et al, Adv. Mater. 2017, 29, 1604797.)."*

Response : The authors thank the reviewer for this valuable comment.

1. Since there was indication of oxygen vacancies from Raman spectra, in order to confirm these defects, the authors further conducted X-Ray Photoelectron Spectroscopy (XPS) and Electron Paramagnetic Resonance Spectroscopy (EPR) on the samples.
2. On analysis of the oxygen curve in XPS, the authors have confirmed the presence of oxygen vacancies in the quantum probe. This was further confirmed with the g factor of 1.96 from EPR.
3. An additional figure of curve fitting of Zn2P curve and deconvolution of Oxygen curve from the XPS spectra to substantiate the oxygen vacancies, EPR results and deconvolution of Raman Spectra is added in the supplementary information (Supplementary Fig 4).
4. Two new references was added - S. Cong et al., "Noble metal-comparable SERS enhancement from semiconducting metal oxides by making oxygen vacancies," *Nat. Commun.*, vol. 6, p. 7800, 2015 and L. Jin, G. She, X. Wang, L. Mu, and W. Shi, "Enhancing the SERS performance of semiconductor nanostructures through a facile surface engineering strategy," *Appl. Surf. Sci.*, vol. 320, pp. 591–595, 2014.

The presence of oxygen vacancies was reinforced in the manuscript as follows

1. Introduction – Page 6, Line 9 “We observed a new phenomenon of exponential increased in SERS from ZnO based semiconductor quantum material with surface defects like oxygen vacancies & stacking faults along-with display of SERS at multiple excitation wavelengths for the first time to the best of our knowledge.”
2. Results & Discussion - ‘Formation, Characterization and Classification of label free Quantum SERS Probe’ (Page 9 – Line 10) “In order to investigate presence of Oxygen vacancies, X-Ray Photoelectron Spectroscopy (XPS) and Electron Paramagnetic Resonance Spectroscopy (EPR) were performed. The Zn2P curve from XPS showed symmetric peaks at 1022.27eV and 1045.36 eV. This confirmed formation of hexagonal wurtzite structures. The gap in the binding energies (23.09eV) fitted with the values from the literature. (Refer supplementary fig 4). The oxygen curve was fitted into three peaks at 531.05eV; 531.81eV and 532.85eV for the quantum probe where as for the native only two peaks were fitted at 530.26eV and 532.02eV. The peaks at lower binding energy were assigned to the presence of O²⁻ ions in Zn-O bonding of the wurtzite structure, the peak at 531.81eV for the nanostructures and 532.02eV for native were assigned to C=O bonding due to the surface adsorption of the atmospheric carbon and the peak at 532.85eV for the nanostructures was assigned to O⁻ and O²⁻ ion deficient regions[1], [2]. EPR provided “g factor” value of 1.96 indicating presence of oxygen vacancies [3].”
3. Results & Discussion – SERS Based Molecular Detection, Page 13, line 2 “This charge transfer resonance seems to exponentially increased due to the contribution of interfacial defects like oxygen vacancies [4]”
4. Results & Discussion – SERS Based Molecular Detection, Page 14, line 3 “Charge transfer being the dominant mechanism for SERS with the ZnO based quantum probe, which seems to have exponentially increased due to the presence of oxygen vacancies, surface defects, stacking faults affiliating interfacial charge transfer.”
5. Supplementary Information- Supplementary figure 4 showing XPS, EPR & Raman spectroscopy results confirming presence of oxygen vacancies was added on Page 7.

Comment 2: “*In the manuscript, authors used R6G and CV dye molecules as probe molecules. My suggestion is the author can use sulfhydryl molecules, such as 4-mercaptobenzoic acid (4-MBA) and 4-Aminothiophenol (4-ATP), to further check the charge transfer mechanism. If the enhancement derived from charge transfer, the b2 vibration mode of these dye molecules and sulfhydryl molecules would be selectively enhancement, the author may check this phenomenon and refer to the following two papers (X. T. Wang, et al, J. Am. Chem. Soc. 2011, 133, 16518; Wang, X. et al. Angew. Chem. Int. Ed. 2017, 56, 9851–9855).*”

Response: The authors thank the reviewer for this valuable suggestion.

1. The authors have undertaken a new experiment with 4-MBA and 4-ATP in order to further validate presence of the charge transfer mechanism as suggested by the reviewer.
2. The experiment was conducted at Mili-Mole concentration with all four analyte molecules (4-ATP, 4-MBA, CV & R6G).
3. Figure 4 has been revised and spectra for SERS from all 4 molecules are incorporated in the figure.
4. Supplementary table S.T4 for b2 band assignments was added.
5. Two new references X. Wang *et al.*, “Remarkable SERS activity observed from amorphous ZnO nanocages,” *Angew. Chem. Int. Ed.*, 2017 & Wang Xiaotian, Shi Wensheng, She Guanwei, and Mu Lixuan, “Using Si and Ge Nanostructures as Substrates for Surface-Enhanced Raman Scattering Based on Photo induced Charge Transfer Mechanism,” *J. Am. Chem. Soc.*, no. 133, pp. 16518–16523, 2011 were added in the main manuscript.
6. Additional reference - L. G. Quagliano, “Observation of Molecules Adsorbed on III-V Semiconductor Quantum Dots by Surface-Enhanced Raman Scattering,” *J. Am. Chem. Soc.*, vol. 126, no. 23, pp. 7393–7398, Jun. 2004 was added in the supplementary information.
7. Methods section was revised to incorporate the new experimentation

The discussion on presence of charge transfer mechanism was reinforced as follows:

1. Results & Discussion – SERS Based Molecular Detection, Page 12 line 3 - For better understanding of charge transfer mechanism, two thiol based molecules (4-MBA & 4-ATP) bonding covalently by chemisorption process were used for comparison of the SERS mechanism at Mili-Mole concentration with excitation using 785 nm wavelength. All four molecules showed not-totally symmetric b2 modes (Please refer to the supplementary table S.T4 for b2 band assignments of all the molecules) confirming charge transfer mechanism due to Herzberg-Teller coupling [5], [6]. As expected, EF of the chemisorbed molecules (4ATP 3.8×10^6 & 4MBA 6.9×10^6) was much higher as compared to the physically adsorbed molecules (R6G 1.71×10^5 & CV 8.26×10^5) at Mili-Molar concentration.
2. Results & Discussion – SERS Based Molecular Detection, Page 12 line 20 - This selective enhancement of b2 modes demonstrated contribution from charge transfer mechanism due to Herzberg-Teller coupling. [7]. Refer Fig 3, pulsed laser deposition produces very high carrier mobilities. According to the Hamiltonian of semiconductors, very high mobilities of charge carriers because of the 3-D arrangements of ZnO quantum crystals may result into the combination of exciton and plasmon resonances [8]. The exciton resonance results in SERS due to charge transfer by Herzberg-Teller coupling either from molecular HOMO to the conduction band edge or from valance band edge to molecular LUMO. This charge transfer resonance seems to exponentially increased due to the contribution of interfacial defects like oxygen vacancies [4].
3. The methods were revised as follows - Additional experiment with two chemisorbed analyte molecules (4-ATP, 4-MBA) at Mili-Molar concentration and 785 nm excitation wavelength was undertaken in order to get better understanding of the charge transfer mechanism.
4. Supplementary Information – Page 9 – Table S.T 4 for the not-totally symmetric b2 modes of all the molecules

Comment 3: “ Author indicated that the enhancement mechanism in this work can be ascribed to the charge transfer and SPR effect. To investigate which effect play the dominant role, the author should perform the wavelength-dependent SERS (two or three laser wavelengths) measurement to identify the contribution of physical enhancement and chemical enhancement.”

Response: The authors thank the reviewer for this important comment.

1. The authors undertook a new experiment with three different excitation wavelengths (532 nm, 638 nm, and 785 nm) as per the reviewer’s suggestion.
2. Four analyte molecules (4-ATP, 4-MBA, CV & R6G) were used at Mili-Mole concentration.
3. Discussion was revised to incorporate the analysis for study of dominance of a certain resonance for SERS mechanism.
4. Figure 3 was revised and SERS spectra of all molecules at the three different wavelengths were incorporate (Page 15).
5. Methods section was revised
6. Supplementary table S.T3 was incorporated with wavelength based enhancement factors of all analyte molecules.
7. Supplementary figure 5 was added with graphical representation of enhancement factor (page 9 – Supplementary information).

The dominance of a certain resonance in SERS mechanism was reinforced as follows:

1. Results and Discussion – SERS based molecular detection – Page 12,13 & 14 “Since all analyte molecules showed high enhancement at the excitation wavelength of 785nm which was not in the close vicinity of the absorption maxima of the molecules, dominance of molecular resonance was ruled out. In order to further investigate dominant mechanism in the SERS, all four analyte molecules at Mili-Molar concentration were excited at three different wavelengths 532nm, 638nm & 785nm. We successfully demonstrated SERS at all three wavelengths (Refer supplementary table S.T3 for enhancement factors) showing multi-wavelength excitation ability of the quantum probe. Refer Figure 3, all analyte molecules showed maximum enhancement with 785 nm excitation wavelength followed by 638 nm and showed minimum enhancement at the excitation wavelength of 532 nm. The not-totally symmetric b2 modes of all the molecules were clearly visible at all wavelengths and showed maximum intensity at 785 nm excitation. This selective enhancement of b2 modes demonstrated contribution from charge transfer mechanism due to

Herzberg-Teller coupling. [7]. Refer Fig 3, pulsed laser deposition produces very high carrier mobilities. According to the Hamiltonian of semiconductors, very high mobilities of charge carriers because of the 3-D arrangements of ZnO quantum crystals may result into the combination of exciton and plasmon resonances [8]. The exciton resonance results in SERS due to charge transfer by Herzberg-Teller coupling either from molecular HOMO to the conduction band edge or from valance band edge to molecular LUMO. This charge transfer resonance seems to exponentially increased due to the contribution of interfacial defects like oxygen vacancies [4]. Photo-induced charge transfer also contributed to SERS. The ZnO quantum probes have polyhedral morphology and appearance of defects everywhere in the crystal lattice. We experienced a phenomenon similar to the tip enhanced Raman scattering due to the quantum scale of the corners and defects in the edges [9]. The contribution of increased surface charges due to the geometry of the corners of polyhedrons can lead to localized and intense electromagnetic fields at the corner apexes. The confinement and intensity of the field can further increase exponentially in the proximity of quantum scale apexes of many more quantum probes due to the gap effect [10]. Refractive index of ZnO is estimated ~2, which can provide moderate to high refractive index contrast with air. Since, the quantum crystals are arranged in the form of 3-D networks, the cumulative contrast gets increased exponentially which can result into recovery and trapping of most incoming light minimizing Rayleigh scattering & substrate interaction [11]. This kind of effective trapping of photons may have boosted Raman scattering. The total internal reflection due to the evanescent fields allowed probing of the surface adsorbates with very high sensitivity providing photo-induced charge transfer.

The plasmon resonance can occur due to Coulomb interaction when electron density is sufficiently high resulting in observation of collective behaviors of electrons [7]. Pulsed laser deposition results in very high thermal effect which gives rise to the increased electron density in conduction band. This in addition to electron density in valance bands which is a dominant factor in semiconductors generates plasmon [8]. In case of ZnO, the electron density is estimated at $17\text{eV}[12] \sim 4 \times 10^{19}$ to $8 \times 10^{19} \text{ cm}^{-3}$ carriers, which may able to display a large oscillation strength to generate surface plasmon resonance at mid-infrared region [13], [14] which seems to be contributing at the near IR excitation wavelength of 785 nm. This could be the reason for maximum intensity at 785 nm excitation. ”

2. Results & Discussion – SERS based molecular detection – Page 14, line 1 “We theorize that the resultant SERS obtained with the Molecule-Quantum Probe system was due to the co-existence of Charge transfer resonance due to Herzberg-Teller vibronic coupling and Photo induced charge transfer; Surface plasmon resonance and molecular resonance – Charge transfer being the dominant mechanism for SERS with the ZnO based quantum probe, which seems to have exponentially increased due to the presence of oxygen vacancies, surface defects, stacking faults affiliating interfacial charge transfer.”

Comment 4: “In the introductory section, authors should supplement the background and related literature on semiconductor-based SERS effect to make it easier for readers to understand this work.”

Response: The authors thank the reviewer for this comment. Additional paragraph (Page 4, line 19) was added in order to provide background in semiconductor based SERS as follows – “Use of semiconductors for manufacturing of sensors is imperative in today’s ever-evolving microelectronic-integrated circuit industry. So far there has been limited amount of research in the field of semiconductor based SERS using TiO_2 , SiO , NiO & GaP due to relatively weak enhancement effects [15],[16]. The theory based on semi-conductor enhanced Raman scattering is still evolving [17],[7]. According to Lombardi & Birke, it is possible to get SERS due to a combined molecule-semiconductor system. The enhancement obtained is because of the combined effect of various resonances existing in the molecule-semiconductor system. These resonances coexist and should not be considered separately. The resultant enhancement was predicted by Lombardi & Birke to be of multiplicative nature [7]. In the study based on semiconductor nanostructures, amplification of molecular polarization intensifying the mechanism of interfacial charge transfer for SERS has been reported [5]. Reports on vibrational coupling between surface defects like oxygen vacancies and molecular vibration enhancing SERS were presented Cong et al [4]. But, generally SERS obtained with semiconductor based nanostructures is typically quite low in the range of 10^{-10^2} so

there is a tendency of using semiconductor mediated enhanced Raman scattering where semiconductors are used for analyte adsorption and plasmonic metal nanoparticles for surface enhancement [18].”

Comment 5: “The selected-area electron diffraction (SEAD) of the ZnO sample should be provided.”

Response: The authors have revised Figure 1 and SAED is added in the inset of TEM images as requested by the reviewer.

Comment 6:” The format of Manuscript should be further improved, the current version does not meet the NC's style.”

Response: The authors thank the reviewer for this comment. The authors have revised the format of the manuscript to meet the NC's style.

Reviewer #2 –

Comment 1: “The authors highlight the “quantum size” of the probes but the reported probes are on the same size range as plasmonic gold and silver nanostructures which are very successfully applied in cellular SERS studies. Unfortunately, the authors cite only very few papers in this field. In general, I have problems to identify in this paper something that goes beyond what has been reported on cellular SERS studies in recent years.”

Response: The authors thank the reviewer for this comment and would like to state the following- It is well known and has been widely reported that SERS with plasmonic nanomaterials like gold & silver often suffer from coagulation[19], selectivity[20] and adverse biocompatibility[21] which limits the applicability of label-free plasmonic nanomaterials for in-vitro cellular detection of cancer. Plasmonic nanomaterials need to be distributed in such a way that highly localized Raman hot spots remain discrete, which is very challenging due to uncontrolled agglomeration of these materials[22]. Plasmonic nanomaterials, thus need surfactants to provide good SERS response by controlling the hot-spots. Plasmonic nanomaterials also need to be functionalized for specific targeting with SERS active tags[23]. This ligand based SERS sensing often provides a contaminated spectra. The SERS tags as well as the surfactants often interact with the cells and damage the integrity of cellular structure [21]. Further, on interaction plasmonic nanomaterials cells often undergo oxidative stress causing damage to DNA & proteins. The process of functionalization of the nanomaterial is often too complicated apart from being very expensive. Hence, current research with plasmonic materials is limited to stand-alone biomolecular sensing. There is no reported work on label-free plasmonic nanomaterials for in-vitro SERS detection of cancer to the best of authors’ knowledge.

For all these reasons, there is a need for a biocompatible, label-free, non-plasmonic substrate with substantial SERS response for in-vitro SERS detection of cancer. In order to develop an alternative to the plasmonic SERS, the authors decided to explore a new field of non-plasmonic, semiconductor based SERS detection. The main focus of this study was to introduce a label-free, biocompatible, non-plasmonic SERS based detection mechanism for in-vitro cancer diagnosis. Typically, semiconductor based nanomaterials show poor SERS response, so the authors pushed the boundaries to quantum scale and observed a new phenomenon of substantial SERS from a non-plasmonic material at quantum scale. The authors successfully achieved enhancement factor of $\sim 10^6$ at nano-Molar concentration. This is substantial for a non-plasmonic material. This type of semiconductor based label-free, SERS based in-vitro detection of cancer has never been reported before to the best of the authors’ knowledge.

The authors will like to point out the novelties and significance of this manuscript here

1. The authors have successfully introduced a biocompatible, label free, ZnO based 3D semiconductor quantum probe as a pathway for in-vitro detection of cancer for the first time to the best of authors knowledge.

2. This manuscript demonstrates ability to achieve substantial SERS comparable to plasmonic SERS from a semiconductor material. This enhancement is significant for a non-plasmonic material.
3. The authors have introduced the concept of three dimensional ZnO based semiconductor quantum probe in order to achieve this significant SERS enhancement.
4. In addition to this, the authors have also demonstrated ability of the ZnO based semiconductor quantum probe for multi-wavelength SERS excitation.
5. We have successfully reported functionalization of a label-free ZnO based semiconductor nano-dendrite platform for cell adhesion and proliferation which is critical for in-vitro detection.
6. This type of non-plasmonic, label-free, ZnO based semi-conductor quantum probe, decorated on a nano-dendrite platform displayed multi-wavelength SERS ability, significant SERS enhancement, single cell level detection level of cancer for the first time to the best of authors knowledge.

These novelties have been reinforced by revising the abstract, introduction and conclusion sections of the manuscript.

1. Abstract – Page 2 – “Therefore, this research is limited to stand-alone biomolecule sensing.”
2. Introduction –Page 4- Line 9 – “ In order for the highly localized Raman hot spots to remain discrete, plasmonic materials often need surfactants to provide good SERS response. This is very challenging due to uncontrolled agglomeration of these materials[22] . This type of materials also need to be functionalized for specific targeting with SERS active tags [8] which can result in a contaminated spectra along-with adversely affecting the integrity of cellular structure [9].”
3. Introduction- Page 6 – line 7– “In this study, we have introduced the next generation of label free in-vitro SERS diagnosis of cancer by reducing the size of a biocompatible ZnO based probe to quantum scale.” And “We observed a new phenomenon of exponential increased in SERS from ZnO based semiconductor quantum material with surface defects like oxygen vacancies & stacking faults along-with display of SERS at multiple excitation wavelengths for the first time to the best of our knowledge. Enhancement factor $\sim 10^6$ was achieved with limit of detection up to Nano Molar concentration.” And “The quantum probe was decorated on a nano dendrite platform. This label free platform was engineered by three dimensional assemblies of nano dendrites with various morphologies mimicking the extracellular matrix for self-targeting, cell adherence and proliferation. This ZnO semiconductor quantum probe based SERS detection for in-vitro diagnosis of cancer was achieved for the first time to the best of our knowledge.”
4. Conclusion – Page 28 – “By reducing the size to quantum scale and engineering three dimensional assemblies of quantum network, very high SERS was achieved from a non-plasmonic material. Multi-wavelength excitation was demonstrated with substantial enhancement factor.” And “This type of non-plasmonic, label-free, ZnO based semi-conductor quantum probe, decorated on a nano-dendrite platform displayed multi-wavelength SERS ability, significant SERS enhancement, single cell level detection for the first time to the best of our knowledge.”

Comment 2: “The text in many places is misleading, such as in the discussion of the enhancement mechanisms operative in SERS on semiconductors. The discussion of potential contributions on various “resonances” appears very superficial and non-professional. Fig.4 and particularly also Fig. 7 are not supported by data and are based on pure speculation.”

Response: The authors thank the reviewer for this comment.

1. The authors have added more details as a background read on semiconductor based SERS in order to make it easier to understand the significance of this manuscript.
2. The authors have undertaken a new experiment with 4-MBA and 4-ATP in order to further validate presence of the charge transfer mechanism. The experiment was conducted at miliMole concentration with all four molecules (4-ATP, 4-MBA, CV & R6G). Supplementary table S.T4 for b2 band assignments was added.
3. The authors undertook another new experiment with three different excitation wavelengths (532 nm, 638 nm, 785 nm) to study dominance of a certain resonance. Four analyte molecules (4-ATP, 4-MBA, CV & R6G) were used at miliMole concentration.

4. The authors have undertaken new experiments for time based study of cellular uptake of quantum probe to substantiate the theory for in-vitro SERS. New samples were prepared and cell TEM imaging was undertaken
5. Fig 4 & Fig 7 in the original manuscript were schematic representation for better understanding of concepts. These figures are now revised and substantiated with new data from the new experiments.
6. The authors have added four new references to back-up the theory behind the SERS due to quantum probe. The theory is well accepted in the literature.
 - a. S. Cong *et al.*, "Noble metal-comparable SERS enhancement from semiconducting metal oxides by making oxygen vacancies," *Nat. Commun.*, vol. 6, p. 7800, 2015
 - b. L. Jin, G. She, X. Wang, L. Mu, and W. Shi, "Enhancing the SERS performance of semiconductor nanostructures through a facile surface engineering strategy," *Appl. Surf. Sci.*, vol. 320, pp. 591–595, 2014.
 - c. X. Wang *et al.*, "Remarkable SERS activity observed from amorphous ZnO nanocages," *Angew. Chem. Int. Ed.*, 2017
 - d. Wang Xiaotian, Shi Wensheng, She Guanwei, and Mu Lixuan, "Using Si and Ge Nanostructures as Substrates for Surface-Enhanced Raman Scattering Based on Photoinduced Charge Transfer Mechanism," *J. Am. Chem. Soc.*, no. 133, pp. 16518–16523, 2011 were added in the main manuscript.

The discussion on the enhancement mechanism is revised with additional data as follows-

1. Paragraph on semiconductor based SERS was added in the introduction as follows – Page 4, line 19 – "Use of semiconductors for manufacturing of sensors is imperative in today's microelectronic-integrated circuit industry. So far there has been limited amount of research in the field of semiconductor based SERS using TiO₂, SiO, NiO & GaP due to relatively weak enhancement effects [15],[16]. The theory based on semi-conductor enhanced Raman scattering is still evolving [17],[7]. According to Lombardi & Birke, it is possible to get SERS from semiconductor materials due to a combined molecule-semiconductor system. The enhancement obtained is because of the unified effect of various resonances existing in the molecule-semiconductor system. These resonances coexist and should not be considered separately. The resultant enhancement was predicted by Lombardi & Birke to be of multiplicative nature [7]. Very recently, amplification of molecular polarization intensifying the mechanism of interfacial charge transfer from semiconductor nanostructure SERS has been reported [5]. Reports on vibrational coupling between surface defects like oxygen vacancies and molecular vibration enhancing SERS were presented Cong et al [4]. But, generally SERS obtained with semiconductor based nanostructures is typically quite low in the range of 10-10² so there is a tendency of using semiconductor mediated enhanced Raman scattering where semiconductors are used for analyte adsorption and plasmonic metal nanoparticles for surface enhancement [18]."
2. SERS Based Molecular Detection, Page 12-14 – "For better understanding of charge transfer mechanism, two thiol based molecules (4-MBA & 4-ATP) bonding covalently by chemisorption process were used for comparison of the SERS mechanism at millimole concentration with excitation using 785 nm wavelength. All four molecules showed not-totally symmetric b2 modes (Please refer to the supplementary table S.T4 for b2 band assignments of all the molecules) confirming charge transfer mechanism due to Herzberg-Teller coupling [5], [6]. As expected, EF of the chemisorbed molecules (4ATP 3.8X10⁶ & 4MBA 6.9 X10⁶) was much higher as compared to the physically adsorbed molecules (R6G 1.71X10⁵ & CV 8.26X10⁵) at millimolar concentration. Since all analyte molecules showed high enhancement at the excitation wavelength of 785nm which was not in the close vicinity of the absorption maxima of the molecules, dominance of molecular resonance was ruled out. In order to further investigate dominant mechanism in the SERS, all four analyte molecules at miliMolar concentration were excited at three different wavelengths 532nm, 638nm & 785nm. We successfully demonstrated SERS at all three wavelengths (Refer supplementary table S.T3 for enhancement factors) showing multi-wavelength excitation ability of the quantum probe. Refer Figure 3, all analyte molecules showed maximum enhancement with 785 nm excitation wavelength followed by 638 nm and showed minimum enhancement at the excitation wavelength of 532 nm. The not-totally symmetric b2

modes of all the molecules were clearly visible at all wavelengths and showed maximum intensity at 785 nm excitation. This selective enhancement of b2 modes demonstrated contribution from charge transfer mechanism due to Herzberg-Teller coupling. [7]. Refer Fig 3, pulsed laser deposition produces very high carrier mobilities. According to the Hamiltonian of semiconductors, very high mobilities of charge carriers because of the 3-D arrangements of ZnO quantum crystals may result into the combination of exciton and plasmon resonances [8]. The exciton resonance results in SERS due to charge transfer by Herzberg-Teller coupling either from molecular HOMO to the conduction band edge or from valance band edge to molecular LUMO. This charge transfer resonance seems to exponentially increased due to the contribution of interfacial defects like oxygen vacancies [4]. Photo-induced charge transfer also contributed to SERS. The ZnO quantum probes have polyhedral morphology and appearance of defects everywhere in the crystal lattice. We experienced a phenomenon similar to the tip enhanced Raman scattering due to the quantum scale of the corners and defects in the edges [9]. The contribution of increased surface charges due to the geometry of the corners of polyhedrons can lead to localized and intense electromagnetic fields at the corner apexes. The confinement and intensity of the field can further increase exponentially in the proximity of quantum scale apexes of many more quantum probes due to the gap effect [10]. Refractive index of ZnO is estimated ~ 2 , which can provide moderate to high refractive index contrast with air. Since, the quantum crystals are arranged in the form of 3-D networks, the cumulative contrast gets increased exponentially which can result into recovery and trapping of most incoming light minimizing Rayleigh scattering & substrate interaction [11]. This kind of effective trapping of photons may have boosted Raman scattering. The total internal reflection due to the evanescent fields allowed probing of the surface adsorbates with very high sensitivity providing photo-induced charge transfer. The plasmon resonance can occur due to Coulomb interaction when electron density is sufficiently high resulting in observation of collective behaviors of electrons [7]. Pulsed laser deposition results in very high thermal effect which gives rise to the increased electron density in conduction band. This in addition to electron density in valance bands which is a dominant factor in semiconductors, generates plasmon [8]. In case of ZnO, the electron density is estimated at $17\text{eV}[12] \sim 4 \times 10^{19}$ to $8 \times 10^{19} \text{ cm}^{-3}$ carriers, which may able to display a large oscillation strength to generate surface plasmon resonance at mid-infrared region [13], [14] which seems to be contributing at the near IR excitation wavelength of 785 nm. This could be the reason for maximum intensity at 785 nm excitation.”

3. *This analysis was further concluded by reinforcing the theory behind SERS observed by quantum probe – Page 14* “We theorize that the resultant SERS obtained with the Molecule-Quantum Probe system was due to the co-existence of Charge transfer resonance due to Herzberg-Teller vibronic coupling and Photo induced charge transfer; Surface plasmon resonance and molecular resonance – Charge transfer being the dominant mechanism for SERS with the ZnO based quantum probe, which seems to have exponentially increased due to the presence of oxygen vacancies, surface defects, stacking faults affiliating interfacial charge transfer.”
4. *Figure 3* has been revised and additional spectrum demonstrating multi-wavelength SERS due to the quantum probe have been incorporated. This figure is now supported by additional data proving the dominance of charge transfer resonance in the SERS mechanism of ZnO based semiconductor Quantum Probe.
5. *Table S.T.3 and Figure 5* in the supplementary information is added for supporting the discussion regarding multiple wavelength SERS excitation.
6. *Table S.T4* is added in the supplementary information for molecular b2 band assignments supporting the theory of charge transfer mechanism.
7. *Additional reference* L. G. Quagliano, “Observation of Molecules Adsorbed on III-V Semiconductor Quantum Dots by Surface-Enhanced Raman Scattering,” *J. Am. Chem. Soc.*, vol. 126, no. 23, pp. 7393–7398, Jun. 2004 is added in the supporting information supporting b2 band assignments.
8. In order to substantiate the analysis of contribution of resonances (charge transfer or SPR) for bio-molecular SERS within a cell, analysis and discussion based on the new experimentation was added on *Page 19, line 19* – “The peaks assigned to DNA and RNA were clearly visible in the spectra of 6 Hr samples. Refer Figure 6, cell TEM does not show any evidence of quantum

probes within the nucleus for 6 Hr sample. Thus, we theorize dominance of SPR at 6 Hrs with cells show beginning of adhesion to the substrate. There is evidence of few quantum probes in the nucleus at 24 Hrs, hence increased intensity at 24 hrs can be interpreted as SERS due to increased charge transfer mechanism in addition to SPR. At 48 hr we observed substantial increase in the signal intensity, which can be due to cellular uptake of a large amount of quantum probe along-with more number of probes penetrating the nucleus giving rise to more charge transfer mechanism and hence more intense signal. Another reason could be the growth and proliferation of cells in 48 hr, more cells providing higher intensity of the signal”

9. Also Page 25, line 4 – “It is evident that there are more number of quantum probes in the cell membrane and cytoplasm as compared to the nucleus. This was because the quantum probes needed more time to penetrate the nucleus as compared to the cytoplasm. Thus, at any time interval, number of probes inside the nucleus are far less than in the cell membrane and cytoplasm. This could be the reason for highest enhancement of signal of lipids, which are located in or near the cell membrane and lowest for the innermost part of a cell – nucleus.”

Comment 3 – “Regarding the calculation of the enhancement factors, there is no way to infer the claimed numbers from reported data, nor is any estimate or discussion of error bars provided.”

Response – The authors thank the reviewer for this valuable comment. The authors will like to state that the authors have used enhancement factor calculation mechanism adopted in this field.

1. The calculation of the enhancement factor is explained in detail in the supplementary information.
2. The enhancement factor was calculated by obtaining three spectra and taking average of the spectra. Thus, error bars cannot be provided.
3. Additional table S.T3 is added in the supplementary information with EF at multiple excitation wavelengths.

The enhancement factor calculation is reinforced as follows –

1. The technical details which were not present in the original manuscript were added in this revision in the supplementary information – Page 8, line 1” Excitation wavelength of 532nm, 638nm & 785 nm were used. Other parameters like HC plan APO lense with matched polarizer/ analyzer optics (magnification of 20X, N.A.0.70), inspected spot size was of 0.625 μm radius, focal length of 250 mm, solid state laser with excitation wavelength 532nm (5mW), 638nm (5mW) & 785nm (12.5mW), matched polarizer optics power of lasers and spectral resolution of 0.5 cm^{-1} in visible and 1 cm^{-1} in NUV and IR & spatial resolution: < 1 μm (lateral), <2 μm (depth) were used.”
2. The X and Y axis of all the spectra are made visible as requested by the reviewer.

Comment 4:” I have also serious concerns about the assignment of the bands in the spectra measured from cell constituents. This makes the claim of “In-Vitro Cancer Detection” more than questionable. “

Response: The authors will like to thank the reviewer for this comment.

1. The authors have followed standard procedures to analyze cellular spectra.
2. The band assignments are well researched and many research articles based on the band assignments are well accepted in the literature.
3. The author has provided following additional references for the Raman band assignments in the manuscript.
 - a. W. A. El-Said, T.-H. Kim, H. Kim, and J.-W. Choi, “Analysis of intracellular state based on controlled 3D nanostructures mediated surface enhanced Raman scattering,” *PloS One*, vol. 6, no. 2, p. e15836, 2011.
 - b. C. Onogi, M. Motoyama, and H. Hamaguchi, “High concentration trans form unsaturated lipids detected in a HeLa cell by Raman microspectroscopy,” *J. Raman Spectrosc.*, vol. 39, no. 5, pp. 555–556, 2008.
 - c. Z. Movasaghi, S. Rehman, and I. U. Rehman, “Raman spectroscopy of biological tissues,” *Appl. Spectrosc. Rev.*, vol. 42, no. 5, pp. 493–541, 2007.
 - d. A. C. S. Talari, Z. Movasaghi, S. Rehman, and I. U. Rehman, “Raman spectroscopy of biological tissues,” *Appl. Spectrosc. Rev.*, vol. 50, no. 1, pp. 46–111, 2015.
 - e. P. Dittrich and N. Jakubowski, *Current trends in single cell analysis*. Springer, 2014.

4. The authors have also *revised table S.T5* in the supplementary information, providing relevant Raman peak assignments for in-vitro SERS application from the literature.

Comment 5: “Raman spectra do not show x-axis and y-axis and therefore, it is impossible for the reader to draw any conclusion from the shown spectra”.

Response: The authors have made all the X Axis and Y Axis information visible in all the graphs in all the figures as requested by the reviewer.

Reviewer #3

Comment 1: “Although it is clear that the important point and the novelty of the manuscript is the use of a non-plasmonic probe for SERS, some more information should be given on the preparation of the probe. It is mentioned that “manipulation of size and structure is possible by manipulation of peak power and condensation time interval”. This is reasonable and known but authors should give (at least) information on irradiation time and repetition rate. Also, how is the irradiation performed? Is the laser impinging on the sample vertically and normal to the surface? Is the sample translated while irradiation? How? What is the range of scanning speed used? Why are larger crystals formed for higher scanning speed? How many pulses reach the same position of the sample? Could be that successive pulses are modifying the crystals already formed? Authors repeat throughout the manuscript that properties of the nanostructures can be tuned but it remains as a kind of mystery since they do not give details.”

Response: The authors thank the reviewer for this comment .

1. The authors have added more details on synthesis of the probe in the supplementary information.
2. The section on synthesis is supported by additional figure in supplementary (Supplementary figure 1).
3. Three new references were added in the supplementary information.
 - a. Powell Jeffery Alexander, Tan Bo, and Venkatakrisnan Krishnan, “Programmable SERS active substrates for chemical and biosensing applications using amorphous/ crystalline hybrid silicon nanomaterial,” *Sci. Rep.*, 2015.
 - b. A. G. Milekhin *et al.*, “Surface enhanced Raman scattering of light by ZnO nanostructures,” *J. Exp. Theor. Phys.*, vol. 113, no. 6, pp. 983–991, 2011.
 - c. S. J. Chen, Y. C. Liu, Y. M. Lu, J. Y. Zhang, D. Z. Shen, and X. W. Fan, “Photoluminescence and Raman behaviors of ZnO nanostructures with different morphologies,” *J. Cryst. Growth*, vol. 289, no. 1, pp. 55–58, 2006.

The supplementary information was revised as follows-

1. Supplementary information- Page 2 & 3 –“ Supplementary Method 1- Formation of the quantum probe : A pulsed Yb-doped laser was used to synthesize ZnO based semiconductor quantum probes. Some parameters were fixed for maximization of control over the quantum probe and nano dendrite platform. These parameters were fixed on the basis of ionization characteristics of ZnO. Laser wavelength (1030nm), polarized (circular) and laser power (16W) were fixed to get the optimal nanostructure . The nanostructure were created on 5 X 5mm array of lines with 2 μ m point spacing. Samples were in the form of 99% pure zinc plates, surface of which was placed in perpendicular with the propagation of laser beam. The laser processing was carried out by translating the sample plane parallel to sample surface and focus was achieved by translating the sample in perpendicular direction. Scanning speed was varied from 0.5 mm/s to 5 mm/s for controlling the peak power and repetition rate was varied from 4MHz to 25 MHz for achieving control over condensation time interval. Each ablation area was examined using SEM microscope to confirm the nanostructure formation. Please refer to Fig S1, we were able to synthesize ZnO nano dendrite platform decorated with quantum crystals by multi-photon ionization mechanism. The bottom layer is in the form of nano dendrites which are decorated by three dimensional assemblies of quantum probes.

The femtosecond pulsed laser ablation causes rapid fluctuation in the temperature of the plume [14]. Presence of several morphologies of randomly oriented structures like 3-D networks of quantum crystals, random assemblies of nano rods as well as nano wires was observed [24] indicating multiple formation mechanism [25]. Different scanning speeds and repetition rates were used for fabrication. Control on etch depth was achieved by changing scanning speed. By changing repetition rate, control on generation of peak power for each pulse can be achieved. Hence, the energy transmitted to the sample surface can be controlled. As the scanning speed increased, the thickness of the rods decreased and size of the quantum probes increased. As the repetition rate increased, the morphology of the rods changed from thick, blunt and long rods to sharp needlelike long rods to short and thin rods in addition to reduction in size of the crystal mesh. So by programming the synthesis parameters engineering of functionalized the nano dendrite platform for cell adhesion and quantum probe for in-vitro SERS enhancement is possible.”

Comment 2: “Some technical information on the Raman measurements are also missing. What is the magnification of the objective used? What is the size of the area inspected? The spectral resolution? The spatial resolution? What is the power of the laser so that no side effect is induced (no heating or damage)?”

Response: The authors thank the reviewer for this comment and will like to clarify the missing technical information.

1. Excitation wavelength of 532nm, 638nm & 785 nm were used.
2. Other parameters - magnification of 20X, N.A.0.70
3. Inspected spot size was of 0.625 μm radius
4. Focal length of 250 mm
5. Solid state laser with excitation wavelength 532nm (5mW), 638nm (5mW) & 785nm (12.5mW)
6. Matched polarizer optics power of lasers
7. Spectral resolution of 0.5 cm^{-1} in visible and 1 cm^{-1} in NUV and IR & spatial resolution: < 1 μm (lateral), <2 μm (depth)

This information was incorporated in the supplementary information as follows –

1. Supplementary Method 4 – Enhancement Factor Calculation - Page 8- “Excitation wavelength of 532nm, 638nm & 785 nm were used. Other parameters like HC plan APO lense with matched polarizer/ analyzer optics (magnification of 20X, N.A.0.70), inspected spot size was of 0.625 μm radius, focal length of 250 mm , solid state laser with excitation wavelength 532nm (5mW), 638nm (5mW) & 785nm (12.5mW) , matched polarizer optics power of lasers and spectral resolution of 0.5 cm^{-1} in visible and 1 cm^{-1} in NUV and IR & spatial resolution: < 1 μm (lateral), <2 μm (depth) were used.”

Comment 3: “Although not the purpose of the work authors should explain the changes observed in the SERS spectra as a function of the concentration used, since they are showing these results. Why do the bands observed differ depending on the concentration? Is there any photodegradation? Photobleaching? Reactivity? Why is the enhancement factor increasing when the concentration is lower? Again: reactivity?”

Response: The authors will like to thank the reviewer for this comment.

1. The increased enhancement at decreased concentration was due to the decrease of tilt angle of the adsorbed analyte molecules providing more charge transfer resonance.
2. The decreased tilt angle must have been the reason for changed spectra.
3. The authors have added a reference for the said mechanism in the revised manuscript. S. Cong *et al.*, “Noble metal-comparable SERS enhancement from semiconducting metal oxides by making oxygen vacancies,” *Nat. Commun.*, vol. 6, p. 7800, 2015.

The manuscript was revised as follows-

1. Results and Discussion- SERS Based Molecular Detection – Page 11 & 12- “At lower concentrations, the enhancement increased indicating gradual decrease in the tilt angle on the substrate, resulting in more surface contact and hence more charge transfer[15] . This can also be the reason for change in the spectral data at lower concentrations.”

Comment 4:“The nanostructures providing maximum enhancement for model analytes crystal violet and R6G do not provide maximum enhancement for in-vitro SERS and it is also mentioned that the affinity of different cell lines for in-vitro enhancement varies based on the size of quantum probe. Is there any way to predict or to know what is the optimal size of the nanostructures depending on the target we want to analyse?”

Response: The authors will like to thank the reviewer for this comment.

1. SEM imaging demonstrated that the nanostructures with maximum enhancement for the analyte molecules was unable to allow cell adherence and proliferation, hence could not provide maximum in-vitro SERS enhancement.
2. The authors have undertaken three new experiments for time based study of cellular uptake of quantum probe which was helpful in understanding cellular behavior of probe uptake.
3. New samples were prepared and cell TEM imaging was undertaken.
4. From the TEM & fluorescence images, it was evident that the cellular uptake was related to the spectral intensity.
5. Raman spectra showed varied intensities with respect to the size of the quantum probe.
6. Thus, the authors concluded that optimization of the probe size is dependent on the ability of the cells for uptake of the probes which will be directly proportional to the cellular pore size which varies based on cell lines. By understanding this relationship can enable us to predict the optimal size of the probes for target cells.

The manuscript was revised as follows-

1. Results and Discussion- In-Vitro Biomolecule Analysis By ZnO Quantum Probe - Page 25, line 13” It should be noted here, that the nano-dendrites providing maximum enhancement for crystal violet & R6G could not provide maximum enhancement for in-vitro SERS. This happened because the cells displayed less or no ability of adherence and proliferation on the nano-dendrite platform with sharp needlike morphology displaying maximum SERS for the tag molecules. This shows the importance of functionalization of the nano dendrite platform for cell adhesion.”
2. Results and Discussion- In-Vitro Biomolecule Analysis By ZnO Quantum Probe - Page 25, line 20” The selection of the optimal quantum probe will have to be based on effective cellular uptake of the quantum probe. This mechanism will depend on the size of cellular pores, so by understanding the relationship between the pore size and quantum probe size, it is possible to engineer optimum quantum probes for maximum cellular SERS response.”

Comment 5: “How universal is this method in general for other cancer cells? Could **authors comment on this?**”

Response: The authors will like to state that this method is expected to be universal for various cancer cell lines. The cellular uptake mechanism will vary from cell to cell. Example – for an epithelial cell line like breast cancer, the percentage of quantum probe inside the cells will be much more than that of the cells of bone cancer due to the much bigger size of the bone cancer cells. So the enhancement from bone cancer cell may be far less than that from breast cancer cell. Further experimentation should be done to investigate this mechanism. The authors have not carried out this experimentation as it was not in the scope of the study.

The authors have added this information in the ‘In-Vitro biomolecule analysis by ZnO Quantum Probe’ Page 25 as follows- “It should be noted here, that the nano-dendrites providing maximum enhancement for crystal violet & R6G could not provide maximum enhancement for in-vitro SERS. This happened because the cells displayed less or no ability of adherence and proliferation on the nano-dendrite platform

with sharp needlelike morphology displaying maximum SERS for the tag molecules. This shows the importance of functionalization of the nano dendrite platform for cell adhesion. This method is expected to be universal for discrimination amongst various cancer cell lines. It is evident from figure 9 that the quantum probe displays varied cellular affinity displaying different results of SERS enhancement. The selection of the optimal quantum probe will have to be based on effective cellular uptake of the quantum probe. This mechanism will depend on the size of cellular pores, so by understanding the relationship between the pore size and quantum probe size, it is possible to engineer optimum quantum probes for maximum cellular SERS response.”

Comment 6: “The enhancement in the case of DNA, RNA, proteins and lipids is much lower than in the case of the model analytes. Could be this further optimized?”

Response: The authors will like to thank the reviewer for this comment.

1. The authors have undertaken three new experiments to study the cellular uptake of quantum probe.
2. The authors co-related cellular uptake to intensity of the biomolecular SERS in the time study.
3. Evidence of direct relationship between the time duration to cellular uptake and SERS intensity was observed.

The authors revised the manuscript with the relationship between time and SERS intensity as follows-

Results and Discussion - In-Vitro biomolecule analysis by ZnO Quantum Probe- Page 25 line10 “SERS intensity for cellular biomolecules is less as compared to the experiment based on tag molecules. This proves the importance of in-vitro sensing over stand-alone biomolecular sensing. More intense signal can be achieved by incubation with more concentration of quantum probes or incubation for more time enabling increased cellular uptake for more intensity.”

Comment 7: “Although the figures are very nice and easy to understand, some additional information should be included. In particular, in the case of the Raman spectra (which is the relevant result to be shown) wavenumber scale should be displayed. Additionally, it should be mentioned that the intensity scale is not real (or not the same for every condition shown).”

Response: Authors thank the reviewer for this comment. The authors have added the X axis (wavenumber cm^{-1}) and Y axis (counts) as part of all the spectral diagrams in all the figures as requested by the reviewer.

Comment 8: “What is labelled as discussion is rather a conclusions section.”

Response: The author have made the necessary changes in the headings as suggested by the reviewer.

Comment 9: “There are also some mistakes (for example in the caption of the figure 1 in the supplementary information it is mentioned that the wavelength is 780 but it should be 785 nm) and some typos.”

Response- The authors thank the reviewer for this comment. The authors have rectified all the errors and typos in the manuscript.

References:

- [1] N. T. Mai, T. T. Thuy, D. M. Mott, and S. Maenosono, “Chemical synthesis of blue-emitting metallic zinc nano-hexagons,” *CrystEngComm*, vol. 15, no. 33, pp. 6606–6610, 2013.

- [2] R. Al-Gaashani, S. Radiman, A. R. Daud, N. Tabet, and Y. Al-Douri, "XPS and optical studies of different morphologies of ZnO nanostructures prepared by microwave methods," *Ceram. Int.*, vol. 39, no. 3, pp. 2283–2292, 2013.
- [3] J. Lin, Y. Shang, X. Li, J. Yu, X. Wang, and L. Guo, "Ultrasensitive SERS detection by defect engineering on single Cu₂O superstructure particle," *Adv. Mater.*, vol. 29, no. 5, 2017.
- [4] S. Cong *et al.*, "Noble metal-comparable SERS enhancement from semiconducting metal oxides by making oxygen vacancies," *Nat. Commun.*, vol. 6, p. 7800, 2015.
- [5] X. Wang *et al.*, "Remarkable SERS activity observed from amorphous ZnO nanocages," *Angew. Chem. Int. Ed.*, 2017.
- [6] Wang Xiaotian, Shi Wensheng, She Guanwei, and Mu Lixuan, "Using Si and Ge Nanostructures as Substrates for Surface-Enhanced Raman Scattering Based on Photoinduced Charge Transfer Mechanism," *J. Am. Chem. Soc.*, no. 133, pp. 16518–16523, 2011.
- [7] J. R. Lombardi and R. L. Birke, "Theory of surface-enhanced Raman scattering in semiconductors," *J. Phys. Chem. C*, vol. 118, no. 20, pp. 11120–11130, 2014.
- [8] G. V. Naik and A. Boltasseva, "Semiconductors for plasmonics and metamaterials," *Phys. Status Solidi RRL-Rapid Res. Lett.*, vol. 4, no. 10, pp. 295–297, 2010.
- [9] S. Berweger, C. C. Neacsu, Y. Mao, H. Zhou, S. S. Wong, and M. B. Raschke, "Optical nanocrystallography with tip-enhanced phonon Raman spectroscopy," *Nat. Nanotechnol.*, vol. 4, no. 8, pp. 496–499, 2009.
- [10] T. Deckert-Gaudig, A. Taguchi, S. Kawata, and V. Deckert, "Tip-enhanced Raman spectroscopy—from early developments to recent advances," *Chem. Soc. Rev.*, vol. 46, no. 13, pp. 4077–4110, 2017.
- [11] O. M. Maragò, P. H. Jones, P. G. Gucciardi, G. Volpe, and A. C. Ferrari, "Optical trapping and manipulation of nanostructures," *Nat. Nanotechnol.*, vol. 8, no. 11, pp. 807–819, 2013.
- [12] A. B. Djurišić and Y. H. Leung, "Optical properties of ZnO nanostructures," *small*, vol. 2, no. 8-9, pp. 944–961, 2006.
- [13] E. Sachet, M. D. Losego, J. Guske, S. Franzen, and J.-P. Maria, "Mid-infrared surface plasmon resonance in zinc oxide semiconductor thin films," *Appl. Phys. Lett.*, vol. 102, no. 5, p. 051111, 2013.
- [14] R. L. Hengehold and F. L. Pedrotti, "Plasmon excitation energies in ZnO, CdO, and MgO," *J. Appl. Phys.*, vol. 47, no. 1, pp. 287–291, 1976.
- [15] D. Qi, L. Lu, L. Wang, and J. Zhang, "Improved SERS sensitivity on plasmon-free TiO₂ photonic microarray by enhancing light-matter coupling," *J. Am. Chem. Soc.*, vol. 136, no. 28, pp. 9886–9889, 2014.
- [16] Z. Dai *et al.*, "Obviously angular, cuboid-shaped TiO₂ nanowire arrays decorated with Ag nanoparticle as ultrasensitive 3D surface-enhanced Raman scattering substrates," *J. Phys. Chem. C*, vol. 118, no. 39, pp. 22711–22718, 2014.
- [17] L. Jin, G. She, X. Wang, L. Mu, and W. Shi, "Enhancing the SERS performance of semiconductor nanostructures through a facile surface engineering strategy," *Appl. Surf. Sci.*, vol. 320, pp. 591–595, 2014.
- [18] G. Sinha, L. E. Depero, and I. Alessandri, "Recyclable SERS substrates based on Au-coated ZnO nanorods," *ACS Appl. Mater. Interfaces*, vol. 3, no. 7, pp. 2557–2563, 2011.
- [19] H. Marks, M. Schechinger, J. Garza, A. Locke, and G. Côté, "Surface enhanced Raman spectroscopy (SERS) for in vitro diagnostic testing at the point of care," *Nanophotonics*.
- [20] D. Lin *et al.*, "Direct transfer of subwavelength plasmonic nanostructures on bioactive silk films," *Adv. Mater.*, vol. 24, no. 45, pp. 6088–6093, 2012.
- [21] Jun Deng, Mengyun Yao, and Changyon Gao, "Cytotoxicity of gold nanoparticles with different structures and surface-anchored chiral polymers," *Acta Biomater.*, no. 53, pp. 610–618, 2017.
- [22] Kyle D. Osberg, Matthew Rycenga, Gilles R. Bourret, Keith A. Brown, and Chad A. Mirkin, "Dispersible Surface-Enhanced Raman Scattering Nanosheets," *Advanced Mater.*, no. 24, pp. 6065–6070, 2012.
- [23] Jiawen Hu, Linghui Lu, Weiming He, Jianguo Pan, Weiyu Wang, and Jiannan Xiang, "Ligand exchange based water-soluble, surface-enhanced Raman scattering-tagged gold nanorod probes with improved stability," *Chem. Phys. Lett.*, no. 513, pp. 241–245, 2011.

- [24] C. Fauteux, R. Longtin, J. Pegna, and D. Therriault, "Fast synthesis of ZnO nanostructures by laser-induced decomposition of zinc acetylacetonate," *Inorg. Chem.*, vol. 46, no. 26, pp. 11036–11047, 2007.
- [25] L. Wen, R. Xu, Y. Mi, and Y. Lei, "Multiple nanostructures based on anodized aluminium oxide templates," *Nat. Nanotechnol.*, vol. 12, no. 3, pp. 244–250, 2017.

Reviewers' Comments:

Reviewer #1:

Remarks to the Author:

Although the author has addressed most of my previous questions, the background of semiconductor-based SERS is not satisfied. For example, the authors believe that the enhancement factor (EF) of semiconductor nanostructures is only about 10~100 orders of magnitude (page 5, line 6), while the recent EF of semiconductor nanostructures have already achieved at the order of 10^4 to 10^5 (Angew. Chem. Int. Ed., volume 56, Issue 33, 2017, 9851-9855; Advanced Materials, 2017, 29, 1604797.; Small, 2018, 14, 1703274.)

In addition, the description of some references are not accurate. For example, the very recent result (Angew. Chem. Int. Ed., volume 56, Issue 33, 2017, 9851-9855) reported by Wang et al. is that the numerous metastable electronic states of amorphous semiconductor nanostructures can facilitate the interfacial charge transfer and effectively increase the molecular polarization, resulting in the remarkable SERS activity of amorphous ZnO nanocages.

If the author can carefully investigate the background and revised the introduction section, I would suggest its publication in Nature Communication.

Reviewer #2:

Remarks to the Author:

The authors reported ZnO-based nano dendrite structures for cancer cell detection by SERS

1. First of all, I have serious concerns on claiming cancer or cancer cell detection in this study. The authors measured general biomolecules (DNA, RNA, Proteins, lipids) present in most mammalian cells and thus identifying cancer cells based on these measurements would be technically challenging. Even if there are differences among HeLa, MDAMB-231, NIH 3T3 cells measured here, these could be due to several other factors including cell-to-cell variations, human (HeLa, MDAMB-231) vs mouse (NIH 3T3) cell lines, epithelial (HeLa, MDAMB-231) vs fibroblast (NIH 3T3). No evidence was provided if the SERS mapping of protein, DNA, and RNA are specific and correct. Also the measurements were done after air-drying the cells, which could cause different shapes and biomolecule distributions depending on how these cells were dehydrated. I am also worried about over-fitting issues in PCA analyses as there are insufficient measurement data compared to the number of variables.

2. The authors pointed out coagulation, selectivity and advise biocompatibility as limitations of Au and Ag nanoparticles, but these have been addressed by using proper surface coating and functionalization of Au nanoparticles. In fact, these nanoparticles have been used for both in vitro and in vivo cell imaging and SERS measurements. The author also mentioned that specific targeting is necessary for plasmonic materials, but indeed this is a critical step to identify cancer cells by measuring over-expressed cancer biomarkers. It is also unclear if the ZnO nanoparticles shown in this study are free of these limitations.

3. It is unclear how the particle aggregates get into the intracellular environments. EM images in Figs. 1 and 2 show big aggregates of ZnO nanoparticles attached on bigger dendrite structures. Are these aggregates come out of the dendrite structures and get into cells? EM images in Fig. 10 show dispersed single particles inside of cells. Are these from the aggregates attached on the dendrite structures? How are the enhancement factors different between aggregated and dispersed nanoparticles?

4. The author should provide more information to claim the EF of 10^6 . Which SERS peak did the author use to calculate the EF? How did the author calculate the surface area of the nanostructures (i.e. N_{surf})? Did the author take account of different sizes and aggregation of particles in the calculation? In overall, several figure labels are too small to read and legends do not provide

detailed information.

In overall, I believe this study is more suitable for specialized journals in material sciences focusing on studying enhancement mechanisms of the nanostructures.

Reviewer #3:

Remarks to the Author:

The work has been improved after implementing all the suggestions made by the reviewers and questions have been answered in a proper way. Information which was missing in the previous version has been now included together with additional references and results corresponding to new experiments. As a result, the manuscript is more complete and clearer for the reader and the community. As a minor comment there are still some errors or typos which should be corrected before the publication.

Point by point response to the reviewers' comments

With reference to your letter regarding revision of the manuscript entitled "Non Plasmonic Semiconductor Quantum SERS Probe – Pathway For *In-Vitro* Detection" by Rupa Haldavnekar, Dr. Bo Tan and I, we will like to thank you and the reviewers for the feedback on the revised manuscript. We have addressed the reviewers' comments by undertaking new experimentation and new analysis. The new results are incorporated in the manuscript. Following new experiments, data gathering and analysis were undertaken for submission of this major revision of the original manuscript.

1. In order to validate cellular uptake of the quantum probe by energy dependent endocytosis mechanism, a new experiment of incubation of the cells with the quantum probes at 37°C and 4°C was undertaken. The cells exposed to the quantum probe were then analyzed with the help of TEM analysis. (Reviewer #2) New samples were prepared, new imaging & analysis was undertaken to prove energy dependent cellular uptake mechanism.
2. A new figure (Fig 6) for cellular uptake mechanism was incorporated with the new data.
3. In order to demonstrate the ability of the ZnO based semiconductor quantum probe for *in-vitro* cancer diagnosis, additional ratio analysis was undertaken for cell classification. The data was analyzed and results were incorporated in the manuscript. (Reviewer #2). This ratio analysis was undertaken to substantiate the results of two types of multi-variate analyses previously undertaken.
4. Figures 7, 8, 9 & 10 were modified and new data was added for more clarity. (Reviewer #2)
5. Details of enhancement factor methodology were incorporated in "Methods" for more clarity. (Reviewer #2)
6. Careful investigation on background of semiconductor based SERS was undertaken and introduction section was modified with more in-depth information. (Reviewer #1)
7. Manuscript was checked for all typos and errors were corrected. (Reviewer #3)

Point-by-point responses to the reviewer's remarks are outlined below.

Reviewer #1 (Remarks to the Author):

Comment : *Although the author has addressed most of my previous questions, the background of semiconductor-based SERS is not satisfied. For example, the authors believe that the enhancement factor (EF) of semiconductor nanostructures is only about 10~100 orders of magnitude (page 5, line 6), while the recent EF of semiconductor nanostructures have already achieved at the order of 10⁴ to 10⁵ (Angew. Chem. Int. Ed., volume 56, Issue 33, 2017, 9851-9855; Advanced Materials, 2017, 29, 1604797.; Small, 2018, 14, 1703274.) In addition, the description of some references are not accurate. For example, the very recent result (Angew. Chem. Int. Ed., volume 56, Issue 33, 2017, 9851-9855) reported by Wang et al. is that the numerous metastable electronic states of amorphous semiconductor nanostructures can facilitate the interfacial charge transfer and effectively increase the molecular polarization, resulting in the remarkable SERS activity of amorphous ZnO nanocages.*

If the author can carefully investigate the background and revised the introduction section, I would suggest its publication in Nature Communication.

Response: The authors thank the reviewer for this comment. The authors have carefully investigated the recent publications related to semiconductor based SERS and have incorporated following paragraph in the introduction section.

Introduction- Page 5: In the past, SERS obtained with semiconductor based nanostructures was typically quite low in the range of 10⁻¹ -10², many novel strategies have been explored recently to improve this performance. Remarkable SERS activity of amorphous ZnO nanocages due to the numerous metastable electronic states of amorphous semiconductor nanostructures facilitating interfacial charge transfer amplifying molecular polarization was reported by Wang et al [1]. Reports on vibrational coupling between surface defects like oxygen vacancies and molecules and morphology induced magnification of substrate-analyte molecule interaction enhancing SERS were presented by Cong et al [2]. Lin et al reported defect engineering strategy facilitating photo induced charge transfer in addition to vacancy defect induced

electrostatic adsorption strategy for SERS enhancement up to nano molar concentration [3]. Charge transfer efficiency was improved by vibronic coupling of the conduction and valence band in a molecule-semiconductor system to improve SERS performance by Wang et al [4]. Facet dependent SERS effect in semiconductors improving sensitivity due to interfacial charge transfer leading to large molecular polarization was investigated by Lin et al [5]. So, there is a considerable amount of increased interest in exploration of semiconductor based SERS.

A new reference - J. Lin *et al.*, "Direct Experimental Observation of Facet-Dependent SERS of Cu₂O Polyhedra," *Small*, vol. 14, no. 8, p. 1703274, 2018 was added.

Reviewer #2 (Remarks to the Author):

Comment 1: *"First of all, I have serious concerns on claiming cancer or cancer cell detection in this study. The authors measured general biomolecules (DNA, RNA, Proteins, lipids) present in most mammalian cells and thus identifying cancer cells based on these measurements would be technically challenging. Even if there are differences among HeLa, MDAMB-231, NIH 3T3 cells measured here, these could be due to several other factors including cell-to-cell variations, human (HeLa, MDAMB-231) vs mouse (NIH 3T3) cell lines, epithelial (HeLa, MDAMB-231) vs fibroblast (NIH 3T3). No evidence was provided if the SERS mapping of protein, DNA, and RNA are specific and correct. Also the measurements were done after air-drying the cells, which could cause different shapes and biomolecule distributions depending on how these cells were dehydrated. I am also worried about over-fitting issues in PCA analyses as there are insufficient measurement data compared to the number of variables."*

Response: The authors thank the reviewer for this valuable comment. The authors will like to state that the main objective of the manuscript was to develop a new non-plasmonic probe for Surface Enhanced Raman Scattering as a method for label-free, simultaneous, multiple sensing. The authors' strategy to generate substantial SERS from a material with typically poor SERS response was to reduce the size of the probe to quantum scale. By reducing the size to quantum scale, the authors have observed a new phenomenon of exponential increase in the SERS response. The quantum probes were decorated on a nano dendrite platform mimicking the extra cellular matrix functionalized for cell adhesion and proliferation. This type of ZnO based, semiconductor quantum probe for label-free and simultaneous *in-vitro* cancer detection was used for the first time to the best of the authors' knowledge. This type of novel research will be of paradigm importance in the field of sensing, specifically biomedical sensing.

1. The main objective of this manuscript was to demonstrate the ability of ZnO based non-plasmonic semiconductor quantum probe to provide SERS enhancement for *in-vitro* cancer detection which the authors have successfully demonstrated for the first time to the best of the authors' knowledge. The authors have used established Raman peaks from the literature in this study [6], [7], [8], [9], [10],[11], [12]. The authors' results are concurrent and matching with the known literature.
2. For SERS mapping, the authors have used standard "Wire 3.3" software which has been developed by Renishaw for spectroscopy as well as Raman mapping. Before starting of the experiment, calibration of the software was tested with silicon peak assignment and two Raman reporter dyes - R6G and crystal violet as well as two more Analytes (4ATP & 4MBA) for correctness of the data. The peak assignment agreeable results matching with the peaks reported in the literature. Thus, the authors concluded on correctness of calibration of the software. The Raman mapping to assign the peak positions of proteins, lipids, DNA & RNA were based on previously reported Raman bands from the literature. The functionality of Wire software was used to obtain cellular Raman mapping.
3. The authors have added more detailed information in the "Methods" in order to bring more clarity for the methodology to address the reviewer's concerns.
4. Use of PCA by scientists for various discriminations using cells and other biological samples is a standard practice [13], [14]. Since, Raman spectra are very complex and there is a possibility of overlap over a broad band, the authors extended the qualitative analysis by undertaking two types of multi-variate analyses (principal component analysis and discriminant analysis) to avoid introduction of error due to qualitative analysis. The results of discriminant analysis are in agreement with the results of PCA.

5. In addition, in this revision, the authors have undertaken a new statistical ratio analysis for classification of cancer and non-cancer cells in order to substantiate results derived from PCA & DA.
6. More data was added in Figure No 7, 8 & 9.
7. More references were added [8], [15], [13], [16].

The accuracy of the *in-vitro* cell analysis was reinforced by undertaking additional ratio analysis. The authors have brought more clarity to the manuscript by adding and modifying information in the Methodology, Results & Discussion as follows:

1. Abstract- Page 2: "The results for *in-vitro* sensitivity were statistically analyzed demonstrating cell discrimination ability along-with *in-vitro* biomolecular sensing of DNA, RNA, proteins and lipids for cancer cells."
2. Introduction- Page 7: "Raman bands were analyzed for peak positions, intensity to demonstrate differences between cancer and non-cancer cells. The results were also statistically analyzed using combination of principal component analysis (PCA) and discriminant analysis (DA). Well defined clustering of cell lines during time series analysis as well as for cell discrimination was observed. Ratio of peak intensities of lipids and proteins (I_{1445}/I_{1654}) was analyzed for classification of cells into cancerous and non-cancerous cells."
3. In-Vitro Cell Discrimination by ZnO Quantum Probe – Page 24,25: Discussion was modified as follows: "Fig 9 shows distinct spectral differences for cancer cell lines and non-cancer cells [13]. Breast cancer (MDAMB231) showed strong vibrational modes amide I & amide III peaks in the region of 2800 cm^{-1} - 3000 cm^{-1} . Presence of excess lipids indicated by peaks at 879 cm^{-1} , 1095 cm^{-1} , 1304 cm^{-1} and 1450 cm^{-1} were marked as a signature of malignancy in breast cancer cells [15]. HeLa cells showed high concentration of *trans* form 1667 cm^{-1} & *cis* form 1660 cm^{-1} unsaturated lipids [13]. Peak at 1304 cm^{-1} was due to in-phase CH_2 twist mode while the peak at 1439 cm^{-1} was due to bending modes occurring due to degenerated deformation of hydrocarbon chains in HeLa. Also high concentration of lipids - 1304 cm^{-1} CH_2 deformation (lipid), 1313 cm^{-1} CH_3CH_2 twisting mode of collagen (lipid), 1319 cm^{-1} CH_3CH_2 twisting (collagen assignment), 1324 cm^{-1} CH_3CH_2 wagging mode present in collagen was expressed by HeLa. At 1000 cm^{-1} , cancer cell lines showed more intense, almost identical peak for Phenylalanine, for proteins. Cancer cell lines showed CH_2 deformation at 1438 cm^{-1} & 1454 cm^{-1} . Cancer cell lines showed high concentration of Nucleic acid at 1662 cm^{-1} . In the region of 2920 cm^{-1} to 2960 cm^{-1} , cancer cell lines showed overexpression of C-H vibrations in lipids & proteins, CH_2 asymmetric stretch. Cancer cell lines showed overexpression of cholesterol & cholesterol ester at $418, 429, 538, 548, 608, 614$ & $960, 1440, 1441, 1444\text{ cm}^{-1}$ indicating high concentration of cholesterol, signature of cancer due to mitochondrial membrane damage. At 1450 , cancer cell lines show CH_2 bending mode in malignant tissues which is absent in the non-cancer cells. The non-cancer NIH3T3 cells showed de-formative vibrations of quinoid ring at 1161 cm^{-1} , antisymmetric phosphate stretching vibration at 1230 cm^{-1} which were absent in cancer cells. Peak at 1750 due to $\text{C}=\text{O}$ for lipid in normal tissues showed low intensity in cancer cells. The discrimination was further substantiated by ratio of Raman intensities at 1445 cm^{-1} to 1655 cm^{-1} . The bands at 1445 cm^{-1} and 1655 cm^{-1} display sensitivity to histological abnormalities. The band at $\sim 1655\text{ cm}^{-1}$ (proteins and phospholipids) and 1445 cm^{-1} (Phospholipids and C-H scissor in CH_2 due to protein to lipid ratio) [16], [17]. The mean value of the ratio was 1.00 ± 0.11 for HeLa & 0.82 ± 0.19 for MDAMB separating clearly from non-cancer cell line of NIH3T3 2.34 ± 2.26 . Two multivariate analyses (Principal Component Analysis-PCA and Discriminant Analysis-DA) were used to further confirm the earlier discrimination. Eigen values of F1, F2 and F3 were used to draw a 3-D scatter plot of F1 Vs F2 Vs F3. F1 interprets 45.96 % of variances, F2 interprets 20.67 % of variances and F3 interprets 14.02 % of variances. Top 3 principal components contributed to 80.68 % of cumulative contribution covering most of the information from original spectra in the three dimensional space. From the cluster graphs, it is evident that there are structural differences in all three cell lines. We were able to discriminate the cell lines with PCA. The peak at 917 cm^{-1} which belongs to the ribose vibration of RNA was used as a supplementary quantitative data to understand variation between the cell lines. It can be seen from the centroids, that HeLa & MDAMB (cancer cells) show more activity as compared to normal cells. This can also be interpreted as more tumor suppressing genes from mRNA are altered in cancers as compared to normal cells. A plot based on

discriminant analysis was obtained which shows clear clusters of various cell types. The classification rates were 100 % with first two discriminant functions F1 & F2.”

4. Methods- Cell Culture, Cell Fixation and Cell Analysis- Page 32,33: “Spectra of different cell lines as well as at different time intervals were then analyzed using widely accepted biomolecular Raman assignments from the literature in order to identify the effect of specific resonances for cellular Raman enhancements. Please refer to following table for Raman peak assignments for the cells. Spectral bands of various biomolecules were analyzed to distinguish between cancer cells and non-cancer cells fixed at 24 hr time interval using signature Raman assignments for HeLa as well as MDAMB231 cancer cell lines widely accepted in the literature [15], [13], [18], [19] and comparing with a non-cancer mammalian cell line of NIH3T3. Although Raman spectroscopy can provide fingerprint spectra of various cancer cell lines, the peaks generated due to proteins, lipids, nucleic acids, polysaccharides are very complex and can overlap over a broad band. There is a possibility of introducing inaccuracies in the interpretation of data when the peaks are analyzed qualitatively. In order to reduce the errors due to guessing of biochemical components, qualitative as well as quantitative analysis of Raman bands was performed [20]. Combination of multivariate analysis of Principal components (PCA) and Discriminant analysis (DA) was employed. PCA analysis is a non-parametric approach which does not need any explicit background model [21]. Dimensionality of the data can be reduced with PCA, as mathematical decomposition of Raman spectra into principal components is done. PCA is effectively used in Raman studies for discrimination of cells [20] [14] [13]. Since cells contain over 10,000 to 15,000 mRNAs, it is possible to understand approximately 150 tumor suppressing genes from the study of RNA [22], [23]. The intensity values of RNA wavenumber at 915 cm⁻¹ were used as an independent variable for discriminant analysis for discrimination of cell lines. In order to distinguish between cancer and non-cancer cell lines, ratio of Raman intensities at 1445 cm⁻¹ and 1655 cm⁻¹ were compared. This ratio has been used for classification between normal and malignant cells in the brain, breast, colon and cervix [24]. The sensitivities of the band for CH₂ scissoring for proteins at 1445 cm⁻¹ and the band for C=O stretching of collagen and elastin for lipids, for histological abnormalities has been widely utilized as a test for malignancy in the literature. The decision ratio (I_{1445}/I_{1655}) is able to separate malignant cells from normal cells. Scatter plots for all three cell lines were plotted for this classification.”
5. Additional table for cellular peak assignments was added to the “Methods” (page 35)
6. Fig 7(Page 23) was modified with additional SERS spectra of various cell lines on nanostructures decorated with quantum probe compared with the spectra on native samples to demonstrate the ability of the quantum probe for SERS enhancement.
7. Fig 8 (Page 24) was modified and time dependent cell TEM images were incorporated to demonstrate direct relationship between the SERS enhancement and cellular uptake and internalization of the quantum probe.
8. Fig 9 (page 25) was modified and additional ratio analysis was incorporated by a scatter plot for lipid to protein ratio. A column chart with mean & standard deviation for the ratio was added. Cell TEM were incorporated to demonstrate the relationship between ability of the quantum probe to provide simultaneous multiple detection of biomolecules and position of the quantum probe near cell membrane, cytoplasm as well as nucleus.
9. Methods – Page 34 : “Single cells on the nanostructures were identified with the help of Leica DMI 6000 epifluorescence microscope. Raman mapping was carried out at 3 X 3 steps for the entire selected area. In all 318 spectra for all cell lines for 24-hour samples were obtained. 2 D maps of intensity at 752 cm⁻¹ , 915 cm⁻¹ , 1030 cm⁻¹, 1334 cm⁻¹, 1655 cm⁻¹, 2854 cm⁻¹, 2940 cm⁻¹ (2-D intensity maps for wavenumbers representing various contents of a cell like DNA, RNA, Nucleic acid, lipids and proteins) were generated and were then merged to get the final map of single cell. The single cell maps were built with the help of in-built functionality of wire 3.3 software.”
10. Additional references were added –
 - M. V. P. Chowdary, K. K. Kumar, J. Kurien, S. Mathew, and C. M. Krishna, “Discrimination of normal, benign, and malignant breast tissues by Raman spectroscopy,” *Biopolymers*, vol. 83, no. 5, pp. 556–569, 2006.
 - Z. Huang, A. McWilliams, H. Lui, D. I. McLean, S. Lam, and H. Zeng, “Near-infrared Raman spectroscopy for optical diagnosis of lung cancer,” *Int. J. Cancer*, vol. 107, no. 6, pp. 1047–1052, 2003.

- K. Gajjar *et al.*, “Diagnostic segregation of human brain tumours using Fourier-transform infrared and/or Raman spectroscopy coupled with discriminant analysis,” *Anal. Methods*, vol. 5, no. 1, pp. 89–102, 2013.
- C. H. Liu *et al.*, “Human breast tissues studied by IR Fourier-transform Raman spectroscopy,” in *Conference on Lasers and Electro-Optics*, 1991, p. CWF51
- C. J. Frank, R. L. McCreery, and D. C. Redd, “Raman spectroscopy of normal and diseased human breast tissues,” *Anal. Chem.*, vol. 67, no. 5, pp. 777–783, 1995.

Comment 2: “The authors pointed out coagulation, selectivity and adverse biocompatibility as limitations of Au and Ag nanoparticles, but these have been addressed by using proper surface coating and functionalization of Au nanoparticles. In fact, these nanoparticles have been used for both *in vitro* and *in vivo* cell imaging and SERS measurements. The author also mentioned that specific targeting is necessary for plasmonic materials, but indeed this is a critical step to identify cancer cells by measuring over-expressed cancer biomarkers. It is also unclear if the ZnO nanoparticles shown in this study are free of these limitations.”

Response: The authors thank the reviewer for bring up this important comment and would like to state the following. The objective of this manuscript was to find a semiconductor based non-plasmonic material as an alternative to plasmonic SERS which can be used as a label free platform for simultaneous, multiple detection especially in the case of a heterogeneous and complex disease like cancer.

1. The authors agree with the reviewer about the ability of the functionalized Nobel metal nano particles demonstrate in cell imaging and SERS applications. Although functionalized gold & silver based nanoparticles can provide information on specific biomarkers responsible for cancer, it has been demonstrated in the literature that single markers/ specific markers are inadequate to provide complete information on a heterogeneous disease like cancer [25]. Hence there is a need for simultaneous, multiple detection. Simultaneous, multiple detection will also enable in monitoring progression of a disease [26]. That way, generation of significantly more information from a small amount of sample is possible. The objective of this manuscript was to develop a new material in order to obtain such simultaneous, multiple detection.
2. SERS with Au & Ag is very challenging and has many inherent limitations. Also, it is very difficult to use Au & Ag based nano-structures for wide practical applications (especially cellular application) due to the coagulation, cost, optical loss, limited wavelength range, need for functionalization with some label and adverse biocompatibility [27]. The use of labels can lead to contamination of spectra by introduction of experimental artefacts. Labels can also adversely affect the integrity of cellular structure. So, the authors were in search of a label-free platform for *in-vitro* cancer detection. The authors have successfully demonstrated the functionalization of the label-free nano-dendrite platform for cell adhesion. The quantum probe decorated on the dendrites, were functionalized for SERS demonstration in this study with a ZnO based non-plasmonic, semiconductor material for *in-vitro* cancer detection for the first time to the best of the authors’ knowledge.
3. Also, Nobel metal based research has led to well understood chemical and physical properties of the nano particles, which is not the case with non-Nobel material based SERS. So, the researchers are continuously expanding their scope to get better understanding of non-Nobel material based SERS [27].
4. The authors have demonstrated dominance of charge transfer mechanism for SERS enhancement. Since the charge transfer occurs when the analyte of interest gets physically or chemically adsorbed on the quantum probe, more adsorption will lead to more SERS. Since, this type of enhancement is not dependent on formation of hot spots like the electromagnetic enhancement in plasmonic SERS, coagulation will not play any adverse effect on the enhancement factor. The authors have demonstrated this property through single cell mapping.
5. More cellular data was gathered and added in Figure 8.

The authors have added a new paragraph on the need for this new non-plasmonic, semiconductor based material with biocompatible, label-free properties. This is reinforced in the manuscript as follows:

1. Abstract – Page 2: “Current work in SERS focusses on plasmonic nano materials which suffer from coagulation, selectivity and adverse biocompatibility when used *in-vitro*.”

2. Introduction- Page 4: “Although the enhancement observed with plasmonic metal nanoparticles is substantial, it suffers from coagulation[28], selectivity[29], cost, optical loss, limited wavelength range as well as adverse biocompatibility [1],[4]. In order for the highly localized Raman hot spots to remain discrete, plasmonic materials often need surfactants to provide good SERS response. This is very challenging due to uncontrolled agglomeration of these materials[30]. This type of materials also need to be functionalized for specific targeting with SERS active tags [31] which can result in a contaminated spectra. It can also adversely affect the integrity of cellular structure [32].”

The importance of need for multiple SERS for simultaneous detection was reinforced in the manuscript as follows:

1. Introduction- Page 4: “Identification of and information on specific cancer biomarkers are not adequate to provide complete information of a heterogeneous and complex disease like cancer. Hence, it is necessary to get simultaneous information on multiple biomarkers for robust diagnosis and disease monitoring of cancer [25].”

The authors have brought more clarity to the desirable properties of ZnO based semiconductor material by adding and modifying information in the manuscript as follows:

1. Introduction- Page 4: “ZnO quantum crystals have high surface area, good crystallinity as well as demonstrated biocompatibility which makes it very desirable for multiple applications of sensing and diagnostics [33]. ZnO can dissolve in acidic as well as basic conditions. So, if applied to a tumor cell, there is a very high probability of ZnO getting dissolved into Zn^{2+} & O_2^- which follows use of both the ions by the cells [34].”
2. Results & Discussion- Page 18: “Label-Free Cell Adhesion On Nano Dendrite Platform- In conclusion, we were successful in functionalizing the nano-dendrite platform for label-free cell adhesion and proliferation. We were also able to program the size and shape of the nano-dendrite platform to enhance self-targeting, cell adhesion and cell proliferation for label-free *in-vitro* application.”
3. Fig 8 was modified to demonstrate label-free, simultaneous *in-vitro* application to incorporate time dependent cell TEM.
4. Discussion on In-Vitro Single Cell Raman mapping with ZnO quantum antenna was modified and discussion on effect of aggregation was added as follows: “Since the quantum probe demonstrated dominance of the charge transfer mechanism in SERS enhancement, the aggregation of quantum probe in the solution or in the lysosomes of a cell played no adverse effect on the enhancement. This was evident from the many spectra obtained for the entire single cell which did not show any sudden increase or reduction in the peak intensities (Fig 9).”

Comment 3: *“It is unclear how the particle aggregates get into the intracellular environments. EM images in Figs. 1 and 2 show big aggregates of ZnO nanoparticles attached on bigger dendrite structures. Are these aggregates come out of the dendrite structures and get into cells? EM images in Fig. 10 show dispersed single particles inside of cells. Are these from the aggregates attached on the dendrite structures? How are the enhancement factors different between aggregated and dispersed nanoparticles?”*

Response: The authors will like to thank the reviewer for this important comment.

1. The quantum probes are decorated on a nano-dendrite platform loosely with weak bonds, so the quantum probe in solution can get easily dissociated from the dendrites into the solution in order to be internalised by the cells. This information is added in characterization of the quantum probe for more clarity.
2. The quantum probe under cellular SEM look like aggregates due to the drying effect during sample preparation for imaging. The quantum probes exist as particles in the solution and are internalized by the cells by endocytosis.
3. The authors have undertaken extensive new experimentation to demonstrate the energy dependent endocytosis of the quantum probes. This type of energy dependent endocytosis has been recently observed by Kim et al [35], McParland et al [36].

4. In order to substantiate the evidence of presence of quantum probe inside the cells, the authors have employed EDX mapping in addition to cell TEM. Presence of quantum probes is evident throughout the cell from cell TEM as well as EDX.
5. The effect of aggregated and dispersed nanoparticles had no adverse effect on the enhancement factor. This is due to the dominance of charge transfer mechanism in the SERS enhancement of ZnO based non plasmonic semiconductor quantum probe. This is evident from the single cell Raman mapping. The spectra collected from a single cell do not show any sudden increase or decrease in the intensity of the peaks.
6. More cellular data was gathered and added in Figure 6 & 10.

The cellular uptake mechanism and additional discussion on aggregated and dispersed quantum probe is reinforced as follows:

1. Results & Discussion- (Page 8): “The bonds between the quantum crystals and nano rods were weak and quantum crystals were easily detachable from the nanorods.”
2. Results & Discussion (Page 18) : New Discussion on cellular uptake mechanism was incorporated as follows. “The cellular uptake of the quantum probes was through endosomes. Many quantum probes were trapped inside lysosomes dispersed throughout the cytoplasm. Many quantum probes were visible on cell membrane and throughout the cytoplasm. The cellular TEM of cells incubated at 37°C showed internalization of the quantum probe whereas at 4°C presence of the quantum probes was not evident (Fig 6). This indicated energy dependent internalization process of endocytosis. This happened because the active transport mechanism of the cells gets blocked at 4°C due to less consumption of ATP by the cells demonstrating very less or no quantum probes inside the cells in the TEM [35].”
3. In-Vitro biomolecule analysis (Page 26): “The cell TEM as well as the fluorescence images in Fig 10 show cellular uptake of quantum probes. This is also supported by EDX map of the cells showing presence of zinc.”
4. Figure 6 : New TEM images were added and figure was modified to substantiate new discussion on cellular uptake mechanism.
5. Figure10: New additional spectra for the entire cell were added to demonstrate no adverse effect on SERS enhancement due to aggregation. Additional TEM images demonstrating the position of quantum probe near cell membrane (SERS signal for lipids), cytoplasm (SERS signal for proteins) & nucleus (SERS signal for DNA) reinforcing simultaneous SERS detection & mapping of various biomolecules.
6. Results & Discussion (Page 28): In-Vitro Single Cell Raman mapping with ZnO quantum antenna was modified and discussion on the effect of aggregation was added as follows: “Since the quantum probe demonstrated dominance of the charge transfer mechanism in SERS enhancement, the aggregation of quantum probe in the solution or in the lysosomes of a cell played no adverse effect on the enhancement. This was evident from the many spectra obtained for the entire single cell which did not show any sudden increase or reduction in the peak intensities (Fig 10).”

Comment 4: “The author should provide more information to claim the EF of 10^6 . Which SERS peak did the author use to calculate the EF? How did the author calculate the surface area of the nanostructures (i.e. N_{surf})? Did the author take account of different sizes and aggregation of particles in the calculation?”

Response: The authors thank the reviewer for this comment.

1. The authors have calculated the enhancement factor with two Raman reporter dyes (Crystal violet and Rhodamine 6G) and extended this study to two Analytes (4 ATP, 4 MBA) which can covalently adsorbed on the quantum probe.
2. The authors undertook extensive new experimentation with three different excitation wavelengths to substantiate dominance of charge transfer mechanism in the SERS signal.
3. Three different concentrations of the Analytes were employed to get the limit of detection.
4. The authors have used the standard protocols for calculation of enhancement factor was derived from the EF calculation used in the article “Cyclic Electroplating and Stripping of Silver on Au@SiO₂ Core/Shell Nanoparticles for Sensitive and Recyclable Substrates of Surface-enhanced Raman Scattering” by Li et-al [37].

5. In order to address the reviewer's concerns, the authors have included the standard protocol used in this study in "Methods".

The information on EF is reinforced as follows:

Methods – SERS based molecular detection (page 31): "Enhancement factor was calculated crystal violet (CV) and Rhodamine 6G (R6G) at 1 Milimole, 1 micromole and 1 Nano mole concentration and 4-ATP & 4-MBA for milimole concentration. 10 µg of the solution was dropped directly on the nanostructures and native substrate. The solution was allowed to dry and Raman spectra of the air dried samples were taken. Renishaw inVia Raman microscope was used for this analysis. Excitation wavelength of 532nm, 638nm & 785 nm were used. Other parameters like HC plan APO lenses with matched polarizer/ analyzer optics (magnification of 20X, N.A.0.70), inspected spot size was of 0.625 µm radius, focal length of 250 mm, solid state laser with excitation wavelength 532nm (5mW), 638nm (5mW) & 785nm (12.5mW), matched polarizer optics power of lasers and spectral resolution of 0.5 cm⁻¹ in visible and 1 cm⁻¹ in NUV and IR & spatial resolution: < 1 µm (lateral), <2µm (depth) were used.

The enhancement factor (EF) can be derived from the following equation.

Enhancement Factor = $(I_{\text{Quantum Probe}} / I_{\text{Native}}) * (N_{\text{Native}} / N_{\text{Quantum Probe}})$ ----- Equation 1

$N_{\text{Quantum Probe}} = \Gamma \times \pi \times r^2$ where $\Gamma = 1 / (\text{Surface density} * \text{Avogadro number}) \text{mole/cm}^2$ ----- Equation 2

$N_{\text{bulk}} = \pi r^2 h \rho / \text{atomic weight}$ ----- Equation 3

This equation was derived from the EF calculation used in the article "Cyclic Electroplating and Stripping of Silver on Au@SiO₂ Core/Shell Nanoparticles for Sensitive and Recyclable Substrates of Surface-enhanced Raman Scattering" by Li et-al [37]. $I_{\text{Quantum Probe}}$ & I_{Native} are the intensities of the characteristic peaks associated with the Analytes adsorbed on the quantum probe. Peak at 1621 cm⁻¹ is assigned to the strongest CV vibrational mode of c-c ring stretching. (In this case, this peak was red shifted to 1618 cm⁻¹). This peak was used for calculating enhancement at the milimole concentration. For micromole and Nano mole concentrations, peak at 1621 cm⁻¹ disappeared. So we have used the intensity of peak at 917 cm⁻¹ assigned to ring skeletal vibration for the other two concentrations. For R6G, the vibrational mode at 1512cm⁻¹ was used to calculate the enhancement factor for milimole concentration. But this peak is absent at micromole & Nano mole concentration, so the peak at 1276 cm⁻¹ assigned to C-O-C stretching was used. $N_{\text{Quantum Probe}}$ is calculated based from the estimation of concentration of the analyte on the surface using surface density of the quantum probe. N_{bulk} is calculated using ρ the density of the molecule of the analyte, atomic weight and h the confocal depth of the Raman Laser Beam.

For wavelength dependent SERS experiment, 4-ATP & 4-MBA molecules were used in addition to R6G and Crystal Violet. For 4-ATP, the peak at 1590 and for 4 MBA the peak at 1596 assigned to C=C stretching mode of benzene ring was used to calculate the enhancement factor at milimole concentration. Identification of b₂ modes for all four Analytes was undertaken in order to identify the dominance of a particular resonance in the SERS enhancement with the quantum probes."

Reviewer #3 (Remarks to the Author):

Comment: "The work has been improved after implementing all the suggestions made by the reviewers and questions have been answered in a proper way. Information which was missing in the previous version has been now included together with additional references and results corresponding to new experiments. As a result, the manuscript is more complete and clearer for the reader and the community. As a minor comment there are still some errors or typos which should be corrected before the publication."

Response: The authors thank the reviewer for this comment. The authors have carefully edited the manuscript and rectified the typo errors.

References:

- [1] X. Wang *et al.*, "Remarkable SERS activity observed from amorphous ZnO nanocages," *Angew. Chem. Int. Ed.*, 2017.
- [2] S. Cong *et al.*, "Noble metal-comparable SERS enhancement from semiconducting metal oxides by making oxygen vacancies," *Nat. Commun.*, vol. 6, p. 7800, 2015.

- [3] J. Lin, Y. Shang, X. Li, J. Yu, X. Wang, and L. Guo, "Ultrasensitive SERS detection by defect engineering on single Cu₂O superstructure particle," *Adv. Mater.*, vol. 29, no. 5, 2017.
- [4] Wang Xiaotian, Shi Wensheng, She Guanwei, and Mu Lixuan, "Using Si and Ge Nanostructures as Substrates for Surface-Enhanced Raman Scattering Based on Photoinduced Charge Transfer Mechanism," *J. Am. Chem. Soc.*, no. 133, pp. 16518–16523, 2011.
- [5] J. Lin *et al.*, "Direct Experimental Observation of Facet-Dependent SERS of Cu₂O Polyhedra," *Small*, vol. 14, no. 8, p. 1703274, 2018.
- [6] Z. Movasaghi, S. Rehman, and I. U. Rehman, "Raman spectroscopy of biological tissues," *Appl. Spectrosc. Rev.*, vol. 42, no. 5, pp. 493–541, 2007.
- [7] A. C. S. Talari, Z. Movasaghi, S. Rehman, and I. U. Rehman, "Raman spectroscopy of biological tissues," *Appl. Spectrosc. Rev.*, vol. 50, no. 1, pp. 46–111, 2015.
- [8] G. R. Lloyd *et al.*, "Discrimination between benign, primary and secondary malignancies in lymph nodes from the head and neck utilising Raman spectroscopy and multivariate analysis," *Analyst*, vol. 138, no. 14, pp. 3900–3908, 2013.
- [9] M. V. P. Chowdary, K. K. Kumar, J. Kurien, S. Mathew, and C. M. Krishna, "Discrimination of normal, benign, and malignant breast tissues by Raman spectroscopy," *Biopolymers*, vol. 83, no. 5, pp. 556–569, 2006.
- [10] C. Onogi, M. Motoyama, and H. Hamaguchi, "High concentration trans form unsaturated lipids detected in a HeLa cell by Raman microspectroscopy," *J. Raman Spectrosc.*, vol. 39, no. 5, pp. 555–556, 2008.
- [11] C. H. Liu *et al.*, "Human breast tissues studied by IR Fourier-transform Raman spectroscopy," in *Conference on Lasers and Electro-Optics*, 1991, p. CWF51.
- [12] J. Zhu *et al.*, "Surface-enhanced Raman spectroscopy investigation on human breast cancer cells," *Chem. Cent. J.*, vol. 7, no. 1, p. 37, 2013.
- [13] Y. H. Ong, M. Lim, and Q. Liu, "Comparison of principal component analysis and biochemical component analysis in Raman spectroscopy for the discrimination of apoptosis and necrosis in K562 leukemia cells," *Opt. Express*, vol. 20, no. 20, pp. 22158–22171, 2012.
- [14] S. Šašić, D. A. Clark, J. C. Mitchell, and M. J. Snowden, "Raman line mapping as a fast method for analyzing pharmaceutical bead formulations," *Analyst*, vol. 130, no. 11, pp. 1530–1536, 2005.
- [15] W. A. El-Said, T.-H. Kim, H. Kim, and J.-W. Choi, "Analysis of intracellular state based on controlled 3D nanostructures mediated surface enhanced Raman scattering," *PloS One*, vol. 6, no. 2, p. e15836, 2011.
- [16] P. Dittrich and N. Jakubowski, *Current trends in single cell analysis*. Springer, 2014.
- [17] J. L. Pichardo-Molina *et al.*, "Raman spectroscopy and multivariate analysis of serum samples from breast cancer patients," *Lasers Med Sci*, vol. 22, pp. 229–236.
- [18] H. Abramczyk and B. Brozek-Pluska, "Raman imaging in biochemical and biomedical applications. Diagnosis and treatment of breast cancer," *Chem. Rev.*, vol. 113, no. 8, pp. 5766–5781, 2013.
- [19] K. E. Shafer-Peltier *et al.*, "Raman microspectroscopic model of human breast tissue: implications for breast cancer diagnosis in vivo," *J. Raman Spectrosc.*, vol. 33, no. 7, pp. 552–563, 2002.
- [20] Y. H. Ong, M. Lim, and Q. Liu, "Comparison of principal component analysis and biochemical component analysis in Raman spectroscopy for the discrimination of apoptosis and necrosis in K562 leukemia cells," *Opt. Express*, vol. 20, no. 20, pp. 22158–22171, 2012.
- [21] A. B. de Carvalho *et al.*, "Chemotherapeutic response to cisplatin-like drugs in human breast cancer cells probed by vibrational microspectroscopy," *Faraday Discuss.*, vol. 187, pp. 273–298, 2016.
- [22] R. Sager, "Expression genetics in cancer: shifting the focus from DNA to RNA," *Proc. Natl. Acad. Sci.*, vol. 94, no. 3, pp. 952–955, 1997.
- [23] D. Bauer *et al.*, "Identification of differentially expressed mRNA species by an improved display technique (DDRT-PCR)," *Nucleic Acids Res.*, vol. 21, no. 18, pp. 4272–4280, 1993.
- [24] Z. Huang, A. McWilliams, H. Lui, D. I. McLean, S. Lam, and H. Zeng, "Near-infrared Raman spectroscopy for optical diagnosis of lung cancer," *Int. J. Cancer*, vol. 107, no. 6, pp. 1047–1052, 2003.
- [25] G. Zheng, F. Patolsky, Y. Cui, W. U. Wang, and C. M. Lieber, "Multiplexed electrical detection of cancer markers with nanowire sensor arrays," *Nat. Biotechnol.*, vol. 23, no. 10, p. 1294, 2005.
- [26] S. Laing, K. Gracie, and K. Faulds, "Multiplex in vitro detection using SERS," *Chem. Soc. Rev.*, vol. 45, no. 7, pp. 1901–1918, 2016.
- [27] S. Kim, J.-M. Kim, J.-E. Park, and J.-M. Nam, "Noble-Metal-Based Plasmonic Nanomaterials: Recent Advances and Future Perspectives," *Adv. Mater.*, 2018.

- [28] K. D. Osberg, M. Rycenga, G. R. Bourret, K. A. Brown, and C. A. Mirkin, "Dispersible Surface-Enhanced Raman Scattering Nanosheets," *Adv. Mater.*, vol. 24, no. 45, pp. 6065–6070, 2012.
- [29] D. Lin *et al.*, "Direct transfer of subwavelength plasmonic nanostructures on bioactive silk films," *Adv. Mater.*, vol. 24, no. 45, pp. 6088–6093, 2012.
- [30] Kyle D. Osberg, Matthew Rycenga, Gilles R. Bourret, Keith A. Brown, and Chad A. Mirkin, "Dispersible Surface-Enhanced Raman Scattering Nanosheets," *Advanced Mater.*, no. 24, pp. 6065–6070, 2012.
- [31] Jiawen Hu, Linghui Lu, Weiming He, Jiangaop Pan, Weiyu Wang, and Jiannan Xiang, "Ligand exchange based water-soluble, surface-enhanced Raman scattering-tagged gold nanorod probes with improved stability," *Chem. Phys. Lett.*, no. 513, pp. 241–245, 2011.
- [32] Jun Deng, Mengyun Yao, and Changyon Gao, "Cytotoxicity of gold nanoparticles with different structures and surface-anchored chiral polymers," *Acta Biomater.*, no. 53, pp. 610–618, 2017.
- [33] T. Kang, R. Guan, X. Chen, Y. Song, H. Jiang, and J. Zhao, "In vitro toxicity of different-sized ZnO nanoparticles in Caco-2 cells," *Nanoscale Res. Lett.*, vol. 8, no. 1, p. 496, Nov. 2013.
- [34] J. Zhou, N. S. Xu, and Z. L. Wang, "Dissolving behavior and stability of ZnO wires in biofluids: a study on biodegradability and biocompatibility of ZnO nanostructures," *Adv. Mater.*, vol. 18, no. 18, pp. 2432–2435, 2006.
- [35] Kim J S *et al.*, "Cellular uptake of magnetic nanoparticle is mediated through energydependent endocytosis in A549 cells," *J. Veterinary Sci.*, vol. 7, no. 4, pp. 321–6, Dec. 2006.
- [36] Sonya A. MacParland *et al.*, "Phenotype Determines Nanoparticle Uptake by Human Macrophages from Liver and Blood," *ACS Nano*, vol. 11, pp. 2428–2443, 2017.
- [37] D. Li, D.-W. Li, Y. Li, J. S. Fossey, and Y.-T. Long, "Cyclic electroplating and stripping of silver on Au@ SiO₂ core/shell nanoparticles for sensitive and recyclable substrate of surface-enhanced Raman scattering," *J. Mater. Chem.*, vol. 20, no. 18, pp. 3688–3693, 2010.

Reviewers' Comments:

Reviewer #1:

Remarks to the Author:

This is a revised manuscript after 3rd round of review, and changes made in revision is quite satisfactory. I recommend now this manuscript can be published as it is in Nature Communications. The length of the manuscript is slightly longer and it may be necessary to put some results in the SI.

Reviewer #2:

Remarks to the Author:

The authors have addressed a few of my concerns, but several others still remain unclear. I understand the the main objective of the study was to develop and demonstrate the ability of ZnO based nanostructures for SERS applications. However, the SERS application for cancer detection is poorly designed and is not supported by enough scientific data. In overall, I think the manuscript is not good enough for the publication in Nature Communications, but is more suitable for specialized journals in spectroscopy or material sciences.

1. I agree that the authors detected biomolecules in cells just like other previous work referred as [6-12] in the point-by-point response. However, these are not cancer-specific molecules, and thus detecting them does not directly lead to "cancer detection."

2. The authors used just 3 cell lines, which are too few for PCA or discriminant analysis considering the number of variables. The rule of thumb is that the number of observations should be much larger than the number of variables. The references 8 and 9 in the point-by-point response used PCA analyses based on thousands Raman spectral from 50-100 different samples and looked for higher levels of lipid and protein contents in cancer specimens. In this study, however, neither enough sample sizes nor specific variables are not provided.

3. The SERS maps in Figure 10 are only from small areas and there are no other corresponding data to support the accuracy of the mapping.

4. The measurements were done after air-drying cells, which could cause different shapes and biomolecule distributions depending on how these cells were dehydrated. This also causes particle aggregations inside of cells. If high SERS enhancements occur on the nanoparticle surface, uniform distribution of nanoparticles inside of cells is highly desired for accurate SERS mapping and therefore particle aggregation should be avoided.

5. I agree with the authors about the importance of multiplexed sensing. However, highly multiplexed SERS sensing both in vitro and in vivo sensing have been also demonstrated using metallic nanoparticles. The challenges with metallic nanoparticles can be technically addressed and their in vitro and in vivo applications have been successfully demonstrated.

S. Liang et al., Surface-enhanced Raman spectroscopy for in vivo biosensing, Nature Reviews Chemistry, 1, 0060 (2017).

L. A. Lane et al., SERS Nanoparticles in Medicine: From Label-Free Detection to Spectroscopic Tagging, Chemical Reviews, 115, 10489 (2015).

6. I agree that studying the SERS effects of nonplasmonic materials is scientifically important, but it is not clear yet if they are better probes than metallic particles for SERS sensing.

7. In the SERS enhancement factor (EF) calculation, it is still unclear how the authors calculated N_{surf} of the particles. 10 μ g of the solution was applied to the particles and air-dried, which will lead to the formation of molecular multiplayers on the surface. It is also unclear how many

particles contributed to the Raman intensity, I_{probe} , in a given laser spot. For accurate EF assessment, it is very critical that the amount of molecules on the particle surface is carefully calculated and thus forming a monolayer is highly desired. However, the air-drying method used in the study will likely form multilayers that could lead to higher estimation of EF.

Reviewer #3:

None

Point by point response to the reviewers' comments

With reference to your letter regarding revision of the manuscript entitled "Non Plasmonic Semiconductor Quantum SERS Probe – Pathway For In-Vitro Detection" by Rupa Haldavnekar, Dr. Bo Tan and I, we will like to thank you and the reviewers for the feedback on the revised manuscript. The reviewer's comments have been addressed by bringing more clarity to the analysis, discussion and methods.

1. Discussion about cell discrimination was revised for more in-depth analysis of the data.
2. New tables for Eigenvalues of F1, F2 & F3 in PCA and factor loadings of F1 were added in supplementary information.
3. More clarity to the manuscript was brought by revising the introduction, methods, and supplementary information.

Point-by-point responses to the reviewer's remarks are outlined below.

Comment 1: *"I agree that the authors' detected biomolecules in cells just like other previous work referred as [6-12] in the point-by-point response. However, these are not cancer-specific molecules, and thus detecting them does not directly lead to cancer detection."*

Response: The authors thank the reviewer for this comment.

1. The main objective of the manuscript was to develop a non-plasmonic quantum probe for Surface Enhanced Raman Scattering as a method for label-free, simultaneous, multiplex detection of cancer.
2. Due to the complexity of the disease and its heterogeneous nature, only certain cancer biomarker detection is not adequate to get complete information about this disease [1]. So, use of the information gathered from the entire cell spectra for the disease diagnosis is a more comprehensive alternative method compared to the more traditional method of detection of cancer biomarkers. That way, the data from the entire spectra which can provide rich information about the disease does not get lost.
3. The authors have followed established protocols from the literature, by use of multivariate analyses, taking into account the rich information from the entire spectra [2], [3], [4], [5]. In this study, combination of multivariate analysis of Principal components (PCA) and Discriminant analysis (DA) were employed. Raman spectra of cancer cells and normal cells differ due to the variation in metabolic pathways, changed regulation of surface receptors etc [6]. Hence, cancer cell lines produce fingerprint Raman spectra which enable detection of cancer by variety of methods other than the measurement of cancer biomarkers.
4. The authors also employed analysis of ratios between certain Raman bands in combination with qualitative analysis of the characteristic Raman peaks to substantiate the results. The ratio analysis of Raman bands at 1445 cm⁻¹ and 1655 cm⁻¹ (fig 9) has been used many times in the literature for classification of cancer vs non-cancer, metastatic vs benign etc [5] as this decision ratio (I_{1445}/I_{1655}) can separate cancer cells from normal cells in combination with the qualitative analysis of characteristic Raman peaks.
5. Further, the method of detection of cancer biomarkers can also not be treated as a direct disease detection method because plasmonic nanoparticles need labels (functionalization with the Raman reporters or tags) for identification of cancer biomarkers. This type of detection is based on identification of the Raman reporters (labels binding to the cancer biomarkers) resonating with the excitation wavelength. Hence, this methodology also does not directly detect the disease [6].

The use of variety of analyses for label-free, multiplex detection of cancer was reinforced in the manuscript by revision of introduction, results and methods bringing more clarity as follows:

1. Introduction (Page 5) : “Discrimination between cancer and normal cells was done on the basis of ratio of peak intensities of lipids and proteins (I_{1445}/I_{1654}) which acts as a decision ratio providing separation between cancer and normal cells. Raman bands were analyzed for peak positions, intensity to demonstrate differences between cancer and non-cancer cells. The results were also substantiated by statistical analysis using combination of principal component analysis and discriminant analysis.”
2. Methods (Page 34): “In order to distinguish between cancer and non-cancer cell lines, ratio of Raman intensities at 1445 cm^{-1} and 1655 cm^{-1} were compared. This ratio has been used for classification between normal and malignant cells in the brain, breast, colon and cervix [5]. The sensitivities of the band for CH₂ scissoring for proteins at 1445 cm^{-1} and the band for C=O stretching of collagen and elastin for lipids at 1655 cm^{-1} , for histological abnormalities has been widely utilized as a test for malignancy in the literature. The decision ratio (I_{1445}/I_{1655}) was able to separate cancer cells from normal cells. Scatter plots for all three cell lines were plotted for this classification.

Although Raman spectroscopy can provide fingerprint spectra of various cancer cell lines, the peaks generated due to proteins, lipids, nucleic acids, polysaccharides are very complex and can overlap over a broad band. There is a possibility of introducing inaccuracies in the interpretation of data when the peaks are analyzed qualitatively. In order to reduce the errors due to guessing of biochemical components, qualitative as well as quantitative analysis of Raman bands was performed [7]. Combination of multivariate analysis of Principal components (PCA) and Discriminant analysis (DA) was employed with ‘XLstat’. PCA analysis is a non-parametric approach which does not need any explicit background model [8]. Dimensionality of the data can be reduced with PCA, as mathematical decomposition of Raman spectra into principal components is done. PCA is effectively used in Raman studies for discrimination of cells [7] [9] [10]. Cells were identified using an inverted microscope and spectra were gathered from 200 cm^{-1} to 3275 cm^{-1} wavenumbers (3075 observations per sample). The dimensionality of 3075 observations recorded for each spectra were reduced by getting three principal components (F1, F2 & F3), so three PCs per spectra were obtained. A three dimensional cluster graph of F1 Vs F2 Vs F3 was plotted for all 60 samples. Similarly, for discriminant analysis, a cluster graph of F1 Vs F2 (Discriminant functions F1 & F2) was plotted for all 60 samples. Since cells contain over 10,000 to 15,000 mRNAs, it is possible to understand approximately 150 tumor suppressing genes from the study of RNA [11], [12]. The intensity values of RNA wavenumber at 915 cm^{-1} were used as an independent variable for discriminant analysis for discrimination of cell lines.”

Comment 2: “The authors used just 3 cell lines, which are too few for PCA or discriminant analysis considering the number of variables. The rule of thumb is that the number of observations should be much larger than the number of variables. The references 8 and 9 in the point-by-point response used PCA analyses based on thousands Raman spectral from 50-100 different samples and looked for higher levels of lipid and protein contents in cancer specimens. In this study, however, neither enough sample sizes nor specific variables are not provided.”

Response: The authors thank the reviewer for this comment.

1. The authors will like to bring more clarity to the data analysis here.
2. In this experiment 20 spectra from cells at various locations for each cancer cell line were gathered resulting in total 60 samples. The protocol for gathering of spectra was as follows: cells were identified using an inverted microscope and spectra were gathered from 200 cm^{-1} to 3275 cm^{-1} wavenumbers. The dimensionality of the observations recorded for each spectra were reduced by getting three principal components (F1, F2 & F3), so three PCs per spectra were obtained. A three dimensional cluster graph of F1 Vs F2 Vs F3 was plotted for all 60 samples. Similarly, for discriminant analysis, a cluster graph of F1 Vs F2 (Discriminant functions F1 & F2) was plotted for all 60 samples. Clear clustering was observed for different cell lines, showing ability of the quantum probe for cancer & non-cancer discrimination.
3. Presence of collagen was the highest discriminating factor (wavenumber 1439 cm^{-1}), discussion about PCA was revised to incorporate the discussion on the discriminating PC (F1) to bring more clarity. Eigen values and loadings of F1 were added in the form of a table in supplementary material.

4. The authors have followed this standard protocol from the literature (20 spectra [7], 28 spectra [5], 45 spectra [3]) which is in alignment with the number of spectra captured in this study (60 spectra).

The methodology was revised for more clarity to this protocol as follows

1. Methods (Page 35): "In order to reduce the errors due to guessing of biochemical components, qualitative as well as quantitative analysis of Raman bands was performed [7]. Combination of multivariate analysis of Principal components (PCA) and Discriminant analysis (DA) was employed with 'XLstat'. PCA analysis is a non-parametric approach which does not need any explicit background model [8]. Dimensionality of the data can be reduced with PCA, as mathematical decomposition of Raman spectra into principal components is done. PCA is effectively used in Raman studies for discrimination of cells [7] [9] [10]. Cells were identified using an inverted microscope and spectra were gathered from 200 cm^{-1} to 3275 cm^{-1} . The dimensionality of the observations recorded for each spectra were reduced by getting three principal components (F1, F2 & F3). A three dimensional cluster graph of F1 Vs F2 Vs F3 was plotted for all 60 samples. Similarly, for discriminant analysis, a cluster graph of F1 Vs F2 (Discriminant functions F1 & F2) was plotted for all 60 samples."

The discussion on PCA was revised as follows:

1. Results (Page 25): "The cancer and non-cancer classification was further substantiated by two multivariate analyses (Principal Component Analysis-PCA and Discriminant Analysis-DA). Eigen values of F1 , F2 and F3 were used for a 3-D PC scatter plot. F1 interpreted 45.96 %, F2 20.67 % and F3 14.02 % of variances. Top 3 principal components contributed to 80.68 % of cumulative contribution covering most of the information from original spectra in the three dimensional space. Scores of F1 provided reasonable classification amongst the three cell lines. Clusters of all three cell lines were clearly defined. Refer to 'Supplementary Table 7' for Factor F1 loading indicating collagen as the highest discriminating factor. The peak at 917 cm^{-1} which belongs to the ribose vibration of RNA was explored to achieve discrimination amongst cancer and non-cancer cells. Clear clustering based on cell type was observed. It can be seen from the centroids, that HeLa & MDAMB (cancer cells) show more activity as compared to normal cells. This can also be interpreted as more tumor suppressing genes from mRNA are altered in cancers as compared to normal cells. Discriminant analysis provided 100% classification rate with first two functions F1 & F2 demonstrating clear clusters based on cell type."
2. 'Supplementary Table 6 & 7' for Eigenvalues and factor loadings of F1 was added in the supplementary information.

Comment 3: "The SERS maps in Figure 10 are only from small areas and there are no other corresponding data to support the accuracy of the mapping."

Response: The authors thank the reviewer for this comment.

1. Main objective of this manuscript was to achieve single cell level SERS ultra-sensitivity with the non-plasmonic quantum probe. The established Raman peak assignments were obtained from the literature for mapping [6], [7], [8], [9], [10],[11], [12]. The Raman peaks in the mapped data in this study were concurrent and in agreement with the literature. The small areas mapping was undertaken in order to establish the limit of detection up to a single cell level. Bright field images of the cells are available in figure 10.
2. The authors have used standard software and protocols for Raman mapping of the biomolecules.
3. Isolated cells as well as Raman spectra are available in figure 10. Single cell mapping was undertaken for three different cell lines.
4. No labels were used in this study to demonstrate the label-free sensing application. So, the authors used an alternative approach and checked the correctness of the calibration of the Raman instrumentation and Wire 3.3 software before starting of the experiment. The peak assignment for silicon was checked. After confirmation of calibration, two Raman reporter dyes - R6G and crystal violet, two more Analytes (4ATP &

4MBA) was tested. The characteristic peaks for all the test molecules were in agreement with the peaks reported in the literature.

5. The authors then used Raman assignments from the literature for cellular data and mapped the biomolecules.

This alternative approach to ensure correctness of the data was reinforced in the manuscript as follows:

1. Methods (Page 35): "Single cells on the nanostructures were identified with the help of Leica DMI 6000 epifluorescence microscope. Isolated cells were visible in the Raman mapping experiment and the brightfield images of these cells were obtained. Calibration of the software was undertaken using silicon. After confirming the software calibration, Raman spectra of four analytes crystal violet, R6G, 4ATP & 4 MBA were obtained in order to ensure correctness of the data. Once the accuracy for the Raman peaks was confirmed, Raman mapping was carried out at 3 X 3 steps for the entire selected area. 2 D maps of intensity at 752 cm^{-1} , 915 cm^{-1} , 1030 cm^{-1} , 1334 cm^{-1} , 1655 cm^{-1} , 2854 cm^{-1} , 2940 cm^{-1} wavenumbers were obtained. These wavenumbers representing various biomolecular signatures were obtained from the literature (Refer 'Supplementary Table 5'). The individual maps for each wavenumber were then merged to get the final map of single cell using wire 3.3 software. The single cell maps were built with the help of in-built functionality of wire 3.3 software."

The sensitivity of the quantum probe to attain single cell level detection was reinforced in the manuscript as follows:

1. Results (Page 29): "In order to obtain the limit of detection, Raman mapping of cells was undertaken [63], [11]. From the Raman map (Figure 10), more presence of proteins and lipids was indicated. Since proteins were present mostly in cytoplasm and lipids in the cell membrane, more affinity of the quantum probe to cytoplasm as well as cell membrane is indicated from the color brightness related to signal intensity [64]. We were able to obtain the map of a single cell demonstrating the ability of SERS from the ZnO based Semiconductor quantum probe for single cell analysis [65]."

Comment 4: *"The measurements were done after air-drying cells, which could cause different shapes and biomolecule distributions depending on how these cells were dehydrated. This also causes particle aggregations inside of cells. If high SERS enhancements occur on the nanoparticle surface, uniform distribution of nanoparticles inside of cells is highly desired for accurate SERS mapping and therefore particle aggregation should be avoided."*

Response: The authors thank the reviewer for this comment.

1. Phenotype of the cells determines the nanoparticle uptake [15]. The cellular internalization by endocytosis or diffusion results in particles aggregating inside the cells in the vacuoles [16]. The internalization of the quantum probe for all cell lines was validated and was in agreement with the current understanding of cellular mechanism recently reported [17], [18].
2. In this study, the quantum probe was internalized in the form of a few aggregates as well as the quantum probes were dispersed throughout the cell. The distribution of quantum probes throughout the cells was evident from the in-depth TEM analysis (Fig 6, 9, 10) as well as EDX elemental mapping (Fig 6). This in-depth analysis was conducted for three different cell lines.
3. This dispersion of quantum probes throughout the cells resulted in more stable Raman spectra (Fig 10). Since the dominating mechanism for the SERS enhancement with the ZnO based non-plasmonic quantum probes was due to the charge transfer mechanism, which is typically weaker enhancement mechanism, no erratic increase or decrease in the intensity of the peaks was observed due to aggregates.
4. It is evident from the literature that experimentation with air dried cells results in better enhancement and more sensitive spectra as compared to experimentation with hydrated cells [16]. The authors had employed natural air-drying at room temperature for all the cell lines in order to maintain same experimental conditions.
5. Many studies use air-dried cells as a standard protocol, when only chemical information is needed. In our case, the expectation of the results was not the exploratory study like changes in the biomolecules or response to the drug interaction which can be sensitive to changes in the shapes and different biomolecular distributions. Our study was more of a label free diagnostic application with "Cancerous" or "Non-Cancerous" response with ultra-high sensitivity up to a single cell level.

6. So, the few aggregates as a result of endocytosis present in the cells had no adverse effect on the end result of cancer diagnosis.

More clarity on the air-drying protocol was reinforced in the manuscript as follows:

1. Methods (Page 33): “The protocol of air drying of cells at room temperature was followed because the literature suggests that air dried cells result in better enhancement and more sensitive spectra as compared to experimentation with hydrated cells [16]. The expectation from this study was of a diagnostic application with “Cancerous” or “Non-Cancerous” response up to a single cell level detection, so air drying of the samples did not have adverse effect on the SERS spectra.”
2. Additional reference for air drying at room temperature protocol was added- H.-W. Tang, X. B. Yang, J. Kirkham, and D. A. Smith, “Probing intrinsic and extrinsic components in single osteosarcoma cells by near-infrared surface-enhanced Raman scattering,” *Anal. Chem.*, vol. 79, no. 10, pp. 3646–3653, 2007.

Comment 5: “I agree with the authors about the importance of multiplexed sensing. However, highly multiplexed SERS sensing both in vitro and in vivo sensing have been also demonstrated using metallic nanoparticles. The challenges with metallic nanoparticles can be technically addressed and their in vitro and in vivo applications have been successfully demonstrated.

S. Liang et al., *Surface-enhanced Raman spectroscopy for in vivo biosensing*, *Nature Reviews Chemistry*, 1, 0060 (2017).

L. A. Lane et al., *SERS Nanoparticles in Medicine: From Label-Free Detection to Spectroscopic Tagging*, *Chemical Reviews*, 115, 10489 (2015).”

Response: The authors will like to thank the reviewer for providing good reference review papers and highlighting significant importance of using non-plasmonic quantum probe for label-free, simultaneous, multiplexed, in-vitro SERS detection of cancer. These papers validate our claim that the current state of the art with respect to the plasmonic nanoparticles is based on biomolecular assays carried in test tubes. These reviews demonstrate use of gold and silver for multiplexed applications with multiple Raman reporter labels. The label free applications did not demonstrate simultaneous detection or in-vitro/in-vivo detection. Also, plasmonic nanoparticles need to be coated with surfactants to quench the cytotoxicity which can result in a contaminated spectra or presence of artefacts or can adversely affect the integrity of the cellular structure[19]. Hence, there is a need for a non-plasmonic, label free, multiplexed in-vitro SERS detection of cancer. The authors have elaborated this need in the introduction of the manuscript.

1. The main objective of this manuscript was to develop plasmon-free, label-free, simultaneous multiplex SERS detection of cancer using a semiconductor quantum probe. This study explores the possibility of addressing the challenges posed by the plasmonic nanoparticles by developing non-plasmonic quantum scaled probes. The authors have used in-vitro analysis to get more relevant information by reporting of local micro-environment along-with the molecular nano-environment for a more realistic situation than biochemical assays carried out with purified biomolecules in a test tube[20]. Information on specific biomarkers responsible for cancer is inadequate to provide complete information about complex disease like cancer [1]. So simultaneous multiplexed detection is important. Labeled nanoparticles can generate contaminated spectra with experimental artefacts due to the use of labels [21]. Labels can also adversely affect the integrity of cellular structure. So, label-free detection is also equally important.
2. S. Liang et al., in “Surface-enhanced Raman spectroscopy for in vivo biosensing” have described nanosensor design with plasmonic nanoparticles which needs Raman reporters (labels), targeting molecules and protective coating for in-vivo application.
3. The label free approach described by L. A. Lane et al., in “SERS Nanoparticles in Medicine: From Label-Free Detection to Spectroscopic Tagging” with the plasmonic nanoparticles was to detect specific biomolecule like glucose, proteins, nucleic acids removed from the cells which does not in-vitro/ in-vivo application.

This ability of the non-plasmonic quantum probe for label-free, simultaneous multiplexed in-vitro detection was reinforced in the manuscript as follows:

1. Introduction (Page 2): “Typically, plasmonic metal nanoparticles of gold, silver are used for SERS due to their ability to generate strong electromagnetic enhancement [22]. The substantial enhancement

observed with plasmonic metal nanoparticles it suffers from coagulation[23], selectivity[24], cost, optical loss, limited wavelength range and adverse biocompatibility [21]. In order for the highly localized Raman hot spots to remain discrete, plasmonic materials often need surfactants for good SERS response. This is very challenging due to uncontrolled agglomeration of these materials[25]. This type of materials also need to be functionalized for specific targeting with SERS active Raman tags [26] which can result in a contaminated spectra affecting adversely the integrity of cellular structure [19]. This limits Plasmon based label-free, multiplex SERS diagnostics [1].”

Comment 6: “I agree that studying the SERS effects of non-plasmonic materials is scientifically important, but it is not clear yet if they are better probes than metallic particles for SERS sensing.”

Response: In this manuscript, the authors have introduced a label free, simultaneous multiplexed *in-vitro* SERS diagnosis of cancer from a non-plasmonic ZnO based semiconductor material. Since non-plasmonic materials at nano-scale typically provide a poor SERS response, the authors have reduced the size of a biocompatible ZnO based probe to quantum scale and induced defect engineering by stacking faults and oxygen vacancies. The authors observed a new phenomenon of exponential increase in SERS from ZnO based semiconductor quantum probe. The quantum probe was decorated on a nano dendrite platform which was functionalized for cell adherence and proliferation by mimicking the extracellular matrix. This type of label-free, simultaneous multiple SERS detection for *in-vitro* diagnosis of cancer from a non-plasmonic, biocompatible and label-free ZnO based semiconductor material shows the possibilities of future diagnostic devices providing ultra-sensitive and complete cellular information which is significant for early diagnosis of the disease as well as for disease monitoring.

Comment 7: “In the SERS enhancement factor (EF) calculation, it is still unclear how the authors calculated N_{surf} of the particles. 10 μg of the solution was applied to the particles and air-dried, which will lead to the formation of molecular multilayers on the surface. It is also unclear how many particles contributed to the Raman intensity, I_{probe} , in a given laser spot. For accurate EF assessment, it is very critical that the amount of molecules on the particle surface is carefully calculated and thus forming a monolayer is highly desired. However, the air-drying method used in the study will likely form multilayers that could lead to higher estimation of EF.”

Response: The authors will like to thank the reviewer for this comment.

1. The authors have used the most commonly used protocol – the “Drop and Dry” protocol, where 10 μL of analyte is deposited on substrate and allowed to dry naturally. The authors have used this protocol from the literature [27], [28].
2. The authors’ current method produces the least SERS enhancement factor, so the authors have used this conservative method of the enhancement factor calculation in this study which is not going to likely lead to higher estimation of EF. This is evident from the study undertaken by J. A. Guicheteau et-al which compared various protocols and reported EF calculations. The Drop and Dry protocol produced minimum EF [29].
3. Also, for consistency with the *in-vitro* protocol, which was with air drying cells naturally at room temperature, the authors have followed “Drop and Dry” protocol in the EF calculation experiment.

References

- [1] G. Zheng, F. Patolsky, Y. Cui, W. U. Wang, and C. M. Lieber, “Multiplexed electrical detection of cancer markers with nanowire sensor arrays,” *Nat. Biotechnol.*, vol. 23, no. 10, p. 1294, 2005.
- [2] C. Krafft, G. Steiner, C. Beleites, and R. Salzer, “Disease recognition by infrared and Raman spectroscopy,” *J. Biophotonics*, vol. 2, no. 1-2, pp. 13–28, 2009.
- [3] D. Lin *et al.*, “Colorectal cancer detection by gold nanoparticle based surface-enhanced Raman spectroscopy of blood serum and statistical analysis,” *Opt. Express*, vol. 19, no. 14, pp. 13565–13577, 2011.
- [4] K. Gajjar *et al.*, “Diagnostic segregation of human brain tumours using Fourier-transform infrared and/or Raman spectroscopy coupled with discriminant analysis,” *Anal. Methods*, vol. 5, no. 1, pp. 89–102, 2013.
- [5] Z. Huang, A. McWilliams, H. Lui, D. I. McLean, S. Lam, and H. Zeng, “Near-infrared Raman spectroscopy for optical diagnosis of lung cancer,” *Int. J. Cancer*, vol. 107, no. 6, pp. 1047–1052, 2003.
- [6] L. A. Lane, X. Qian, and S. Nie, “SERS nanoparticles in medicine: from label-free detection to spectroscopic tagging,” *Chem Rev*, vol. 115, no. 19, pp. 10489–10529, 2015.
- [7] Y. H. Ong, M. Lim, and Q. Liu, “Comparison of principal component analysis and biochemical component analysis in Raman spectroscopy for the discrimination of apoptosis and necrosis in K562 leukemia cells,” *Opt. Express*, vol. 20, no. 20, pp. 22158–22171, 2012.

- [8] A. B. de Carvalho *et al.*, "Chemotherapeutic response to cisplatin-like drugs in human breast cancer cells probed by vibrational microspectroscopy," *Faraday Discuss.*, vol. 187, pp. 273–298, 2016.
- [9] S. Šašić, D. A. Clark, J. C. Mitchell, and M. J. Snowden, "Raman line mapping as a fast method for analyzing pharmaceutical bead formulations," *Analyst*, vol. 130, no. 11, pp. 1530–1536, 2005.
- [10] Y. H. Ong, M. Lim, and Q. Liu, "Comparison of principal component analysis and biochemical component analysis in Raman spectroscopy for the discrimination of apoptosis and necrosis in K562 leukemia cells," *Opt. Express*, vol. 20, no. 20, pp. 22158–22171, 2012.
- [11] R. Sager, "Expression genetics in cancer: shifting the focus from DNA to RNA," *Proc. Natl. Acad. Sci.*, vol. 94, no. 3, pp. 952–955, 1997.
- [12] D. Bauer *et al.*, "Identification of differentially expressed mRNA species by an improved display technique (DDRT-PCR)," *Nucleic Acids Res.*, vol. 21, no. 18, pp. 4272–4280, 1993.
- [13] Z. Movasaghi, S. Rehman, and I. U. Rehman, "Raman spectroscopy of biological tissues," *Appl. Spectrosc. Rev.*, vol. 42, no. 5, pp. 493–541, 2007.
- [14] A. C. S. Talari, Z. Movasaghi, S. Rehman, and I. U. Rehman, "Raman spectroscopy of biological tissues," *Appl. Spectrosc. Rev.*, vol. 50, no. 1, pp. 46–111, 2015.
- [15] G. R. Lloyd *et al.*, "Discrimination between benign, primary and secondary malignancies in lymph nodes from the head and neck utilising Raman spectroscopy and multivariate analysis," *Analyst*, vol. 138, no. 14, pp. 3900–3908, 2013.
- [16] M. V. P. Chowdary, K. K. Kumar, J. Kurien, S. Mathew, and C. M. Krishna, "Discrimination of normal, benign, and malignant breast tissues by Raman spectroscopy," *Biopolymers*, vol. 83, no. 5, pp. 556–569, 2006.
- [17] C. Onogi, M. Motoyama, and H. Hamaguchi, "High concentration trans form unsaturated lipids detected in a HeLa cell by Raman microspectroscopy," *J. Raman Spectrosc.*, vol. 39, no. 5, pp. 555–556, 2008.
- [18] C. H. Liu *et al.*, "Human breast tissues studied by IR Fourier-transform Raman spectroscopy," in *Conference on Lasers and Electro-Optics*, 1991, p. CWF51.
- [19] J. Zhu *et al.*, "Surface-enhanced Raman spectroscopy investigation on human breast cancer cells," *Chem. Cent. J.*, vol. 7, no. 1, p. 37, 2013.
- [20] P. Dittrich and N. Jakubowski, *Current trends in single cell analysis*. Springer, 2014.
- [21] D. D. Carlo and L. P. Lee, *Dynamic single-cell analysis for quantitative biology*. ACS Publications, 2006.
- [22] P. Actis *et al.*, "Electrochemical nanoprobe for single-cell analysis," *Acs Nano*, vol. 8, no. 1, pp. 875–884, 2014.
- [23] Sonya A. MacParland *et al.*, "Phenotype Determines Nanoparticle Uptake by Human Macrophages from Liver and Blood," *ACS Nano*, vol. 11, pp. 2428–2443, 2017.
- [24] H.-W. Tang, X. B. Yang, J. Kirkham, and D. A. Smith, "Probing intrinsic and extrinsic components in single osteosarcoma cells by near-infrared surface-enhanced Raman scattering," *Anal. Chem.*, vol. 79, no. 10, pp. 3646–3653, 2007.
- [25] E. A. Sykes, J. Chen, G. Zheng, and W. C. Chan, "Investigating the impact of nanoparticle size on active and passive tumor targeting efficiency," *ACS Nano*, vol. 8, no. 6, pp. 5696–5706, 2014.
- [26] K. M. Tsoi *et al.*, "Mechanism of hard-nanomaterial clearance by the liver," *Nat. Mater.*, vol. 15, no. 11, p. 1212, 2016.
- [27] Jun Deng, Mengyun Yao, and Changyon Gao, "Cytotoxicity of gold nanoparticles with different structures and surface-anchored chiral polymers," *Acta Biomater.*, no. 53, pp. 610–618, 2017.
- [28] S. Kim, J.-M. Kim, J.-E. Park, and J.-M. Nam, "Noble-Metal-Based Plasmonic Nanomaterials: Recent Advances and Future Perspectives," *Adv. Mater.*, 2018.
- [29] L. Vigderman, B. P. Khanal, and E. R. Zubarev, "Functional gold nanorods: synthesis, self-assembly, and sensing applications," *Adv. Mater.*, vol. 24, no. 36, pp. 4811–4841, 2012.
- [30] K. D. Osberg, M. Rycenga, G. R. Bourret, K. A. Brown, and C. A. Mirkin, "Dispersible Surface-Enhanced Raman Scattering Nanosheets," *Adv. Mater.*, vol. 24, no. 45, pp. 6065–6070, 2012.
- [31] D. Lin *et al.*, "Direct transfer of subwavelength plasmonic nanostructures on bioactive silk films," *Adv. Mater.*, vol. 24, no. 45, pp. 6088–6093, 2012.
- [32] Kyle D. Osberg, Matthew Rycenga, Gilles R. Bourret, Keith A. Brown, and Chad A. Mirkin, "Dispersible Surface-Enhanced Raman Scattering Nanosheets," *Advanced Mater.*, no. 24, pp. 6065–6070, 2012.
- [33] Jiawen Hu, Linghui Lu, Weiming He, Jianguo Pan, Weiyu Wang, and Jiannan Xiang, "Ligand exchange based water-soluble, surface-enhanced Raman scattering-tagged gold nanorod probes with improved stability," *Chem. Phys. Lett.*, no. 513, pp. 241–245, 2011.
- [34] F. A. Harraz, A. A. Ismail, H. Bouzid, S. A. Al-Sayari, A. Al-Hajry, and M. S. Al-Assiri, "Surface-enhanced Raman scattering (SERS)-active substrates from silver plated-porous silicon for detection of crystal violet," *Appl. Surf. Sci.*, vol. 331, pp. 241–247, 2015.
- [35] C. Yuan *et al.*, "Single clusters of self-assembled silver nanoparticles for surface-enhanced Raman scattering sensing of a dithiocarbamate fungicide," *J. Mater. Chem.*, vol. 21, no. 40, pp. 16264–16270, 2011.
- [36] J. A. Guicheteau, A. Tripathi, E. D. Emmons, S. D. Christesen, and A. W. Fountain, "Reassessing SERS enhancement factors: using thermodynamics to drive substrate design," *Faraday Discuss.*, vol. 205, pp. 547–560, 2017.